# BISLERi: Ask Your Neural Network Not To Forget In Streaming Learning Scenarios

## Abstract

This paper introduces a new method for *class-incremental streaming learning*. In streaming learning, a learner encounters one single training example at a time and is constrained to: (*i*) utilize each sample only once, i.e., single-pass learning, (*ii*) adapt the parameters immediately, (*iii*) effectively predict on any new example at any time step without involving any additional computation, i.e., it can be evaluated anytime, and (*iv*) minimize the storage cost. Moreover, in streaming setting, the input data-stream cannot be assumed i.i.d, that is, there can be a temporal coherence in the input data-stream. Finally, the class-incremental learning implies that the learner does not require any task-id for the inference. A wide variety of the existing lifelong learning approaches are either designed to utilize more than one example once/multiple times or not optimized for the fast update or anytime inference. The premise of their designs, as well as other aspects (e.g., memory buffer/replay size, the requirement of fine-tuning), render some of the existing methods sub-optimal, if not ill-suited, for streaming learning setup. We propose a streaming Bayesian framework that enables fast parameter update of the network, given a single example, and allows it to be evaluated anytime. In addition, we also apply an implicit regularizer in the form of snap-shot self-distillation to effectively minimize the information loss further. The proposed method utilizes a tiny-episodic memory buffer and replays to conform with the streaming learning constraints. We also propose an efficient online memory replay and buffer replacement policies that significantly boost the model's performance. Extensive experiments and ablations on multiple datasets in different scenarios demonstrate the superior performance of our method over several strong baselines.

## 1 Introduction

In this paper, we aim to achieve continual learning by solving an extremely restrictive form of lifelong learning, i.e., *'streaming learning'* (Gama et al., 2013) in the deep neural networks (Hayes et al., 2019a;b). Most of the existing popular and successful methods in continual learning operate in incremental batch learning (IBL) scenarios (Rusu et al., 2016; Shin et al., 2017; Kirkpatrick et al., 2017; Wu et al., 2018; Aljundi et al., 2018a; Nguyen et al., 2017; Mallya & Lazebnik, 2018). In IBL, it is assumed that the current task data is available in batches during training, and the learner visits them sequentially multiple times. However, these methods are ill-suited in a rapidly changing environment, where the learner needs to quickly adapt to an important (since we cannot wait to adapt) and rare (as we may not collect batch) streaming data with no catastrophic forgetting (McCloskey & Cohen, 1989; French, 1999). For example, consider a real-world scenario, where an autonomous agent, such as an autonomous car, might meet with a rare incident/accident, then it could be lifesaving if it can be trained incrementally with that single example without any forgetting. It would be impractical, if not infeasible, to wait and aggregate a batch of samples to train the autonomous agent, as in this case, we may not collect a batch due to its rare nature.

The ability to continually learn effectively from streaming data with no catastrophic forgetting (McCloskey & Cohen, 1989; French, 1999) is a challenging problem and this has not received widespread attention (Hayes et al., 2019a;b). However, its utility is apparent, as it enables the practical deployment of autonomous AI agents. Hayes et al. (2019b) argued that streaming learning is more closer to biological learning than

Table 1: Categorization of the baseline approaches depending on the underlying simplifying assumptions they impose. In $\zeta(n)$, $n$ represents the number of gradient steps required to train the corresponding model. $\zeta(n) \gg \zeta(k) \geq \zeta(2) > \zeta(1)$. '-' indicates, we are unable to find the exact value.
Note: Although GEM (Lopez-Paz & Ranzato, 2017) performs a single gradient update for online learning, it does solve a Quadratic Program with inequality constraints, which is a computationally expensive operation; therefore, we consider it costs higher than a single gradient update. Furthermore, MIR (Aljundi et al., 2019) requires two gradient update, where it first performs a virtual gradient update to select maximally interfered samples from memory, and then performs another gradient update to finally update the network parameters. We provide detailed discussion on each column in the appendix.

| Methods | Type | Bayesian Framework | Batch-Size ($N_t$) | Fine-tunes | Single Pass Learning | CIL | Subset Buffer Replay | Training Time | Inference Time | Violates Any CISL Constraint | Memory Capacity | Regularization Based | Memory Based |
|---|---|---|---|---|---|---|---|---|---|---|---|---|---|
| | | | | | | 'Class-Incremental Streaming Learning' (CISL) Crucial Properties | | | | | | | |
| EWC (Kirkpatrick et al., 2017) | Batch | ✗ | $N_t \gg 1$ | ✗ | ✗ | ✗ | n/a | $\zeta(n)$ | $\zeta(1)$ | ✓ | n/a | ✓ | ✗ |
| MAS (Aljundi et al., 2018a) | Batch | ✗ | $N_t \gg 1$ | ✗ | ✗ | ✗ | n/a | $\zeta(n)$ | $\zeta(1)$ | ✓ | n/a | ✓ | ✗ |
| SI (Zenke et al., 2017) | Batch | ✗ | $N_t \gg 1$ | ✗ | ✗ | ✗ | n/a | $\zeta(n)$ | $\zeta(1)$ | ✓ | n/a | ✓ | ✗ |
| VCL (Nguyen et al., 2017) | Batch | ✓ | $N_t \gg 1$ | ✗ | ✗ | ✗ | n/a | $\zeta(n)$ | $\zeta(1)$ | ✓ | n/a | ✓ | ✗ |
| Coreset VCL (Nguyen et al., 2017) | Batch | ✓ | $N_t \gg 1$ | ✓ | ✗ | ✗ | ✗ | $\zeta(n)$ | $\zeta(n)$ | ✓ | - | ✓ | ✓ |
| Coreset Only (Farquhar & Gal, 2018) | Batch | ✓ | $N_t \gg 1$ | ✓ | ✗ | ✗ | ✗ | $\zeta(n)$ | $\zeta(n)$ | ✓ | - | ✗ | ✓ |
| GDumb (Prabhu et al., 2020) | Online | ✗ | $N_t \gg 1$ | ✓ | ✗ | ✓ | ✗ | $\zeta(1)$ | $\zeta(n)$ | ✓ | - | ✗ | ✓ |
| TinyER (Chaudhry et al., 2019) | Online | ✗ | $N_t \gg 1$ | ✗ | ✓ | ✓ | ✓ | $\zeta(1)$ | $\zeta(1)$ | ✗ | ≤ 5% | ✗ | ✓ |
| DER (Buzzega et al., 2020) | Online | ✗ | $N_t \gg 1$ | ✗ | ✓ | ✓ | ✓ | $\zeta(1)$ | $\zeta(1)$ | ✗ | ≤ 5% | ✓ | ✗ |
| DER++ (Buzzega et al., 2020) | Online | ✗ | $N_t \gg 1$ | ✗ | ✓ | ✓ | ✓ | $\zeta(1)$ | $\zeta(1)$ | ✗ | ≤ 5% | ✓ | ✓ |
| AGEM (Chaudhry et al., 2018b) | Online | ✗ | $N_t \gg 1$ | ✗ | ✓ | ✗ | ✓ | $\zeta(1)$ | $\zeta(1)$ | ✓ | - | ✓ | ✓ |
| GEM (Lopez-Paz & Ranzato, 2017) | Online | ✗ | $N_t \gg 1$ | ✗ | ✓ | ✗ | ✗ | $\zeta(k)$ | $\zeta(1)$ | ✓ | - | ✓ | ✓ |
| MIR (Aljundi et al., 2019) | Online | ✗ | $N_t \gg 1$ | ✗ | ✗ | ✓ | ✓ | $\zeta(2)$ | $\zeta(1)$ | ✓ | - | ✗ | ✓ |
| ExStream (Hayes et al., 2019a) | Streaming | ✗ | $N_t = 1$ | ✗ | ✓ | ✓ | ✗ | $\zeta(1)$ | $\zeta(1)$ | ✓ | ≤ 5% | ✗ | ✓ |
| REMIND (Hayes et al., 2019b) | Streaming | ✗ | $N_t = 1$ | ✗ | ✓ | ✓ | ✓ | $\zeta(1)$ | $\zeta(1)$ | ✗ | ≫ 10% | ✗ | ✓ |
| **Ours** | Streaming | ✓ | $N_t = 1$ | ✗ | ✓ | ✓ | ✓ | $\zeta(1)$ | $\zeta(1)$ | ✗ | ≤ 5% | ✓ | ✓ |
| **CISL Constraints** | | | $N_t = 1$ | ✗ | ✓ | ✓ | ✓ | $\zeta(1)$ | $\zeta(1)$ | | | | |

other existing incremental learning scenarios. In this paper, we are interested in *class-incremental learning* in a *streaming scenario* (CISL), where a learner requires to continually learn in a *'single-pass'* with no forgetting, given a single training example at every time step. That is, the learner is allowed to utilize the single new example only once (Hayes et al., 2019b;a). Class incremental learning (CIL) setting (Rebuffi et al., 2017; Chaudhry et al., 2018a; Belouadah et al., 2020) implies that the label space includes all the classes observed so far and no task id is required during inference, as opposed to task incremental learning methods like VCL/Coreset VCL (Nguyen et al., 2017). In addition, the model being learned should be able to predict efficiently at any time step (Gama et al., 2013) without any additional computation to improve the performance. It implies that the fine-tuning as needed in GDumb (Prabhu et al., 2020), Coreset VCL (Nguyen et al., 2017) is forbidden. Finally, for practical applicability, the learning strategy is expected to update the parameters quickly and leverage only a small memory buffer.

Training a deep neural network in *streaming setting* continuously with no forgetting (McCloskey & Cohen, 1989; French, 1999) is non-trivial due to the aforementioned requirements. Furthermore, adapting an existing online learning method is not straight forward due to various limitations poses by these methods which violates one or multiple conditions of the restrictive class-incremental streaming learning setup. For instance, AGEM (Chaudhry et al., 2018b) requires task id for prediction (i.e., task incremental learning setup). We provide empirical evidence that, when extended to CISL, the gradient computed in AGEM from a single example leads to suboptimal results (perhaps not surprisingly). MIR (Aljundi et al., 2019) infringes the single-pass learning constraint by employing a two-step learning process. It first performs a virtual gradient update to select the samples for memory replay and then performs another gradient update to finally update the model parameters. GDumb (Prabhu et al., 2020) proposes a greedy strategy to maintain past examples in a buffer and retrains on all examples during inference. Although it has been shown to work well on various

online learning setttings, GDumb does not update the model until inference (i.e., it does not accumulate any knowledge in the network) and requires fine-tuning before every inference, which violates (*i*) single-pass learning constraint and (*ii*) any time inference without further training constraint. Finally, the recently proposed streaming learning approach, REMIND (Hayes et al., 2019b) requires a large amount of cached data for good performance, which restricts its applicability. Table 1 illustrates which critical components/requirements of streaming learning are satisfied/infringed by the existing lifelong learning approaches.

**Contributions.** In this paper, we propose a novel method, *'**B**ayesian Class **I**ncremental **S**treaming Learning'* (BISLERi) to facilitate lifelong learning in streaming setting by addressing the aforementioned limitations. In particular, we enable streaming learning by leveraging a tiny episodic memory-buffer replay with dual regularization. It regularises the model from two sources. One focuses on regularizing the model parameters explicitly in a *streaming Bayesian framework* (Neal, 2012; Broderick et al., 2013), while the other regularizes, in the form of self-distillation (Mobahi et al., 2020), by enforcing the updated model to produce similar outputs for previous models on the past observed data. Our approach jointly trains the buffer replay and current task sample by incorporating the likelihood of the replay and current sample. As a result, unlike VCL (Nguyen et al., 2017), GDumb (Prabhu et al., 2020), we do not need explicit fine-tuning to improve the model's performance, and the model can be evaluated on the fly. We also propose novel online *loss-aware* buffer replacement and various replay strategies that helps to fraction buffer replay or replacement in an optimal way, which significantly improves the training and boost the model's performance.

Our experimental results on five benchmark datasets demonstrate the effectiveness of the proposed method to circumvent catastrophic forgetting in highly challenging streaming settings. The proposed approach significantly improves the recent state-of-the-art, and the extensive ablations validate the importance of the proposed components. In Figure 2, we compare the proposed method with the recent strong baselines. *Even though designed for streaming learning, it outperforms the baselines by a significant margin in all three different lifelong learning settings.* It implies that a method designed to work in the most restrictive setting can be thought of as a robust and flexible method for the various lifelong learning settings with the widest possible applicability.

**Our main contributions can be summarised as follows:**
(*i*) we propose a novel replay-based dual-regularization framework (BISLERi), comprising a *streaming Bayesian framework* (Neal, 2012; Broderick et al., 2013) as well as a functional regularizer, to overcome catastrophic forgetting in challenging *class-incremental streaming learning* scenario, (*ii*) we propose novel online *loss-aware* buffer replacement policies and include various sampling strategies which significantly boosts the model's performance, (*iii*) we empirically show that selecting a tiny subset of samples from memory and computing the joint likelihood with the current sample is highly efficient in terms of the model's accuracy, and *enough to avoid explicit finetuning*, and (*iv*) we experimentally show that our method significantly outperforms the recent strong baselines.

## 2 Problem Formulation

In this paper, we study class incremental streaming learning (CISL) in the deep neural networks.

**Streaming Learning (SL).** Streaming learning is the extreme case of online learning, where data arrives sequentially one datum at a time, and the model needs to adapt online in a single pass. That is, it is not allowed to visit any part of the data multiple times, and it can be evaluated at any point of time without waiting for the termination of the training. While in this setting, a learner can use a tiny replay buffer, it is strictly forbidden to use any additional computation, such as fine-tuning, to improve the performance at any time. Therefore, approaches like GDumb (Prabhu et al., 2020), Coreset VCL (Nguyen et al., 2017), are not allowed. Furthermore, it cannot be assumed that the input data-stream is independent and identically distributed (i.i.d); the data-stream can be temporally contiguous and/or there can be a class-based correlation in the data, for e.g.: (*i*) class-instance and (*ii*) instance ordering (for more details, refer to Section 5.1).

Let us consider an example dataset $\mathcal{D}$ divided into $T$ task sequences, i.e., $\mathcal{D} = \bigcup_{t=1}^{T} \mathcal{D}_t$. Each task $\mathcal{D}_t$ consists of $N_t$ labeled data-points, i.e., $\mathcal{D}_t = \left\{ d_t^{(j)} \right\}_{j=1}^{N_t} = \{(\boldsymbol{x}_j, y_j)\}_{j=1}^{N_t}$, and corresponds to different labeling task. $\boldsymbol{x}_j$ represents the input image & $y_j$ is the corresponding class-label, and $N_t$ is a variable across tasks.

In streaming learning, each task $\mathcal{D}_t$ consist only a single data-point, i.e., $|\mathcal{D}_t| = N_t = 1, \forall t$, and the model is required to adapt to this new example with no forgetting in a single pass, i.e., it cannot observe the new example multiple times. This setup is different from the widely popular incremental batch learning (IBL) (Kirkpatrick et al., 2017; Nguyen et al., 2017; Aljundi et al., 2017; 2018a;b). In IBL, it is assumesd that the model have access to the whole dataset, i.e., $\mathcal{D} = \bigcup_{t=1}^{T} \mathcal{D}_t$; the data does not come in online manner. It is further assumed that: $|\mathcal{D}_t| = N_t \gg 1, \forall t$. The model visits each task data $\mathcal{D}_t$ sequentially multiple times to mitigate catastrophic forgetting. Online learning methods (Prabhu et al., 2020), on the other hand, while assume that each task $\mathcal{D}_t$ comes sequentially one at a time, assumes that the each data subset size $|\mathcal{D}_t| = N_t \gg 1$. Furthermore, in online learning setup, it is allowed to fine-tune the network parameters any time with the samples stored in the memory. Specifically, it can fine-tune the network as many times as it wants, only constraint is that the samples are needed to be stored in the memory.

**Class Incremental Learning (CIL).** Class-incremental learning (CIL) (Rebuffi et al., 2017; Chaudhry et al., 2018a; Belouadah et al., 2020; Rios & Itti, 2018) is a challenging variant of continual learning. During inference, it considers the label space over all the classes that have been observed so far. This is in contrast to the task-incremental learning methods, such as VCL (Nguyen et al., 2017), which requires the task-id to be specified while doing inference.

In Table 1, we categorize various recently proposed strong baseline approaches depending on the underlying simplifying assumptions that they impose. It can be observed that TinyER (Chaudhry et al., 2019), REMIND (Hayes et al., 2019b) and BISLERi (Ours) do not violate any constraints of class-incremental streaming learning (CISL) setting. MIR (Aljundi et al., 2019) infringes the single-pass learning constraint by employing a two-step learning process. First, it performs a virtual gradient update to select the samples for memory replay and then performs another gradient update to finally update the model parameters. While MIR (Aljundi et al., 2019) uses two-pass learning, a modified version of MIR can be considered as a single-pass method. That is, first compute the gradient $g_{new}$ on $D_{new}$ and check the interference score after the gradient update (virtual gradient update) to get the dataset $D_{interferred}$. Then, compute the gradient of $D_{new} \cup D_{interferred}$. Since, we already have the gradient w.r.t $g_{new}$, all we need is to compute the gradient over $D_{interferred}$ to get the gradient for the final update. However, we have to remember that in streaming learning (SL), $|D_{new}| = 1$, i.e., in each incremental step, only a single training example arrives. Therefore, the gradient $g_{new}$ will be computed w.r.t a single data-point $D_{new}$, where $|D_{new}| = 1$. Since *SGD with just a single datapoint is extremely noisy and hard to optimize*, the gradient update (virtual gradient update) with $g_{new}$ will result in poor generalization. Henceforth, the selected maximally interfered dataset ($D_{interferred}$) will be sub-optimal, such that, it will suffer from catastrophic forgetting (McCloskey & Cohen, 1989; French, 1999; Goodfellow et al., 2015) in streaming lifelong learning setup. GDumb (Prabhu et al., 2020) violates streaming learning constraints by employing fine-tuning the network parameters before each inference. ExStream (Hayes et al., 2019a) despite being a streaming learning method, it replays all buffer samples instead of replaying only a few samples while training on the new example. Therefore, we consider that it violates the subset buffer replay constraint, which ultimately violates the CISL constraint. It can be observed, among the online learning methods, only TinyER (Chaudhry et al., 2019) can be adapted to the CISL setting without violating any crucial properties that are necessary for the CISL setting. Figure 2 compares the performance of the various baselines in different lifelong learning settings. We can observe that BISLERi (Ours) performs best compared to the baselines consistently throughout the different settings.

In Figure 4, we demonstrate the impact of the presence of temporal coherence in the input data-stream. It can be observed that ($i$) class-instance and ($ii$) instance ordering are more challenging compared to when the data-stream is organized randomly. It further can observed that while the strong baselines continue to suffer from severe forgetting in the presence of temporal coherence, BISLERi (Ours) suffer from minimal amount of forgetting.

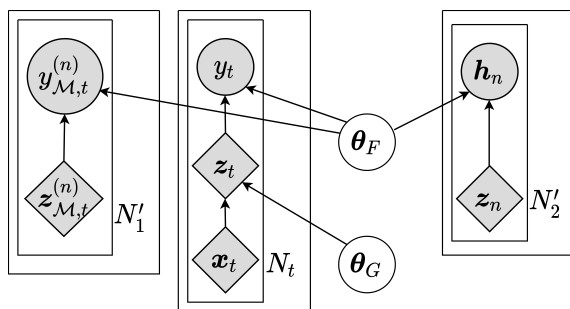

Figure 1: Schematic representation of the proposed model. $\boldsymbol{\theta}_G$ represents the parameters of the feature extractor $G(\cdot)$, whereas $\boldsymbol{\theta}_F$ represents the parameters of the plastic network / Bayesian neural network $F(\cdot)$.

## 3 Proposed Streaming Learning Framework

In the following, we introduce BISLERi, which trains a convolutional neural network (CNN) in streaming learning setup. Formally, it is assumed that we have access to a limited number of labeled training examples, which we use to train the model in a typical offline manner. We term this step as the base initialization step with $t = 0$. Then in each incremental step, i.e., $\forall t > 0$, the model observes *a single new training example*, i.e., $\mathcal{D}_t = \{d_t\} = \{(\boldsymbol{x}_t, y_t)\}$, and adapts to it by doing a single step posterior computation.

### 3.1 Streaming Learning With A Single Example

Formally, we separate the CNN into two neural networks (Figure 1): (*i*) non-plastic feature extractor $G(\cdot)$ consisting the first few layers of the CNN, and (*ii*) plastic neural network $F(\cdot)$ consisting the final layers of the CNN. For a given input image $\boldsymbol{x}$, the predicted class label is computed as: $y = F(G(\boldsymbol{x}))$. We initialize the parameters of the feature extractor $G(\cdot)$ and keep it frozen throughout streaming learning. We use a *Bayesian-neural-network* (BNN) (Neal, 2012; Jospin et al., 2020) as the plastic network $F(\cdot)$, and optimize its parameters with sequentially coming data in streaming setting. We discuss how the parameters of the feature extractor $G(\cdot)$ is initialized in Section 3.5. In the below, we describe how the plastic network $F(\cdot)$ is trained with a single step posterior computation in each incremental step with a single data point $\mathcal{D}_t = \{d_t\} = \{(\boldsymbol{x}_t, y_t)\}$ in streaming learning setup with no catastrophic forgetting (McCloskey & Cohen, 1989; French, 1999).

**Variational Posterior Estimation.** Streaming learning naturally emerges from the Bayes' rule (Broderick et al., 2013); given the posterior $p(\boldsymbol{\theta}|\mathcal{D}_{1:t-1})$, whenever a new data: $\mathcal{D}_t = \{d_t\} = \{(\boldsymbol{x}_t, y_t)\}$ arrives, the new posterior $p(\boldsymbol{\theta}|\mathcal{D}_{1:t})$ can be computed by combining the previous posterior and the new data likelihood, i.e., $p(\boldsymbol{\theta}|\mathcal{D}_{1:t}) \propto p(\mathcal{D}_t|\boldsymbol{\theta})\ p(\boldsymbol{\theta}|\mathcal{D}_{1:t-1})$, where the old posterior is treated as the prior. However, for any complex model, the exact Bayesian inference is not tractable, and an approximation is needed. A Bayesian neural network (Neal, 2012) commonly approximates the posterior with a variational posterior $q(\boldsymbol{\theta})$ by minimizing the following KL divergence (Eq. 1), or equivalently by maximizating the evidence lower bound (ELBO) (Blundell et al., 2015) (Eq. 2):

$$\mathcal{L}_t(\boldsymbol{\theta}) = \arg\min_{q \in \mathcal{Q}} \mathrm{KL}\left(q_t(\boldsymbol{\theta}) \,\|\, \frac{1}{Z_t} q_{t-1}(\boldsymbol{\theta}) p(\mathcal{D}_t|\boldsymbol{\theta})\right) \tag{1}$$

$$\simeq \arg\max_{q \in \mathcal{Q}} \mathbb{E}_{\boldsymbol{\theta} \sim q_t(\boldsymbol{\theta})} \left[\log p(\mathcal{D}_t|\boldsymbol{\theta})\right] - \mathrm{KL}\left(q_t(\boldsymbol{\theta}) \,\|\, q_{t-1}(\boldsymbol{\theta})\right) \tag{2}$$

It is worth mentioning that the KL-divergence in Eq. 2 works as a inherent regularizer, which can prevent forgetting in the network by keeping the prior and the posterior distribution close to each other. However, there are few concerns. Firstly, optimizing the ELBO (in Eq. 2) to approximate the posterior $p(\boldsymbol{\theta}|\mathcal{D}_{1:t})$ with only a single training example, i.e., $\forall t > 0, \mathcal{D}_t = \{d_t\} = \{(\boldsymbol{x}_t, y_t)\}$, in each incremental step can fail in streaming setting (Ghosh et al., 2018). Furthermore, likelihood estimation from a single example:

$\mathcal{D}_t = \{d_t\} = \{(\boldsymbol{x}_t, y_t)\}$ can bias the model towards the new data disproportionately and can maximize the confusion during inference in class incremental learning setup (Chaudhry et al., 2018a).

These concerns cannot be prevented only with the parameters regularization. We, therefore, propose to estimate the likelihood from both the new example: $\mathcal{D}_t$ and the previously observed examples in order to compute the new posterior: $p(\boldsymbol{\theta}|\mathcal{D}_{1:t})$. For this purpose, a small fraction ($\leq 5\%$) of past observed examples (for buffer capacity refer Table 2) are stored as representatives in a *fixed-sized* tiny episodic memory buffer $\mathcal{M}$. Instead of storing raw pixels (images) $\boldsymbol{x}$, we store the embedding $\boldsymbol{z} = G(\boldsymbol{x})$, where $\boldsymbol{z} \in \mathbb{R}^d$. This allows us to avoid the cost of recomputing the image-embeddings and expedites the learning even further. It also saves a significant amount of space and allows us to keep more examples in a small budget.

During training on the new example: $\mathcal{D}_t = \{d_t\} = \{(\boldsymbol{x}_t, y_t)\}$, we select a subset of samples: $\mathcal{D}_{\mathcal{M},t}$ from the memory $\mathcal{M}$, where: $(i)\mathcal{D}_{\mathcal{M},t} \subset \mathcal{M}$, $(ii)|\mathcal{D}_{\mathcal{M},t}| = N_1' \ll |\mathcal{M}|$, *instead of replaying the whole buffer*, and compute the likelihood jointly with the new example $\mathcal{D}_t$ to estimate the new posterior. We, therefore, compute the new posterior as follows: $p(\boldsymbol{\theta}|\mathcal{D}_{1:t}) \propto p(\mathcal{D}_t|\boldsymbol{\theta})\, p(\mathcal{D}_{\mathcal{M},t}|\boldsymbol{\theta})\, p(\boldsymbol{\theta}|\mathcal{D}_{1:t-1})$. Since the exact Bayesian inference is intractable, we approximate it with a variational posterior $q_t(\boldsymbol{\theta})$ as follows:

$$\mathcal{L}_t^1(\boldsymbol{\theta}) = \underset{q \epsilon \mathcal{Q}}{\arg\min} \, \mathrm{KL}\left(q_t(\boldsymbol{\theta}) \, || \, \frac{1}{Z_t} q_{t-1}(\boldsymbol{\theta}) p(\mathcal{D}_t|\boldsymbol{\theta}) p(\mathcal{D}_{\mathcal{M},t}|\boldsymbol{\theta})\right) \quad (3)$$

Note that Eq. 3 is significantly different from VCL (Nguyen et al., 2017), where they assume *task-incremental learning* setting with *separate head networks*, and incorporates the coreset samples only for explicit finetuning before inference. For more details on VCL/Coreset VCL (Nguyen et al., 2017) refer to the appendix.

The above minimization (in Eq. 3) can be equivalently written as the maximization of the evidence lower bound (ELBO) as follow:

$$\mathcal{L}_t^1(\boldsymbol{\theta}) = \mathbb{E}_{\boldsymbol{\theta}\sim q_t(\boldsymbol{\theta})}\left[\log p(y_t|\boldsymbol{\theta}, G(\boldsymbol{x}_t))\right] + \sum_{n=1}^{N_1'} \mathbb{E}_{\boldsymbol{\theta}\sim q_t(\boldsymbol{\theta})}\left[\log p(y_{\mathcal{M},t}^{(n)}|\boldsymbol{\theta}, \boldsymbol{z}_{\mathcal{M},t}^{(n)})\right] - \lambda_1 \cdot \mathrm{KL}\left(q_t(\boldsymbol{\theta}) \, || \, q_{t-1}(\boldsymbol{\theta})\right) \quad (4)$$

where: $(i)$ $\mathcal{D}_t = \{d_t\} = \{(\boldsymbol{x}_t, y_t)\}$, $(ii)$ $\mathcal{D}_{\mathcal{M},t} = \{d_{\mathcal{M},t}^{(n)}\}_{n=1}^{N_1'} = \{(\boldsymbol{z}_{\mathcal{M},t}^{(n)}, y_{\mathcal{M},t}^{(n)})\}_{n=1}^{N_1'}$, $(iii)$ $\mathcal{D}_{\mathcal{M},t} \subset \mathcal{M}$, $(iv)$ $|\mathcal{D}_{\mathcal{M},t}| = N_1' \ll |\mathcal{M}|$, and $(v)$ $\lambda_1$ is a hyper-parameter.

**Snap-Shot Self Distillation.** It is worth noting that the KL divergence minimization (in Eq. 4) between the prior and the posterior distribution works as a inherent regularizer, which tries to keep the changes in the network parameters minimal during streaming learning. However, its effect may weaken over the time due to the presence of distribution shift, temporal coherence in the input data-stream. Furthermore, the initialization of the prior with the old posterior at each incremental step can introduce information loss in the network for a longer sequence of streaming learning. On these grounds, we propose a functional regularizer, which encourages the network to mimic the output responses as produced in the past for the previously observed samples (for significance of self-distillation, see Section 6). Specifically, we propose to minimize the KL divergence between the class-probability scores obtained in the past and current time $t$:

$$\sum_{j=1}^{t-1} \mathbb{E}_{\boldsymbol{\theta}\sim q_t(\boldsymbol{\theta})}\left[\mathrm{KL}\left(\mathrm{softmax}(\boldsymbol{h}_j) \, || \, F_{\boldsymbol{\theta}}(G(\boldsymbol{x}_j))\right)\right] \quad (5)$$

where: $\boldsymbol{x}_j$ and $\boldsymbol{h}_j$ represents input examples and the logits obtained while training on $\mathcal{D}_j$ respectively.

The above objective (in Eq. 5) resembles the teacher-student training approach (Hinton et al., 2015); however, since in this case only a single network is used as both teacher & student network, it is called self knowledge distillation or self distillation (Hinton et al., 2015; Mobahi et al., 2020). It (the objective in Eq. 5), however, requires the availability of the embeddings and the corresponding logits for all the past observed data till time instance $(t-1)$. Since storing all the past examples is not feasible, we only store the logits for all samples in memory $\mathcal{M}$. During training, we uniformly select $N_2'$ samples along with their logits and optimize the

following objective:

$$\mathcal{L}_t^2(\boldsymbol{\theta}) = \sum_{j=1}^{N_2'} \mathbb{E}_{\boldsymbol{\theta} \sim q_t(\boldsymbol{\theta})} \left[ \mathrm{KL}\left( \mathrm{softmax}(\boldsymbol{h}_j) \ || \ F_{\boldsymbol{\theta}}(\boldsymbol{z}_j) \right) \right] \tag{6}$$

where: ($i$) $\boldsymbol{z}_j$ and $\boldsymbol{h}_j$ represents the feature-map and the corresponding logit, and ($ii$) $N_2' \ll |\mathcal{M}|$.

Under the mild assumptions of knowledge distillation (Hinton et al., 2015), the optimization in Eq. 6 is equivalent to minimization of the Euclidean distance between the corresponding logits. In this work, we, therefore, minimize the following objective instead of the objective above (Eq. 6):

$$\mathcal{L}_t^2(\boldsymbol{\theta}) = \lambda_2 \cdot \sum_{j=1}^{N_2'} \mathbb{E}_{\boldsymbol{\theta} \sim q_t(\boldsymbol{\theta})} \left[ ||\boldsymbol{h}_j - f_{\boldsymbol{\theta}}(\boldsymbol{z}_j)||_2^2 \right] \tag{7}$$

where: ($i$) $f(\cdot)$ represents the plastic network $F(\cdot)$ without the softmax activation, and ($ii$) $\lambda_2$ is a hyper-parameter.

It is, however, worth mentioning that the stored logits used in Eq.(7) are updated in an online manner. That is, whenever a sample is selected for memory replay, we replace the corresponding old logits with the newly predicted logits. Therefore, it is called snap-shot self-distillation (Yang et al., 2019), where the model is constrained to match the logits obtained during earlier memory replay, i.e., the last snap-shot. It essentially prevents the model from being constrained to match the sub-optimal initial logits and minimizes the information loss in the network.

**Training.** Training the plastic network (BNN) $F(\cdot)$ requires specification of $q(\boldsymbol{\theta})$ and, in this work, we model $\boldsymbol{\theta}$ by stacking up the parameters (weights & biases) of the network $F(\cdot)$. We use a Gaussian mean-field posterior $q_t(\boldsymbol{\theta})$ for the network parameters, and choose the prior distribution, i.e., $q_0(\boldsymbol{\theta}) = p(\boldsymbol{\theta})$, as multivariate Gaussian distribution. We train the network $F(\cdot)$ by maximizing the ELBO in Eq. 4 and minimizing the Euclidean distance in Eq. 7. For memory replay in Eq. 4, we select past informative samples using the strategies mentioned in Section 3.2, and we use *uniform sampling* to select samples from memory to be used in Eq. 7. Figure 1 shows the schematic diagram of the proposed model as well as the learning process.

### 3.2 Informative Past Sample Selection For Replay

We consider the following strategies for selecting past informative samples for memory replay:

**Uniform Sampling (Uni).** In this approach, samples are selected uniformly random from memory. If we have the buffer of size $K$ then each samples are selected with probability $1/K$.

**Uncertainty-Aware Positive-Negative Sampling (UAPN).** UAPN selects $N_1'/2$ samples with the highest uncertainty scores (negative samples) and $N_1'/2$ samples with the lowest uncertainty scores (positive samples). Empirically, we observe that this sample selection strategy results in the best performance. We measure the predictive uncertainty (Chai, 2018) for an input $\boldsymbol{z}$ with BNN $F(\cdot)$ as follows:

$$\Phi(\boldsymbol{z}) \approx - \sum_c \left( \frac{\sum_k p(\hat{y} = c | \boldsymbol{z}, \boldsymbol{\theta}^k)}{k} \right) \log \left( \frac{\sum_k p(\hat{y} = c | \boldsymbol{z}, \boldsymbol{\theta}^k)}{k} \right) \tag{8}$$

where: $p(\hat{y} = c | \boldsymbol{z}, \boldsymbol{\theta}^k)$ is the predicted softmax output for class $c$ using the $k$-th sample of weights $\boldsymbol{\theta}^k$ from $q(\boldsymbol{\theta})$. We use $k = 5$ samples for uncertainty estimation.

**Loss-Aware Positive-Negative Sampling (LAPN).** LAPN selects $N_1'/2$ samples with the highest loss-values (negative-samples), and $N_1'/2$ samples with the lowest loss-values (positive-samples). Empirically we observe that the combination of most and least certain samples shows a significant performance boost since one ensures quality while the other ensures diversity for the memory replay.

Table 2: Memory buffer capacity used for various datasets.

| Dataset | CIFAR10 | CIFAR100 | ImageNet100 | iCubWorld 1.0 | CORe50 |
|---|---|---|---|---|---|
| **Buffer Capacity** | 1000 | 1000 | 1000 | 180 | 1000 |
| **Training-Set Size** | 50000 | 50000 | 127778 | 6002 | 119894 |

### 3.3 Memory Buffer Replacement Policy

In a practical lifelong learning scenario (Aljundi et al., 2018b;a), data can come indefinitely throughout the time. It imples that the episodic replay buffer will require to have an infinite capacity to store all the observed examples. Otherwise, its capacity will be quickly exhausted and the new instances cannot be accommodated in the replay buffer; it will then require a buffer replacement policy, which will replace a stored example in the memory to accommodate a new one whenever the buffer is full.

In this work, we use a *'fixed-sized'* tiny episodic replay buffer to store a fraction ($\leq 5\%$) of all the previously observed data. In the below, we discuss two buffer replacement policies, which replaces a previously stored example if the buffer is full. Otherwise, the new instance is simply stored. For episodic memory capacity across various dataset, refer to Table 2.

**Loss-Aware Weighted Class Balancing Replacement (LAWCBR).** In this approach, whenever a new sample comes and the buffer is full, we remove a sample from the class with maximum number of samples present in the buffer, i.e., $y_r = \arg\max \text{ClassCount}(\mathcal{M})$. However, instead of removing an example uniformly, we weigh each sample of the majority class inversely w.r.t their loss, i.e., $w_i^{y_r} \propto \frac{1}{l_i^{y_r}}$ and use these weights as the replacement probability; the lesser the loss, the more likely to be removed.

**Loss-Aware Weighted Random Replacement With A Reservoir (LAWRRR).** In this approach, we propose a novel variant of reservoir sampling (Vitter, 1985) to replace an existing sample with the new sample when the buffer is full. We weigh each stored sample inversely w.r.t the loss, i.e., $w_i \propto \frac{1}{l_i}$, and proportionally to the total number of examples of that class in which the sample belongs present in the buffer, i.e., $w_i \propto \text{ClassCount}(\mathcal{M}, y_i)$. Whenever a new example satisfies the replacement condition of reservoir sampling, we combine these two scores and use that as the replacement probability; the higher the weight, the more likely to be replaced.

### 3.4 Efficient Buffer Update

Loss-aware and uncertainty-aware sampling strategies require computing these quantities at every time step during streaming learning with all the samples stored in memory. However, this becomes computationally expensive with the larger replay buffer size. To overcome such limitation, we store the corresponding loss-values and uncertainty-scores along with the stored examples in the replay buffer. Since these quantities are scalar values, the additional storage requirement is negligible but saves the time to compute the loss and uncertainty in each incremental step. Every time a sample is selected for memory replay, its loss and uncertainty are replaced by the new values. We, in addition, replace the stored logits with the new logits. Empirically, we observe that the model's accuracy degrades if we don't update the logits with the new logits.

### 3.5 Feature Extractor

In this work, we separate the representation learning, i.e., learning the feature extractor $G(\cdot)$, and the classifier learning, i.e., learning the plastic network $F(\cdot)$. Similar to several existing continual learning approaches (Kemker & Kanan, 2017; Hayes et al., 2019a; Xiang et al., 2019; Hayes et al., 2019b), we initialize the feature extractor $G(\cdot)$ with the weights learned through supervised visual representation learning (Krizhevsky et al., 2012) task, and keep them fixed throughout streaming learning. The motivation to use a pre-trained feature extractor is that the features learned by the first few layers of the neural networks are highly transferable and not specific to any particular task or dataset and can be applied to several different task(s) or dataset(s) (Yosinski et al., 2014). Furthermore, it is hard, if not infeasible, to learn generalized visual fea-

tures, that can be used across all the classes with having access to only a single example at every time (Zhu et al., 2021).

In our experiments, for all the baselines along with BISLERi, we use Mobilenet-V2 (Sandler et al., 2018) pre-trained on ImageNet-1000 (ILSVRC-2012) (Russakovsky et al., 2015) as the base architecture for the visual feature extractor. It consists of a convolutional base and a classifier network. We remove the classifier network and use the convolutional base as the feature extractor $G(\cdot)$ to obtain embedding, which is fed to the plastic network BNN $F(\cdot)$. For details on the plastic network used for other baselines, refer to Section 5.5.

## 4    Related Work

Existing continual learning approaches can be broadly classified into (Parisi et al., 2019; Delange et al., 2021): (*i*) parameter isolation based approaches, (*ii*) regularization based approaches, and (*iii*) rehearsal based approaches.

Parameter-isolation-based approaches train different subsets of model parameters on sequential tasks. PNN (Rusu et al., 2016), DEN (Yoon et al., 2017) expand the network to accommodate the new task. PathNet (Fernando et al., 2017), PackNet (Mallya & Lazebnik, 2018), Piggyback (Mallya et al., 2018), and HAT (Serra et al., 2018) train different subsets of network parameters on each task.

Regularization-based approaches use an extra regularization term in the loss function to enable continual learning. LWF (Li & Hoiem, 2017) uses knowledge distillation (Hinton et al., 2015) loss to prevent catastrophic forgetting. EWC (Kirkpatrick et al., 2017), IMM (Lee et al., 2017), SI (Zenke et al., 2017) and MAS (Aljundi et al., 2018a) regularize by penalizing changes to the important weights of the network. FRCL (Titsias et al., 2019) employs a functional regularizer based on Bayesian inference over the function space rather than the parameters of the deep neural networks to enable CL. It avoids forgetting a previous task by constructing and memorizing an approximate posterior belief over the underlying task-specific function. UCB (Ebrahimi et al., 2019) based on Bayesian neural networks enables continual learning by controlling the learning rate of each parameter as a function of uncertainty. BGD (Zeno et al., 2018) uses closed-form variational Bayes to mitigate catastrophic forgetting in task agnostic scenarios. That is, (*i*) in contrast to the methods like EWC (Kirkpatrick et al., 2017), MAS (Aljundi et al., 2018a), VCL (Nguyen et al., 2017) which are based on some core action taken on task-switch, BGD (Zeno et al., 2018) does not require any information on task identity, and (*ii*) BGD updates the posterior over the weights in closed-form, unlike VCL/Coreset VCL (Nguyen et al., 2017), Coreset Only (Farquhar & Gal, 2018), BISLERi (Ours) which relies on BBB (Blundell et al., 2015) to update the posterior.

Rehearsal-based approaches replay a subset of past training data during sequential learning. iCaRL (Rebuffi et al., 2017), SER (Isele & Cosgun, 2018), and TinyER (Chaudhry et al., 2019) use memory replay when training on a new task. DER/DER++ (Buzzega et al., 2020) uses knowledge distillation and memory replay while learning a new task. DGR (Shin et al., 2017), MeRGAN (Wu et al., 2018), and CloGAN (Rios & Itti, 2018) retain the past task(s) distribution with a generative model and replay the synthetic samples during incremental learning. Our approach also leverages memory replay from a tiny episodic memory; however, we store the feature maps instead of raw inputs.

Variational Continual Learning (VCL) (Nguyen et al., 2017) leverages Bayesian inference to mitigate catastrophic forgetting. However, the approach, when naïvely adapted, performs poorly in the *streaming learning setting*. Additionally, it also needs task-id during inference. Furthermore, the explicit finetuning with the buffer samples (coreset) before inference violates the *single-pass* learning constraint. Moreover, it still does not outperform our approach even with the finetuning. More details are given in the appendix.

REMIND (Hayes et al., 2019b) is a recently proposed rehearsal-based lifelong learning approach, which combats catastrophic forgetting in *streaming setting*. While it follows a setting close to the one proposed, the model stores a large number of past examples compared to the other baselines; for example, iCaRL (Rebuffi et al., 2017) stores 10K past examples for the ImageNet experiment, whereas REMIND stores 1M past examples. Further, it actually uses a lossy compression to store past samples, which is merely an engineering technique, not an algorithmic improvement, and can be used by any continual learning approach. For more details, please refer to the appendix.

## 5 Experiments

### 5.1 Datasets And Data Orderings

**Datasets.** To evaluate the efficacy of the proposed model we perform extensive experiments on five standard datasets: CIFAR10 (Krizhevsky et al., 2009), CIFAR100 (Krizhevsky et al., 2009), ImageNet100, iCubWorld 1.0 (Fanello et al., 2013), and CORe50 (Lomonaco & Maltoni, 2017). CIFAR10 and CIFAR100 are standard classification datasets with 10 and 100 classes, respectively. ImageNet100 is a subset of ImageNet-1000 (ILSVRC-2012) (Russakovsky et al., 2015) containing randomly chosen 100 classes, with each class containing 700-1300 training samples and 50 validation samples. Since, the test data for ImageNet-1000 (ILSVRC-2012) (Russakovsky et al., 2015) is not provided with labels, we use the validation data for evaluating the model's performance, similar to (Hayes et al., 2019b). iCubWorld 1.0 is an object recognition dataset which contains the sequences of video frames, with each containing a single object. There are 10 classes, with each containing 3 different object instances with 200-201 images each. Overall, each class contains 600-602 samples for training and 200-201 samples for testing. CORe50 is similar to iCubWorld 1.0, containing images from temporally coherent sessions, with the whole dataset divided into 11 distinct sessions characterized by different backgrounds and lighting. There are 10 classes, with each containing 5 different object instances with 2393-2400 images each. Overall, each class contains 11983-12000 samples for training and 4495-4500 samples for testing. *Technically, iCubWorld and CORe50 are the ideal dataset for streaming learning, as it requires learning from temporally coherent image sequences, which are naturally non-i.i.d images.*

**Evaluation Over Different Data Orderings.** The proposed approach is robust to the various *streaming learning* setting; we evaluate the model's streaming learning ability with the following four (Hayes et al., 2019a;b) challenging data ordering schemes: (*i*) *'streaming iid'*: where the data-stream is organized by the randomly shuffled samples from the dataset, (*ii*) *'streaming class iid'*: where the data-stream is organized by the samples from one or more classes, these samples are shuffled randomly, (*iii*) *'streaming instance'*: where the data-stream is organized by temporally ordered samples from different object instances, and (*iv*) *'streaming class instance'*: where the data-stream is organized by the samples from different classes, the samples within a class are temporally ordered based on different object instances. Only iCubWorld 1.0 and CORe50 dataset contains the temporal ordering, therefore *'streaming instance'*, and *'streaming class instance'* setting are evaluated only on these two datasets. Please refer to the appendix for more details.

### 5.2 Metrics

For evaluating the performance of the streaming learner, we use $\boldsymbol{\Omega}_{\text{all}}$ metric, similar to (Kemker et al., 2018; Hayes et al., 2019a;b), where $\boldsymbol{\Omega}_{\text{all}}$ represents normalized incremental learning performance with respect to an offline learner:

$$\boldsymbol{\Omega}_{\text{all}} = \frac{1}{T} \sum_{t=1}^{T} \frac{\boldsymbol{\alpha}_t}{\boldsymbol{\alpha}_{\text{offline},t}} \tag{9}$$

where $T$ is the total number of testing events, $\boldsymbol{\alpha}_t$ is the performance of the incremental learner at time $t$, and $\boldsymbol{\alpha}_{\text{offline},t}$ is the performance of a traditional offline model at time $t$.

### 5.3 Baselines And Compared Methods

The proposed approach follows the *'streaming learning setup'*; to the best of our knowledge, recent works ExStream (Hayes et al., 2019a) and REMIND (Hayes et al., 2019b) are the only methods that follow the same setting. We compare our approach against these strong baselines. We also compare our model with (*i*) a network trained with one sample at a time (Fine-tuning/lower-bound) and (*ii*) a network trained offline, assuming all the data is available (Offline/upper-bound). Finally, we choose recent popular 'batch' (IBL) and 'online' learning methods, such as EWC (Kirkpatrick et al., 2017), MAS (Aljundi et al., 2018a), VCL (Nguyen et al., 2017), Coreset VCL (Nguyen et al., 2017), Coreset Only (Farquhar & Gal, 2018), TinyER (Chaudhry et al., 2019), GDumb (Prabhu et al., 2020), AGEM (Chaudhry et al., 2018b) and DER/DER++ (Buzzega et al., 2020) as baselines and rigorously evaluate our model against these approaches. For a fair comparison, we train all the methods in a streaming learning setup, i.e., one sample at a time. 'Coreset VCL' and 'Coreset

379  Only' both are trained in a streaming manner; however, the network is fine-tuned with the stored samples
380  before inference. Furthermore, GDumb stores samples in memory and fine-tunes the network with them
381  before inference, while fine-tuning is prohibited in *'streaming learning'*. Therefore, Coreset VCL, Coreset
382  Only, and GDumb have an extra advantage compared to the true *'streaming learning'* approaches. Still,
383  BISLERi outperforms these approaches by a significant margin. We provide more details about the baseline
384  methods in the appendix.

Table 3: $\mathbf{\Omega}_{\text{all}}$ results with their associated standard deviations. For each experiment, the method with best performance in *'streaming-learning-setup'* is highlighted in **Bold**. The reported results are average over 10 runs with different permutations of the data. Offline model is trained only once. $\widehat{\text{Offline}} = \frac{1}{T}\sum_{t=1}^{T} \boldsymbol{\alpha}_{\text{offline},t}$, where $T$ is the total number of testing events. '-' indicates experiments we are unable to run, because of compatibility issues.

Note: Methods in Red use fine-tuning, implying that these methods violate streaming learning (SL) constraints and have an extra advantage over true streaming learning (SL) methods, such as 'Ours'.

| Method | iid | | | Class-iid | | |
|---|---|---|---|---|---|---|
| | CIFAR10 | CIFAR100 | ImageNet100 | CIFAR10 | CIFAR100 | ImageNet100 |
| Fine-Tune | $0.1175 \pm 0.0000$ | $0.0180 \pm 0.0035$ | $0.0127 \pm 0.0029$ | $0.3447 \pm 0.0003$ | $0.1277 \pm 0.0022$ | $0.1223 \pm 0.0052$ |
| EWC | - | - | - | $0.3446 \pm 0.0003$ | $0.1292 \pm 0.0037$ | $0.1225 \pm 0.0039$ |
| MAS | - | - | - | $0.3470 \pm 0.0075$ | $0.1280 \pm 0.0029$ | $0.1234 \pm 0.0046$ |
| VCL | - | - | - | $0.3442 \pm 0.0006$ | $0.1273 \pm 0.0041$ | $0.1205 \pm 0.0015$ |
| Coreset VCL | - | - | - | $0.3716 \pm 0.0501$ | $0.1414 \pm 0.0224$ | $0.1259 \pm 0.0122$ |
| Coreset Only | - | - | - | $0.3684 \pm 0.0442$ | $0.1432 \pm 0.0256$ | $0.1273 \pm 0.0182$ |
| GDumb | $0.8686 \pm 0.0065$ | $0.6067 \pm 0.0119$ | $0.8361 \pm 0.0070$ | $\mathit{0.9252 \pm 0.0057}$ | $0.7635 \pm 0.0096$ | $\mathit{0.9197 \pm 0.0081}$ |
| AGEM | $0.1175 \pm 0.0000$ | $0.0182 \pm 0.0035$ | $0.0139 \pm 0.0041$ | $0.3448 \pm 0.0002$ | $0.1290 \pm 0.0037$ | $0.1215 \pm 0.0025$ |
| DER | $0.1175 \pm 0.0000$ | $0.0165 \pm 0.0003$ | $0.0126 \pm 0.0027$ | $0.3449 \pm 0.0011$ | $0.1278 \pm 0.0024$ | $0.1217 \pm 0.0038$ |
| DER++ | $0.1175 \pm 0.0000$ | $0.0173 \pm 0.0028$ | $0.0130 \pm 0.0039$ | $0.3588 \pm 0.0423$ | $0.1290 \pm 0.0054$ | $0.1230 \pm 0.0068$ |
| TinyER | $0.9314 \pm 0.0114$ | $0.7588 \pm 0.0128$ | $0.9415 \pm 0.0085$ | $0.8926 \pm 0.0158$ | $0.7402 \pm 0.0195$ | $0.8995 \pm 0.0122$ |
| ExStream | $0.8866 \pm 0.0244$ | $0.7845 \pm 0.0121$ | $0.9293 \pm 0.0082$ | $0.8123 \pm 0.0209$ | $0.7176 \pm 0.0208$ | $0.8757 \pm 0.0148$ |
| REMIND | $0.8910 \pm 0.0073$ | $0.6457 \pm 0.0091$ | $0.9088 \pm 0.0109$ | $0.8832 \pm 0.0201$ | $0.6787 \pm 0.0215$ | $0.8803 \pm 0.0157$ |
| **Ours** | $\mathbf{0.9579 \pm 0.0040}$ | $\mathbf{0.8679 \pm 0.0057}$ | $\mathbf{0.9640 \pm 0.0060}$ | $\mathbf{0.8991 \pm 0.0089}$ | $\mathbf{0.7724 \pm 0.0188}$ | $\mathbf{0.9171 \pm 0.0073}$ |
| Offline | 1.0000 | 1.0000 | 1.0000 | 1.0000 | 1.0000 | 1.0000 |
| $\widehat{\text{Offline}}$ | 0.8509 | 0.6083 | 0.8520 | 0.8972 | 0.7154 | 0.8953 |

| Method | iid | | Class-iid | | instance | | Class-instance | |
|---|---|---|---|---|---|---|---|---|
| | iCubWorld 1.0 | CORe50 | iCubWorld 1.0 | CORe50 | iCubWorld 1.0 | CORe50 | iCubWorld 1.0 | CORe50 |
| Fine-Tune | $0.1369 \pm 0.0184$ | $0.1145 \pm 0.0000$ | $0.3893 \pm 0.0534$ | $0.3485 \pm 0.0171$ | $0.1307 \pm 0.0000$ | $0.1145 \pm 0.0000$ | $0.3485 \pm 0.0022$ | $0.3430 \pm 0.0003$ |
| EWC | - | - | $0.3790 \pm 0.0419$ | $0.3508 \pm 0.0243$ | - | - | $0.3487 \pm 0.0034$ | $0.3427 \pm 0.0007$ |
| MAS | - | - | $0.3912 \pm 0.0613$ | $0.3432 \pm 0.0004$ | - | - | $0.3486 \pm 0.0019$ | $0.3429 \pm 0.0005$ |
| VCL | - | - | $0.3806 \pm 0.0527$ | $0.3462 \pm 0.0129$ | - | - | $0.3473 \pm 0.0025$ | $0.3420 \pm 0.0009$ |
| Coreset VCL | - | - | $0.3948 \pm 0.0558$ | $0.3424 \pm 0.0019$ | - | - | $0.4705 \pm 0.0165$ | $0.4715 \pm 0.0054$ |
| Coreset Only | - | - | $0.3994 \pm 0.0922$ | $0.3688 \pm 0.0499$ | - | - | $0.4669 \pm 0.0251$ | $0.4748 \pm 0.0035$ |
| GDumb | $0.8993 \pm 0.0413$ | $0.9345 \pm 0.0121$ | $\mathit{0.9660 \pm 0.0201}$ | $\mathit{0.9742 \pm 0.0081}$ | $0.6715 \pm 0.0540$ | $0.7433 \pm 0.0246$ | $0.7908 \pm 0.0329$ | $0.6548 \pm 0.0259$ |
| AGEM | $0.1311 \pm 0.0000$ | $0.1145 \pm 0.0000$ | $0.4047 \pm 0.0632$ | $0.3460 \pm 0.0101$ | $0.1309 \pm 0.0003$ | $0.1145 \pm 0.0000$ | $0.3489 \pm 0.0030$ | $0.3429 \pm 0.0004$ |
| DER | $0.1437 \pm 0.0393$ | $0.1145 \pm 0.0000$ | $0.4057 \pm 0.1046$ | $0.3432 \pm 0.0005$ | $0.3759 \pm 0.2404$ | $0.1168 \pm 0.0072$ | $0.4082 \pm 0.1662$ | $0.3308 \pm 0.0385$ |
| DER++ | $0.1428 \pm 0.0364$ | $0.1145 \pm 0.0000$ | $0.4467 \pm 0.1287$ | $0.3431 \pm 0.0003$ | $0.4518 \pm 0.2510$ | $0.1145 \pm 0.0000$ | $0.4499 \pm 0.2311$ | $0.3429 \pm 0.0006$ |
| TinyER | $0.9590 \pm 0.0378$ | $1.0007 \pm 0.0121$ | $0.9069 \pm 0.0297$ | $0.9573 \pm 0.0125$ | $0.8726 \pm 0.0649$ | $0.8432 \pm 0.0262$ | $0.8215 \pm 0.0341$ | $0.8461 \pm 0.0247$ |
| ExStream | $0.9235 \pm 0.0584$ | $0.9844 \pm 0.0156$ | $0.8820 \pm 0.0285$ | $0.8760 \pm 0.0166$ | $0.8954 \pm 0.0542$ | $0.8257 \pm 0.0295$ | $0.8727 \pm 0.0229$ | $0.8837 \pm 0.0211$ |
| REMIND | $0.9260 \pm 0.0311$ | $0.9933 \pm 0.0115$ | $0.8553 \pm 0.0349$ | $0.9448 \pm 0.0125$ | $0.8157 \pm 0.0600$ | $0.8544 \pm 0.0247$ | $0.7615 \pm 0.0319$ | $0.7826 \pm 0.0377$ |
| **Ours** | $\mathbf{0.9716 \pm 0.0141}$ | $\mathbf{1.0069 \pm 0.0058}$ | $\mathbf{0.9480 \pm 0.0215}$ | $\mathbf{0.9686 \pm 0.0122}$ | $\mathbf{0.9580 \pm 0.0298}$ | $\mathbf{0.9824 \pm 0.0090}$ | $\mathbf{0.9585 \pm 0.0223}$ | $\mathbf{0.9384 \pm 0.0130}$ |
| Offline | 1.0000 | 1.0000 | 1.0000 | 1.0000 | 1.0000 | 1.0000 | 1.0000 | 1.0000 |
| $\widehat{\text{Offline}}$ | 0.7626 | 0.8733 | 0.8849 | 0.9070 | 0.7646 | 0.8733 | 0.8840 | 0.9079 |

## 5.4 Results

386  The detailed results of BISLERi over various experimental settings along with the strong baseline methods
387  are shown in Table 3. We can clearly observe that BISLERi consistently outperforms all the baseline
388  by a significant margin. The proposed model is also robust to the different streaming learning scenarios
389  compared to the baselines. We repeat our experiment ten times, and report the average-accuracy along
390  with the standard-deviations. We observe that 'batch-learning' methods severely suffer from catastrophic
391  forgetting. Moreover, replay-based 'online-learning' method such as AGEM also suffer from information loss
392  badly.

393  Although GDumb achieves higher accuracy on several datasets on class-i.i.d ordering, it fine-tunes the
394  network parameters before each inference, ultimately violating the constraints of streaming learning (refer,
395  Section 2). Therefore, we do not consider GDumb as the best-performing method, even when it achieves
396  higher accuracy.

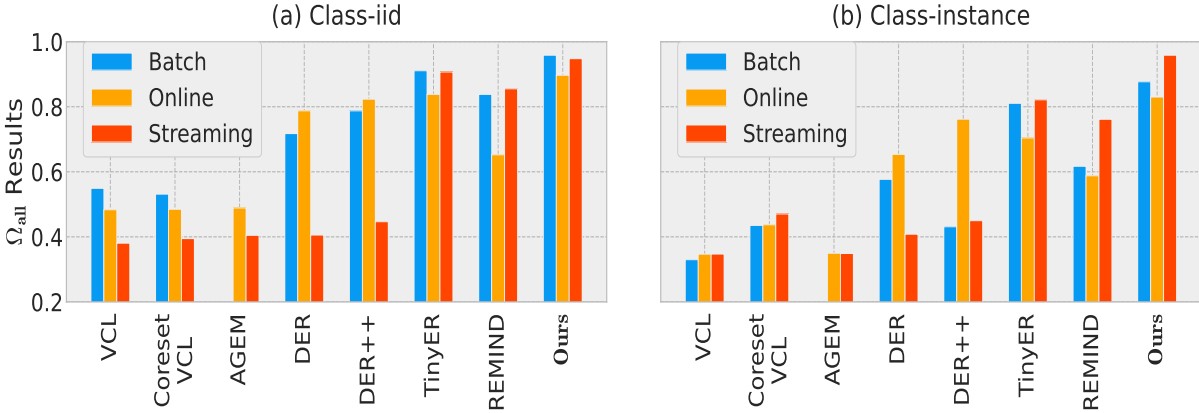

Figure 2: $\mathbf{\Omega}_{\text{all}}$ results for *'batch'*, *'online'* and *'streaming'* versions of baselines on iCubWorld 1.0 on (*a*) Class-iid, and (*b*) Class-instance ordering. An empty plot in AGEM indicates, we are unable to conduct experiment due to compatibility issues.

We believe it is important to highlight that iCubWorld 1.0 and CORe50 are two challenging datasets, which evaluate the models in more realistic scenarios or data-orderings. Particularly, class-instance and instance ordering require the learner to learn from temporally ordered video frames one at a time. From Table 3, we observe that BISLERi obtain up to 8.58% & 6.26% improvement on iCubWorld 1.0, and 5.47% & 12.8% improvement on CORe50, over the state-of-the-art streaming learning approaches. Figure 4 shows the impact of temporal orderings on the streaming learning model's performance. It is evident that class-instance and instance ordering are more difficult, and the baselines continue to suffer from severe forgetting. Figure 3 plots the accuracy ($\boldsymbol{\alpha}_t$) of BISLERi (Ours) and other baselines on (*i*) class-i.i.d and (*ii*) class-instance data-orderings on iCubWorld 1.0 and CORe50 datasets. It can be observed that BISLERi remembers the previous classes better than the other compared baselines. Furthermore, it can also be observed that BISLERi performs significantly better than the baselines in *class-instance* ordering, i.e., when there exists *temporal coherence* in the input data-stream.

Finally, for completeness, we train BISLERi in 'batch' as well as 'online' learning setting to determine its effectiveness and compatibility in these settings. In Figure 2, we compare BISLERi with various baselines. It can be observed that BISLERi outperforms the baselines by a significant margin on both class-i.i.d and class-instance ordering on iCubWorld. *It implies that, even though BISLERi designed to work in the streaming setting, it can be thought of as a robust method for various lifelong learning scenarios with the widest possible applicability.* We provide more details in the appendix.

## 5.5 Implementation Details

In all the experiments, models are trained with one sample at a time. For a fair comparison, the same network structure is used throughout all the models. For all methods, we use fully connected single-head networks with two hidden layers as the plastic network $F(\cdot)$, where each layer contains 256 nodes with ReLU activations; for 'VCL', 'Coreset VCL', 'Coreset Only' and 'BISLERi', $F(\cdot)$ is a BNN, whereas for all other methods $F(\cdot)$ is a deterministic network. For a fair comparison, we store the same number of past examples for all replay-based approaches. For REMIND, we compress and store the feature-maps with Faiss (Johnson et al., 2019) product quantization (PQ) implementation with $s = 32$ sub-vectors and codebook size $c = 256$.

We store the feature-map in memory for all the other methods, including our approach BISLERi. In addition, BISLERi also store the corresponding logits, loss-values, and uncertainty-scores. The capacity of our replay buffer is mentioned in Table 2. For memory-replay, we use *'uncertainty-aware positive-negative'* sampling strategy (discussed in Section 3.2) throughout all data-orderings, except for 'streaming-i.i.d' ordering, we use *'uniform'* sampling. We use *'loss-aware weighted random replacement with a reservoir'* sampling strategy as memory replacement policy for all the experiments. We store the same number of past examples in memory

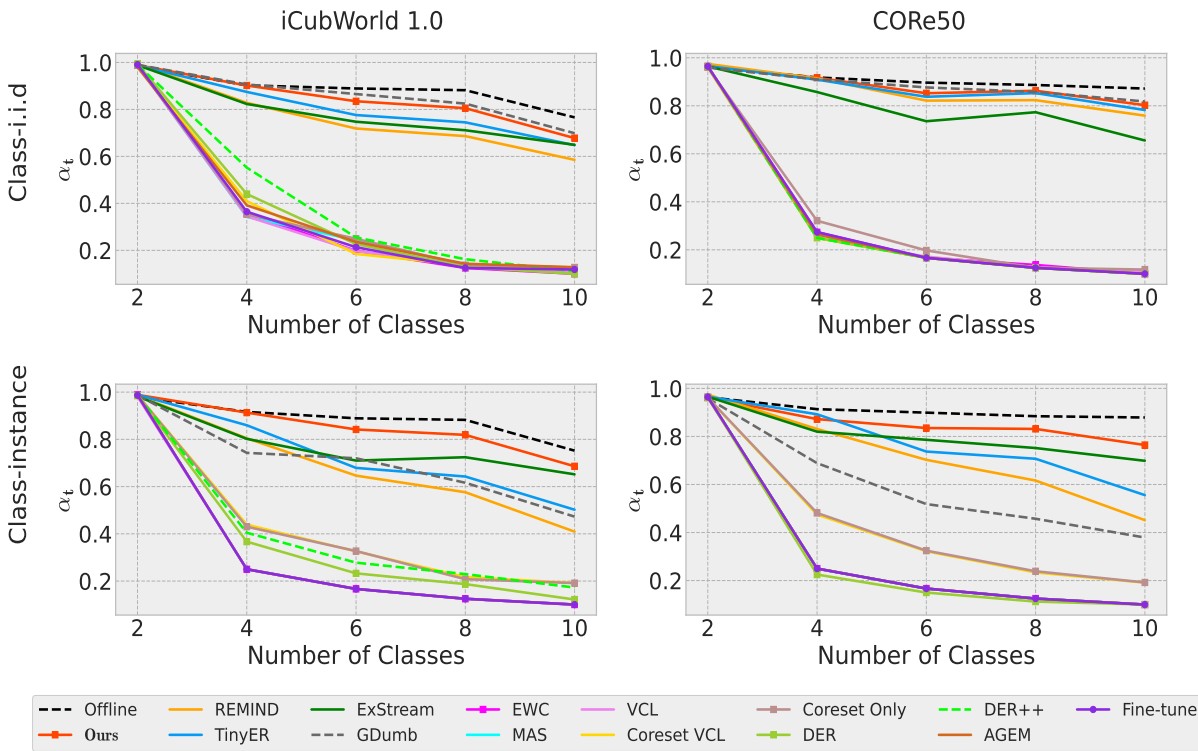

Figure 3: Performance of various incremental learning models on (*i*) streaming class-i.i.d (top row) and (*ii*) streaming class-instance (bottom row) ordering on iCubWorld 1.0 & CORe50 dataset. The plots suggest BISLERi (Ours) remembers earlier classes better than most existing algorithms. The performance gain is even more pronounced in streaming class-instance ordering setting (bottom row) where the baseline incremental learners suffer from severe forgetting. Recall that *GDumb cannot be considered as a streaming learning algorithm* as it requires fine-tuning.

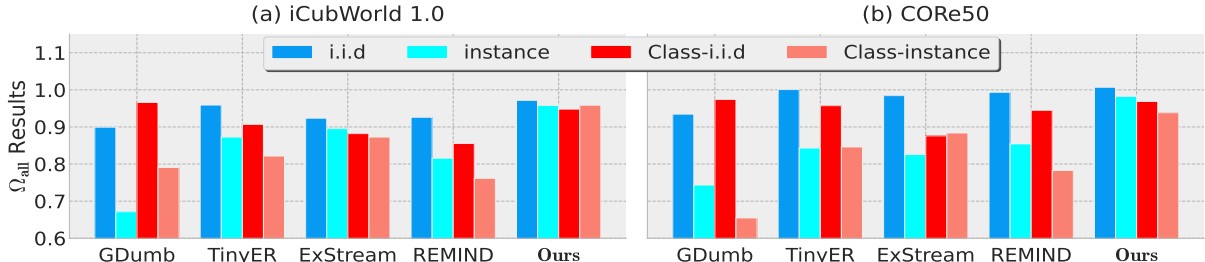

Figure 4: Plots of $\mathbf{\Omega}_{\text{all}}$ as a function of streaming learning model and data-ordering on (*a*) iCubWorld 1.0, and (*b*) CORe50. Only difference between i.i.d vs instance and class-i.i.d vs class-instance ordering, is the presence of temporal ordering (ref. Sec. 5.1); however, its effect on the streaming learner's performance is significant.

Table 4: $\boldsymbol{\Omega}_{\text{all}}$ Results with their associated standard deviations. For each experiment, the method with best performance is highlighted in **Bold**.

| Memory Replacement | Sample Selection | iCubWrold | | ImageNet100 | |
|---|---|---|---|---|---|
| | | instance | Class-instance | iid | Class-iid |
| LAWCBR | Uni | $0.8975 \pm 0.0454$ | $0.8506 \pm 0.0310$ | $0.9582 \pm 0.0037$ | $0.9014 \pm 0.0073$ |
| | UAPN | $0.9346 \pm 0.0395$ | $0.8500 \pm 0.0363$ | $0.9327 \pm 0.0052$ | $0.9135 \pm 0.0081$ |
| | LAPN | $0.9172 \pm 0.0373$ | $0.8536 \pm 0.0343$ | $0.9253 \pm 0.0115$ | $0.9122 \pm 0.0091$ |
| LAWRRR | Uni | $0.9269 \pm 0.0383$ | $0.9346 \pm 0.0191$ | $\mathbf{0.9640 \pm 0.0060}$ | $0.8643 \pm 0.0127$ |
| | UAPN | $\mathbf{0.9580 \pm 0.0298}$ | $\mathbf{0.9585 \pm 0.0223}$ | $0.9578 \pm 0.0035$ | $\mathbf{0.9171 \pm 0.0073}$ |
| | LAPN | $0.9558 \pm 0.0304$ | $0.9497 \pm 0.0239$ | $0.9575 \pm 0.0047$ | $0.9112 \pm 0.0075$ |

across all methods. For memory-replay, we use $N_1' = 16$ past samples throughout all experiments across BISLERi, AGEM, DER/DER++, TinyER, ExStream and REMIND. For knowledge-distillation, BISLERi and DER++ use $N_2' = 16$ samples at any time step $t$. We set the hyper-parameter $\lambda_1 = 1$ and $\lambda_2 = 0.3$ across all experiments; however, for *online/batch* learning experiments, we use $\lambda_2 = 0.2$ and use *uniform sampling* for memory replay. For EWC, we set hyper-parameter $\lambda = 500$, for MAS, we set hyper-parameter $\lambda = 1$, and for DER/DER++, we use $\alpha = \beta = 0.5$. We repeated each experiments for 10 times with different permutations of the data, and reported the results by taking average of 10 runs. More details are given in the appendix.

# 6 Ablation Study

We perform extensive ablation to show the importance of the different components. The various ablation experiments validate the significance of the proposed components.

**Significance Of Different Sampling Strategies.** In Table 4, we compare the performance of BISLERi while using various sampling strategies and memory replacement policies. We observe that for the buffer replacement, LAWRRR performs better compared to LAWCBR. Furthermore, for the sample replay, UAPN, along with LAWRRR memory buffer policy, outperforms other sampling strategies, except *uniform sampling* (Uni) performs better on i.i.d ordering. We provide more details in the appendix.

**Choice Of Hyperparameter ($\boldsymbol{\lambda}_2$).** Figure 5 shows the effect of changing the knowledge-distillation loss weight $\boldsymbol{\lambda}_2$ on the final $\boldsymbol{\Omega}_{\text{all}}$ accuracy for iCubWorld 1.0 on instance and class-instance ordering, while using different sampling strategies and buffer replacement policies. We observe the best model performance for $\boldsymbol{\lambda}_2 = 0.3$, and use this value for all our experiments. We provide detailed ablation on $\boldsymbol{\lambda}_2$ in the appendix.

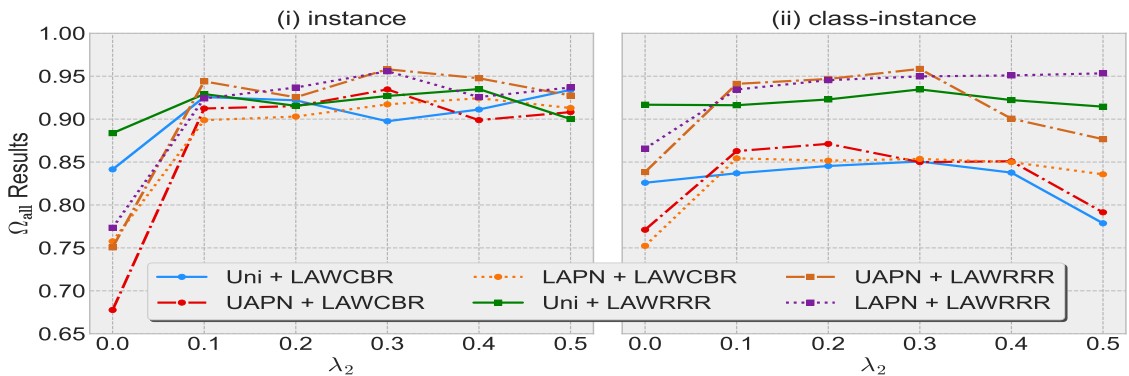

Figure 5: Plots of $\boldsymbol{\Omega}_{\text{all}}$ as a function of hyper-parameter $\boldsymbol{\lambda}_2$ and different sampling strategies and replacement policies for (*i*) instance, (*ii*) class-instance ordering on iCubWorld 1.0.

**Significance Of Knowledge-Distillation Loss.** Figure 5 with $\boldsymbol{\lambda}_2 = 0.0$ represents the model without knowledge distillation. We can observe that the model performance significantly degrades without knowledge

distillation. Therefore, knowledge distillation is a key component to the model's performance. More details are given in the appendix.

**Choice Of Buffer Capacity.** We perform an ablation for the different buffer capacities, i.e., $|\mathcal{M}|$. The results are shown in Figure 6. It is evident that, with the longer sequence of incoming data, the model's (BISLERi) performance improves with the increase in the buffer capacity, as it helps minimize the confusion in the output prediction.

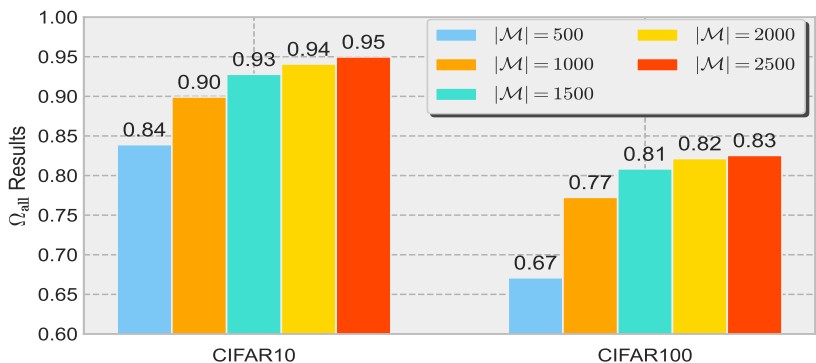

Figure 6: Plots of $\mathbf{\Omega}_{\text{all}}$ as a function of buffer capacity $|\mathcal{M}|$ for class-i.i.d data-ordering on CIFAR10 and CIFAR100.

# 7 Conclusion

*Streaming continual learning* (SCL) is the most challenging and realistic framework for continual learning; most of the recent promising models for the CL are unable to handle this above setting. Our work proposes a dual regularization and loss-aware buffer replacement to handle the SCL scenario. The proposed model is highly efficient since it learns a joint likelihood from the current and replay samples without leveraging any external finetuning. Also, to improve the training efficiency further, the proposed model selects a few most informative samples from the buffer instead of using the entire buffer for the replay. We have conducted a rigorous experiment over several challenging datasets and showed that BISLERi outperforms the recent state-of-the-art approaches in this setting by a significant margin. To disentangle the importance of the various components, we perform extensive ablation studies and observe that the proposed components are essential to handle the SCL setting.

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

## A Preliminaries

### A.1 'Class Incremental Learning' V/S 'Task Incremental Learning'

*'Class incremental learning'* (Rebuffi et al., 2017; Chaudhry et al., 2018a; Hayes et al., 2019a;b; Rios & Itti, 2018; Belouadah et al., 2020), is a challenging variant of lifelong learning, where the classifier needs to learn to discriminate between different class labels from different tasks. The key distinction between *'class incremental learning'* and *'task incremental learning'* (Kirkpatrick et al., 2017; Aljundi et al., 2018a;b; Nguyen et al., 2017; Zenke et al., 2017), lies in how the classifier's accuracy is evaluated at the test time. In *'class incremental learning'*, at the test time, the task identifier $t$ is not specified, and the accuracy is computed over all the observed classes with $\frac{1}{C}$ chance, where $C$ is the total number of classes accumulated so far. However, in *'task incremental learning'*, the task identifier $t$ is known.

For example, consider MNIST divided into 5 tasks: $\{\{0,1\},\ldots,\{8,9\}\}$, which are used for sequential learning of a classifier. Then, at the end of 5-th task, in *'task incremental setting'*, the classifier needs to predict a class out of $\{8,9\}$ only. However, in *'class incremental setting'*, a class label is predicted over all the ten classes that is observed so far, i.e., $\{0,\ldots,9\}$ with $\frac{1}{10}$ chance for each class.

### A.2 Variational Continual Learning (VCL)

Variational Continual Learning (VCL) (Nguyen et al., 2017) is a recently proposed continual learning approach that mitigates catastrophic forgetting (McCloskey & Cohen, 1989; French, 1999) in neural networks in a Bayesian framework (Neal, 2012; Jospin et al., 2020). It sets the posterior of parameters distribution as the prior before training on the next task, i.e., $p_t(\boldsymbol{\theta}) = q_{t-1}(\boldsymbol{\theta})$, the new task reuses the previous task's posterior as the new prior. VCL solves the following KL divergence minimization problem while training on task $t$ with the new data $\mathcal{D}_t$:

$$q_t(\boldsymbol{\theta}) = \underset{q \epsilon \mathcal{Q}}{\arg\min}\, \mathrm{KL}\left(q(\boldsymbol{\theta}) \;||\; \frac{1}{Z_t}q_{t-1}(\boldsymbol{\theta})\, p(\mathcal{D}_t|\boldsymbol{\theta})\right) \tag{10}$$

While offering a principled way of continual learning, VCL follows *task incremental learning* setting, and uses *'task specific head networks'*, for each task $t$, such that, $p(\boldsymbol{\theta}|\mathcal{D}_{1:t}) = p(\boldsymbol{\theta}_t|\mathcal{D}_t)p(\boldsymbol{\theta}_S|\mathcal{D}_{1:t})$, where $\boldsymbol{\theta} = \{\boldsymbol{\theta}_S, \boldsymbol{\theta}_t\}$, $\boldsymbol{\theta}_s$ is shared between all the tasks, whereas $\boldsymbol{\theta}_t$ kept fixed after training on task $t$. This configuration prohibits knowledge transfer across tasks, and results in a poor accuracy in *class incremental setting* (Farquhar & Gal, 2018) for both VCL with or without Coreset.

Coreset VCL (Nguyen et al., 2017) withhold some data points from the task data before training and keeps them in a coreset. These data points are not used for the network training and are only used for finetuning the network before each inference. However, in *streaming learning*, finetuning the network at any time is prohibited, as it makes the training process a *two-step* learning process instead of *single-pass* learning. Furthermore, the coreset is created by sampling data points from the entire task data, whereas in *streaming setting*, each instance arrives one at a time. Finally, the performance of *Coreset VCL* is heavily dependent on the finetuning with the coreset samples before inference (Farquhar & Gal, 2018), and still not comparable enough to our proposed method (BISLERi).

### A.3 REMIND

REMIND (Hayes et al., 2019b) is a recently proposed rehearsal-based lifelong learning approach which combats catastrophic forgetting (French, 1999) in deep neural network in *streaming setting*. While following such a challenging setting, it separates the convolutional neural network into two networks: (*i*) a frozen feature extractor and (*ii*) a plastic neural network. Learning involves the following steps: (*i*) compression of each new input using product quantization (PQ) (Jegou et al., 2010), (*ii*) reconstruction of the previously stored compressed representations using PQ, and (*iii*) mixing the reconstructed past examples with the new input and updating the parameters of the plastic layers of the network.

While it combats catastrophic forgetting and achieves state-of-the-art performance, there are few concerns that can be limiting in the continual learning setup. It stores considerably a large number of past examples

compared to the baselines; for example, iCaRL (Rebuffi et al., 2017) stores 10K past examples for ImageNet (ILSVRC-2012) (Russakovsky et al., 2015) experiment, whereas REMIND stores 1M past examples. Furthermore, REMIND actually uses a lossy compression method (PQ) to store the past samples, which is merely an engineering technique far from any algorithmic improvement and can be used by any lifelong learning approach.

### A.4 Bayesian Neural Network

Bayesian neural networks (Neal, 2012; Jospin et al., 2020) are discriminative models, which extend the standard deep neural networks with Bayesian inference. The network parameters are assumed to have a prior distribution, $p(\boldsymbol{\theta})$, and it infers the posterior given the observed data $\mathcal{D}$, that is, $p(\boldsymbol{\theta}|\mathcal{D})$. However, the exact posterior inference is computationally intractable for any complex models, and an approximation is needed. One such scheme is *'Bayes-by-Backprop'* (BBB) (Blundell et al., 2015). It uses a mean-field variational posterior $q(\boldsymbol{\theta})$ over the network parameters and uses reparameterization-trick (Kingma & Welling, 2013) to sample from the posterior, which are then used to approximate the evidence lower bound (ELBO) via Monte-Carlo sampling.

In our proposed method (BISLERi), we have used a Bayesian neural network (BNN) as the plastic network $F(\cdot)$. We have discussed training the plastic network (BNN) $F(\cdot)$ with a single step posterior update without catastrophic forgetting (French, 1999) in class-incremental streaming learning (CISL) setup in Section 3.1.

## B  Differences Between VCL/Coreset VCL and BISLERi

In this section, we describe the differences between VCL/Coreset VCL (Nguyen et al., 2017; Farquhar & Gal, 2018) and the proposed method (BISLERi). The differences are as follows -

- While BISLERi and VCL/Coreset VCL both utilizes Bayesian framework to enable continual learning in the deep neural networks, VCL/Coreset VCL is a 'incremental batch learning' (IBL) mathod in nature, whereas BISLERi is a streaming/online learning method. That is, in order to approximate the posterior in each incremental step, VCL/Coreset VCL requires visiting the data multiple times, whereas BISLERi approximates the posterior with a single gradient update. In doing so we need to obtain important modifications to obtain correct estimates of likelihood and updation of the posterior. Naively using VCL can be observed to perform quite inferior to the proposed solution. Our work is a principled adaptation of the formulation to the streaming learning setting and this is quite different from the continual learning based on 'batch-based updates'.

- Both VCL and Coreset VCL do not utilize any memory replay, while approximating the new posterior, whereas BISLERi replays a subset of the past stored samples along with the newly available sample in order to approximate the new posterior to enable continual learning. Approximating the posterior in this way, i.e., replaying a subset of past samples with the new sample, allows BISLERi to achieve *'any-time-inference'* ability, which is a key-requirement in streaming learning, as fine-tuning the network parameters with the stored samples is forbidden in the streaming learning setup.

- While VCL do not use coreset samples during any step of the learning, Coreset VCL withholds a few past samples in memory (coreset), which are then used for fine-tuning the network before inference. However, Coreset VCL stores the samples in the coreset in a *task-specific manner*, unlike the methods like GDumb (Prabhu et al., 2020), TinyER (Chaudhry et al., 2019), REMIND (Hayes et al., 2019b), BISLERi (Ours). We explain this with the below example.

  Consider MNIST divided into 5 tasks: $\{\{0, 1\}, \{2, 3\}, \ldots, \{8, 9\}\}$, which are used for sequential learning of a classifier. Therefore, in each incremental step, the classifier observes sample from only two classes. In this case, Coreset VCL stores samples in memory (coreset) in a task-specific manner. That is, it divides the coreset into 5 partitions, where each partition is used to store samples from a single task. Before inference, samples corresponding to the current task is utilized to fine-tune the network parameters to improve performance. For example, at the end of 5-th task, since the

classifier only needs to predict a class out of $\{8, 9\}$, Coreset VCL fine-tunes the network with the withheld samples corresponding to only class $\{8, 9\}$. While this strategy works nicely in case of 'task-incremental learning', it suffers severely in 'class-incremental learning' setup.

In contrast methods like GDumb (Prabhu et al., 2020), TinyER (Chaudhry et al., 2019), RE-MIND (Hayes et al., 2019b), BISLERi (Ours), do not store the samples in memory in a 'task-specific' manner, instead it is populated with the samples from all the classes. Therefore, when methods like GDumb, REMIND, BISLERi replays the past samples, it observe samples across all the classes irrespective of the tasks, whereas Coreset VCL only observes samples corresponding to the specific task, which causes Coreset VCL to suffer from poor generalization in case of 'class-incremental learning' setup.

- Coreset VCL withhold few data-points from the dataset, and do not utilize them during the incremental learning. These withheld samples are only used for fine-tuing the network parameters before inference. In contrast BISLERi maintains a replay buffer which is updated in an online manner as mentioned in Section 3.3. During streaming learning, in each incremental step a subset of samples are selected (Section 3.2) and combined with the newly available sample to compute the new posterior, which enables the network with the *'any-time-inference'* ability, a crucial property required in streaming learning.

### B.1 How VCL/Coreset VCL is adapted in the Streaming Learning?

In this subsection, we describe how the actual VCL/Coreset VCL (Nguyen et al., 2017) is adapted, so that it can be trained incrementally with a single training example in each incremental step in streaming learning.

For both VCL and Coreset VCL, we have used a single-headed Bayesian network as the plastic network ($F$), as also mentioned in Section 5.5. We follow the same strategy as mentioned by Nguyen et al. (2017); Farquhar & Gal (2018) to approximate the new posterior in each incremental step by combining the previously computed posterior with the new data-likelihood. For Coreset VCL, the samples are stored in memory (coreset) in a task-specific manner, while arriving one datum at a time in each incremental step. At the end of each task, Coreset VCL selects the samples specific to the current task and fine-tunes the network parameters. While fine-tuning the network is forbidden in streaming learning, it does not improve the network's overall performance in 'class-incremental learning' setup due to the above mentioned reasons.

## C Various Columns Of Table 1 In Detail

In this section, we describe the various columns that we have used to categorize the existing continual learning approaches on the basis of underlying assumptions as they impose. That is, we categorize each continual learning according to various constraints that they follow/mention in the respective literature.

- **Type.** Each CL approach is classified into one of three types: ($i$) incremental batch learning (IBL/Batch), ($ii$) online learning (Online), and ($iii$) streaming learning (Streaming).

  The key difference between IBL and online/streaming learning approaches is that IBL approaches visit the data multiple times, perform multiple gradient update to adapt to the newly available data. While these approaches works nicely in a static environment, these methods can be applied in a dynamic non-stationary environment. In contrast, online/streaming learning approaches adapt to the newly available data in a single gradient update.

  The key difference between an online learning and a streaming learning approaches mainly lies on whether a method is allowed to do fine-tuning or not. While both the approaches learns with a single gradient update, in online learning, it is permitted to fine-tune the network parameters with the stored samples any time. However, this would also imply that an online learning approach involves multiple gradient updates to improve its performance, which is forbidden in streaming learning. For example, GDumb (Prabhu et al., 2020) requires fine-tuning before each inference, therefore, it uses multiple gradient updated, ultimately violating the single-pass learning constraint of the streaming

learning. On the other hand, in streaming learning, no single method is allowed to use any additional computation, such as fine-tuning, to improve its performance. For example, BISLERi, REMIND do not use any fine-tuning at any stage of learning.

- **Bayesian Framework.** Whether a method uses a Bayesian framework/formulation or not.

- **Batch Size.** IBL and online learning method assumes that a batch of samples arrive in each incremental step, where the batch size: $N_t \gg 1$. In constrast, streaming learning approaches assume that incremental step, only a single training example arrives, such that, the batch size: $N_t = 1$.

- **Fine-tunes.** Whether a method requires fine-tuning or not.

- **Single Pass Learning.** In online/streaming learning, each newly available (training) sample(s) is only allowed to observe only once without storing it in a memory (replay buffer), and requires to be adapted in a single gradient update. This is refered as single-pass learning. In each incremental step, however, it is allowed to replay past observed samples stored in memory along with the newly available data. Please also refer to REMIND (Hayes et al., 2019b) (Section 2), where they have defined this single pass learning formulation to emphasize that we have not invented a new problem formulation, it was already existing.

  In addition, in online learning, it is allowed to fine-tune the network with the stored samples by repeating the fine-tuning for multiple epochs, multiple times. However, this implies that the network would use multiple gradient update instead of a single gradient update to improve its performance, which is essentially forbidden in streaming learning.

  MIR (Aljundi et al., 2019) violates the single pass learning constraint of streaming learning, by employing a two step/pass learning strategy. Initially, it uses the newly available sample(s) to perform a parameter update to select the maximally interfered past stored samples from memory to be used for experience replay. Finally, it combines the new available sample(s), already used once for a gradient update, with the selected maximally interfered samples to perform another (final) gradient update. Therefore, MIR essentially uses a two step/pass learning, instead of a single pass learning as required in streaming learning. For more details on the streaming learning constraints refer to Section 2.

  GDumb (Prabhu et al., 2020) requires fine-tuning the network parameters for multiple epochs, multiple times with the stored replay buffer samples before each inference, as it does not employ any learning when it observes a new sample in each incremental step. It implies that GDumb requires multiple gradient update to improve its performance, ultimately violates the single pass learning constraint.

- **Class Incremental Learning (CIL).** Whether supports class-incremental learning or not.

- **Subset Buffer Replay.** Whether replays a subset of samples selected from memory or replays all the samples stored in memory in each incremental step.

  For example, ExStream (Hayes et al., 2019a) uses memory-replay to enable streaming learning, however, in doing so, it replays all the stored samples along with the newly available sample in each incremental step. While it mitigates catastrophic forgetting in the network, it limits its practical applicability due to obvious reasons. That is, if the buffer capacity is considerably large then time required to complete a single gradient update will also be large. On the other hand, the methods like REMIND (Hayes et al., 2019b), DER/DER++ (Buzzega et al., 2020), BISLERi (Ours) uses subset buffer replay to enable streaming learning. That is, it select only a few samples from memory, combines them with the newly available sample in order to perform single gradient update to enable continual learning, which is computationally an efficient choice.

- **Training Time.** Training time column denotes the number of gradient updates required by the corresponding method according to the underlying assumption that method impose. Therefore, $\zeta(n)$ denotes that the corresponding method would require $n$ gradient update in order to enable continual learning.

For example, EWC (Kirkpatrick et al., 2017), MAS (Aljundi et al., 2018a), VCL/Coreset VCL (Nguyen et al., 2017) visits the data multiple times, performs multiple gradient update, to enable continual learning, therefore, its training time is represented with $\zeta(n)$. On the other hand, methods such as DER/DER++ (Buzzega et al., 2020), REMIND (Hayes et al., 2019b), BISLERi (Ours) can adapt to the newly available in a single gradient update, hence, the training time is $\zeta(1)$.

- **Inference Time.** Similar to Training time column, it denotes the number of gradient updates required by the corresponding method according to the underlying assumption that method impose.

  Methods which do not require fine-tuning can be evaluated directly, therefore, inference time is represented as $\zeta(1)$. However, methods which require fine-tuning before inference, such as GDumb (Prabhu et al., 2020), uses multiple gradient updates to improve its performance, therefore, inference time is represented as $\zeta(n)$, where $n$ denotes the number of gradient updates used during fine-tuning the network.

- **Violates Any CISL Constraint.** Whether violates any 'class-incremental streaming learning' (CISL) constraints or not.

- **Memory Capacity.** It denotes the number of past observed samples are stored in the replay buffer (memory).

- **Regularization Based.** Whether a method uses parameter regularization or not. That is, if a method qualifies as a regularization based method, then it uses parameters regularization to enable continual learning.

  Parisi et al. (2019), Delange et al. (2021) have classified the existing continual learning approaches on the basis of the mechanisms for mitigating catastrophic forgetting into three main categories, namely: ($i$) parameter isolation based approaches, ($ii$) regularization based approaches, and ($iii$) rehearsal / memory-replay based approaches. In this paper, we have followed this same classification to classify the existing CL approaches into one of those three classes.

- **Memory Based.** Whether a method uses memory-replay or not to enable continual learning.

## D Importance Of Streaming Learning

Importance of 'class-incremental streaming learning' (CISL) or 'streaming learning' (SL) can be described as follows:

- It enables practical deployment of the AI agents in real world scenarios, where an AI agent might need to learn from as few as a single (training) example without suffering from the catastrophic forgetting. For example, consider an autonomous car might meet with a rare incident/accident, then it could be lifesaving if it can be trained continuously with that single example without any forgetting. It would be impractical, if not infeasible, to wait and aggregate a batch of samples to train the autonomous agent, as we may not collect a batch of such examples due to its rare nature. Hayes et al. (2019b) refered to streaming learning as the closest alternative to the biological learning than the other existing lifelong learning approaches, due to the fact that it enables continual learning from a single example with no forgetting.

- IBL methods assume the data available in batches and can visit the data multiple times to enable CL. While it can be applicable in a static environment, it lacks the applicability in a rapidly changing dynamic environment, where a learner needs to adapt quickly in a single pass, such that, it achieves *'any-time-inference'* ability. Although, the existing online learning approaches aim to enable continual learning in a dynamic environment from a non-stationary data-stream, these methods have number of limitations, such as: ($i$) require batch-size, ($ii$) require fine-tuning before each inference, ($iii$) require large replay buffer, etc., which limits their applicability in a restrictive streaming lifelong learning. Streaming learning approaches addresses the limitations of the existing IBL and online learning approaches and enables lifelong learning following various constraints: ($i$) single pass

learning, (*ii*) subset buffer replay, (*iii*) tiny replay buffer, etc. It further enables *'any-time-inference'* ability in a continual learner, which enables practical applicability in real world scenarios. Finally, it also enables lifelong learning from a temporally coherent video sequences (images), which are naturally non-i.i.d images. Please refer to Section 2 in REMIND (Hayes et al., 2019b) paper that has defined this problem formulation to emphasize that we have not invented a new problem formulation, it was already existing.

## E   Baselines And Compared Methods In Detail

The proposed approach (BISLERi) follows *'class-incremental streaming learning'* setup, to the best of our knowledge, recent works ExStream (Hayes et al., 2019a), and REMIND (Hayes et al., 2019b) are the only method that trains a *deep neural network* following the same learning setting. We compared BISLERi against these strong baselines. In addition, we have compared various 'batch' and 'online' learning methods, which we describe below.

For a fair comparison, we follow a similar network structure throughout all the methods. We separate a convolutional neural network (CNN) into two networks: (*i*) *non-plastic* feature extractor $G(\cdot)$, and (*ii*) plastic neural network $F(\cdot)$. For a given input image $\boldsymbol{x}$, the predicted class label is computed as: $y = F(G(\boldsymbol{x}))$. Across all the methods, we use the same initialization step for the feature extractor $G(\cdot)$ (discussed in Section 3.5) and keep it frozen throughout the *streaming learning*. For all the methods, only the plastic network $F(\cdot)$ is trained with one sample at a time in *streaming manner*. For details on the structure of the plastic network $F(\cdot)$ across baselines along with BISLERi, refer to Section 5.5.

In the below, we describe the baselines which we have evaluated along with BISLERi in *class-incremental streaming setting*:

1. **EWC** (Kirkpatrick et al., 2017)**:** It is a regularization-based incremental learning method, which penalizes any changes to the network parameters by the important weight measure, the diagonal of the Fisher information matrix.

2. **MAS** (Aljundi et al., 2018a)**:** It is another regularization-based lifelong learning method, where the importance weight of the network parameters are estimated by measuring the magnitude of the gradient of the learned function.

3. **VCL** (Nguyen et al., 2017)**:** It uses variational inference (VI) with a Bayesian neural network (Neal, 2012; Jospin et al., 2020) to mitigate catastrophic forgetting, where it uses the previously learned posterior as the prior while learning incrementally with the sequentially coming data. For more details, please refer to Section A.2.

4. **Coreset VCL** (Nguyen et al., 2017)**:** This method is the same as the pure VCL as mentioned above, except, at the end of training on each task, the network is finetuned with the coreset samples. We adapted the coreset selection in streaming setting and stored data points in coreset in an online manner; before inference, the network is fine-tuned with the coreset samples.

5. **Coreset Only** (Farquhar & Gal, 2018)**:** This method is exactly similar to *Coreset VCL* (Nguyen et al., 2017), except the prior which is used for variational inference is the initial prior each time, i.e., it is not updated with the previous posterior before training on a new task.

6. **GDumb** (Prabhu et al., 2020)**:** It is an online learning method. It stores data points with a greedy sampler and retrains the network from scratch each time with stored samples before inference.

7. **AGEM** (Chaudhry et al., 2018b)**:** It is another online learning approach. It uses past task data stored in memory to build an optimization constraint to be satisfied by each new update. If the gradient violates the constraint, then it is projected such that the constraint is satisfied.

8. **DER** (Buzzega et al., 2020)**:** It stores the past logits in memory and matches them while learning on the new data.

9. **DER++** (Buzzega et al., 2020)**:** It stores the past task data points along with the corresponding logits in memory. It uses memory-replay and knowledge-distillation to enable continual learning while learning on the new data.

10. **TinyER** (Chaudhry et al., 2019)**:** It stores past task data points in a tiny episodic memory and replays them with the current training data to enable continual learning.

11. **ExStream** (Hayes et al., 2019a)**:** It is a streaming learning method, which uses memory replay to enable continual learning. It maintains buffers of prototypes to store the input vectors. Once the buffer is full, it combines the two nearest prototypes in the buffer and stores the new input vector.

12. **REMIND** (Hayes et al., 2019b)**:** Similar to ExStream, it is another streaming learning method, which enables lifelong learning with memory replay. For more details on REMIND, please refer to Section A.3.

13. **Fine-tuning:** It is a streaming learning baseline and serves as the lower bound on the network's performance. In this scenario, the network parameters are fine-tuned with one instance through the whole dataset for a single epoch.

14. **Offline:** It serves as the upper bound on the network's performance, where the network is trained in the traditional way, that is, the complete dataset is divided into multiple batches, and the network loops over them multiple times.

**Note:** In *streaming learning*, fine-tuning the network parameters at any time is prohibited (refer Section 2), and thus, any method which uses fine-tuning has an extra advantage over the true streaming learning methods, and cannot be considered as the best performing method even when they are achieving the best accuracy. Coreset VCL (Nguyen et al., 2017), Coreset Only (Farquhar & Gal, 2018) and GDumb (Prabhu et al., 2020), however, requires fine-tuning before each inference, therefore, we do not consider these methods as the best performing method over the true streaming learning methods.

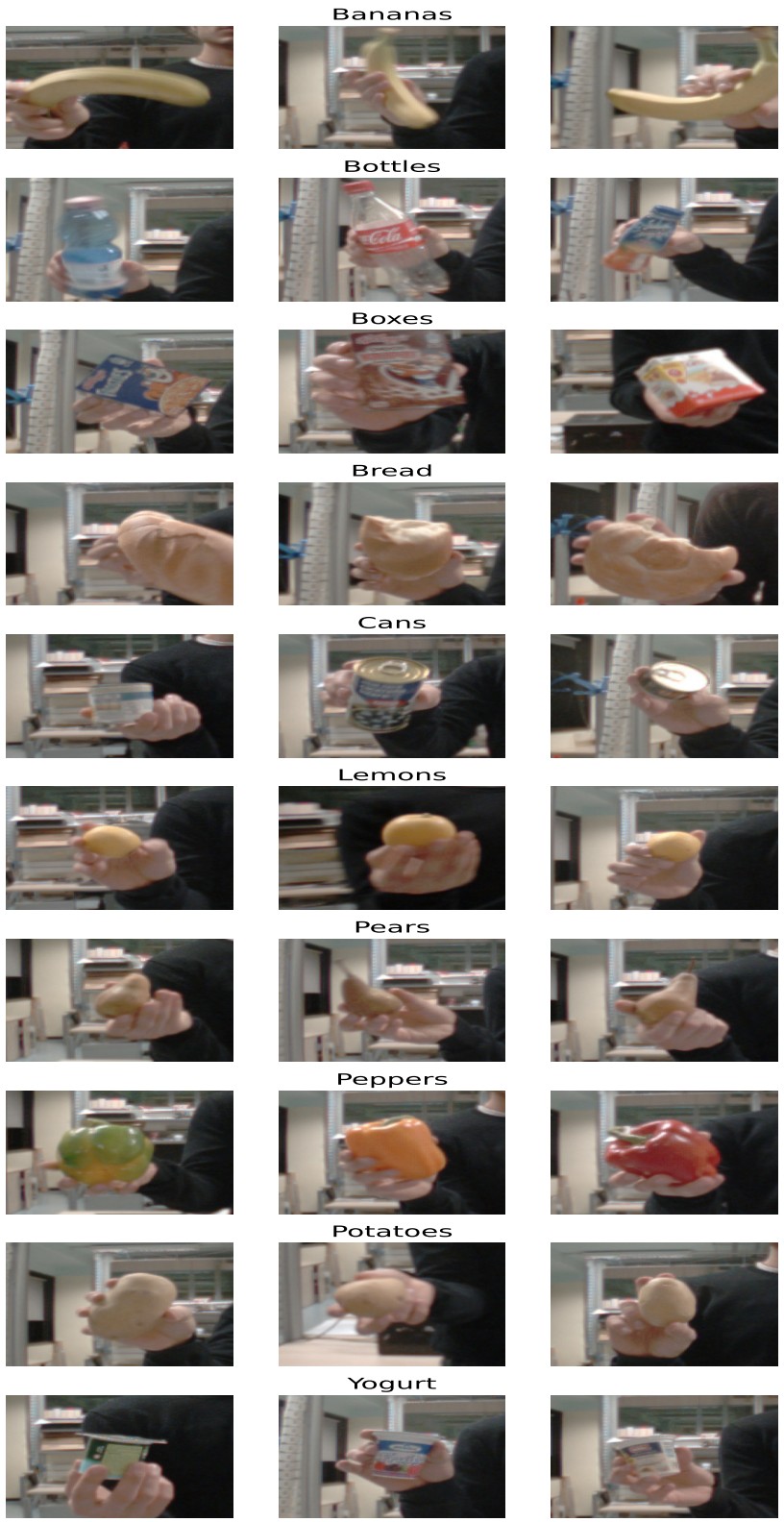

Figure 7: The iCubWorld 1.0 (Fanello et al., 2013) dataset. 10 categories: Bananas, Bottles, Boxes, Bread, Cans, Lemons, Pears, Peppers, Potatoes, Yogurt. Each category contains 3 different instances.

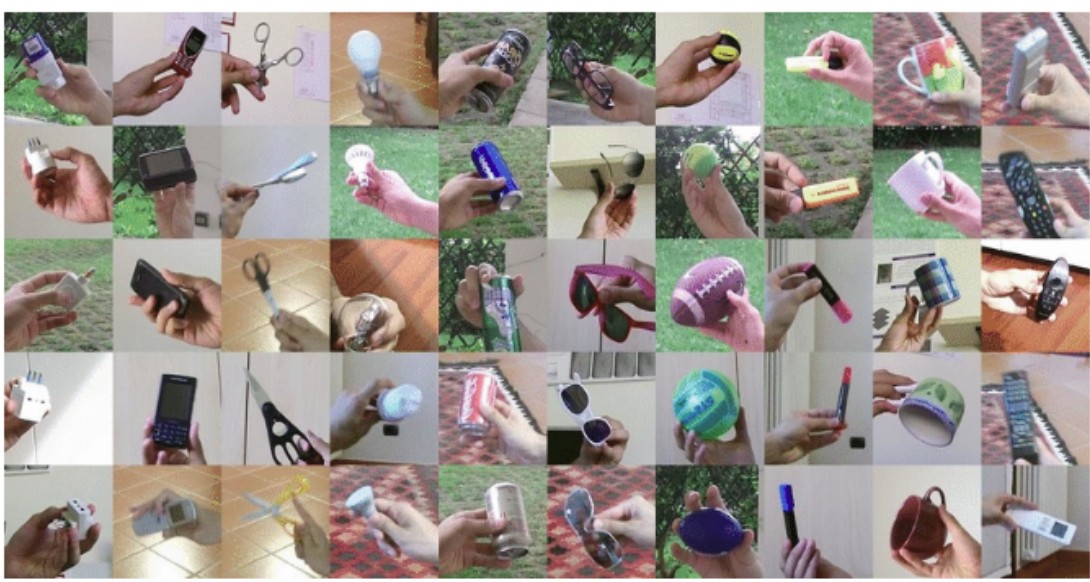

Figure 8: Example images of the 50 object instances in CORe50 (Lomonaco & Maltoni, 2017). Each column denotes one of the 10 categories.

## F    Ablation Study Additional Results

In this section, we provide additional results for the ablation studies.

- **CIFAR10/100.** Table 5 compares the final $\Omega_{\text{all}}$ accuracy of the proposed model (BISLERi) for ($i$) i.i.d and ($ii$) class-i.i.d ordering on CIFAR10 and CIFAR100 respectively while using different values for the knowledge-distillation hyper-parameter $\lambda_2$, and different memory replacement policies and various sample selection strategies.

- **iCubWorld 1.0.** Table 6 and Table 7 compares the final $\Omega_{\text{all}}$ accuracy of BISLERi for ($i$) i.i.d, ($ii$) class-i.i.d, ($iii$) instance, and ($iv$) class-instance ordering on iCubWorld 1.0 dataset while using different knowledge-distillation hyper-parameter $\lambda_2$ and different sampling strategies. For memory replacement policy, Table 6 uses *'loss-aware weighted class balancing replacement (LAWCBR)'* strategy, whereas Table 7 uses *'loss-aware weighted random replacement with a reservoir (LAWRRR)'* strategy.

- **CORe50.** In Table 8, we compare the final $\Omega_{\text{all}}$ accuracy of BISLERi for ($i$) i.i.d, ($ii$) class-i.i.d, ($iii$) instance, and ($iv$) class-instance ordering on CORe50 dataset while using different memory replacement policy and past sample selection strategies.

Table 5: $\mathbf{\Omega}_{\text{all}}$ Results as a function of knowledge-distillation hyper-parameter $\lambda_2$ and different memory replacement policies and sample selection strategies for ($i$) i.i.d ordering, and ($ii$) class-i.i.d ordering on CIFAR10 and CIFAR100 datasets.

| $\lambda_2$ | Memory Replacement | Sample Selection | iid | | class-iid | |
| --- | --- | --- | --- | --- | --- | --- |
| | | | CIFAR10 | CIFAR100 | CIFAR10 | CIFAR100 |
| 0.2 | LAWCBR | Uni | 0.9542 ± 0.0053 | 0.8135 ± 0.0054 | 0.8942 ± 0.0062 | 0.7343 ± 0.0131 |
| | | UAPN | 0.9084 ± 0.0121 | 0.4760 ± 0.0136 | 0.8957 ± 0.0125 | 0.6448 ± 0.0257 |
| | | LAPN | 0.8462 ± 0.0414 | 0.3834 ± 0.0335 | 0.8797 ± 0.0149 | 0.5332 ± 0.0310 |
| | LAWRRR | Uni | **0.9584 ± 0.0035** | 0.8617 ± 0.0091 | 0.8792 ± 0.0104 | 0.7221 ± 0.0149 |
| | | UAPN | 0.9567 ± 0.0031 | 0.8366 ± 0.0107 | 0.8978 ± 0.0107 | 0.7589 ± 0.0185 |
| | | LAPN | 0.9530 ± 0.0037 | 0.8273 ± 0.0141 | 0.8986 ± 0.0127 | 0.7478 ± 0.0191 |
| 0.3 | LAWCBR | Uni | 0.9529 ± 0.0062 | 0.8134 ± 0.0077 | 0.8970 ± 0.0088 | 0.7369 ± 0.0106 |
| | | UAPN | 0.9145 ± 0.0071 | 0.5096 ± 0.0088 | 0.8944 ± 0.0093 | 0.6836 ± 0.0231 |
| | | LAPN | 0.9046 ± 0.0152 | 0.4376 ± 0.0220 | 0.8798 ± 0.0230 | 0.6275 ± 0.0291 |
| | LAWRRR | Uni | 0.9579 ± 0.0040 | **0.8679 ± 0.0057** | 0.8838 ± 0.0088 | 0.7307 ± 0.0122 |
| | | UAPN | 0.9567 ± 0.0031 | 0.8542 ± 0.0066 | 0.8991 ± 0.0089 | **0.7724 ± 0.0188** |
| | | LAPN | 0.9538 ± 0.0044 | 0.8453 ± 0.0120 | **0.9024 ± 0.0116** | 0.7573 ± 0.0193 |

Table 6: $\mathbf{\Omega}_{\text{all}}$ Results as a function of knowledge-distillation hyper-parameter $\lambda_2$, and *'loss-aware weighted class balancing replacement'* (LAWCBR) and different sampling strategies for ($i$) i.i.d, ($ii$) class-i.i.d, ($iii$) instance, and ($iv$) class-instance ordering on iCubWorld 1.0 dataset.

| $\lambda_2$ | Sample Selection | iCubWorld 1.0 | | | |
| --- | --- | --- | --- | --- | --- |
| | | iid | Class-iid | Instance | Class-instance |
| 0.0 | Uni | 0.9431 ± 0.0418 | 0.9105 ± 0.0333 | 0.8414 ± 0.0541 | 0.8259 ± 0.0316 |
| | UAPN | 0.8775 ± 0.0753 | 0.8863 ± 0.0529 | 0.6777 ± 0.0764 | 0.7711 ± 0.0574 |
| | LAPN | 0.8975 ± 0.0697 | 0.8675 ± 0.0498 | 0.7576 ± 0.0739 | 0.7524 ± 0.0655 |
| 0.1 | Uni | **0.9885 ± 0.0245** | 0.9163 ± 0.0237 | 0.9257 ± 0.0299 | 0.8369 ± 0.0329 |
| | UAPN | 0.9781 ± 0.0318 | 0.9167 ± 0.0263 | 0.9124 ± 0.0525 | 0.8627 ± 0.0285 |
| | LAPN | 0.9779 ± 0.0206 | 0.9224 ± 0.0332 | 0.8988 ± 0.0544 | 0.8543 ± 0.0288 |
| 0.2 | Uni | 0.9841 ± 0.0178 | 0.9154 ± 0.0217 | 0.9219 ± 0.0333 | 0.8454 ± 0.0283 |
| | UAPN | 0.9868 ± 0.0181 | 0.9293 ± 0.0306 | 0.9152 ± 0.0229 | **0.8712 ± 0.0266** |
| | LAPN | 0.9645 ± 0.0189 | 0.9310 ± 0.0227 | 0.9030 ± 0.0503 | 0.8516 ± 0.0400 |
| 0.3 | Uni | 0.9777 ± 0.0264 | 0.9257 ± 0.0288 | 0.8975 ± 0.0454 | 0.8506 ± 0.0310 |
| | UAPN | 0.9868 ± 0.0125 | 0.9309 ± 0.0355 | **0.9346 ± 0.0395** | 0.8500 ± 0.0363 |
| | LAPN | 0.9745 ± 0.0174 | **0.9352 ± 0.0266** | 0.9172 ± 0.0373 | 0.8536 ± 0.0343 |
| 0.4 | Uni | 0.9782 ± 0.0200 | 0.9278 ± 0.0295 | 0.9112 ± 0.0327 | 0.8377 ± 0.0292 |
| | UAPN | 0.9815 ± 0.0178 | 0.9160 ± 0.0464 | 0.8988 ± 0.0419 | 0.8509 ± 0.0350 |
| | LAPN | 0.9718 ± 0.0271 | 0.9325 ± 0.0401 | 0.9243 ± 0.0512 | 0.8499 ± 0.0650 |
| 0.5 | Uni | 0.9742 ± 0.0183 | 0.8858 ± 0.1505 | 0.9341 ± 0.0350 | 0.7787 ± 0.2008 |
| | UAPN | 0.9692 ± 0.0197 | 0.8587 ± 0.2278 | 0.9082 ± 0.0758 | 0.7914 ± 0.2033 |
| | LAPN | 0.9725 ± 0.0184 | 0.9006 ± 0.0635 | 0.9129 ± 0.0467 | 0.8357 ± 0.0334 |

Table 7: $\mathbf{\Omega}_{\text{all}}$ Results as a function of knowledge-distillation hyper-parameter $\lambda_2$, and *'loss-aware weighted random replacement with a reservoir'* (LAWRRR) and different sampling strategies for (*i*) i.i.d, (*ii*) class-i.i.d, (*iii*) instance, and (*iv*) class-instance ordering on iCubWorld 1.0 dataset.

| $\lambda_2$ | Sample Selection | iCubWorld 1.0 | | | |
|---|---|---|---|---|---|
| | | iid | Class-iid | Instance | Class-instance |
| 0.0 | Uni | $0.9298 \pm 0.0329$ | $0.9063 \pm 0.0396$ | $0.8837 \pm 0.0544$ | $0.9168 \pm 0.0312$ |
| | UAPN | $0.9184 \pm 0.0379$ | $0.8818 \pm 0.0396$ | $0.7507 \pm 0.0732$ | $0.8384 \pm 0.0675$ |
| | LAPN | $0.9285 \pm 0.0357$ | $0.8912 \pm 0.0430$ | $0.7735 \pm 0.0458$ | $0.8657 \pm 0.0521$ |
| 0.1 | Uni | $\mathbf{0.9830 \pm 0.0207}$ | $0.9240 \pm 0.0276$ | $0.9292 \pm 0.0344$ | $0.9162 \pm 0.0255$ |
| | UAPN | $0.9644 \pm 0.0260$ | $0.9368 \pm 0.0228$ | $0.9439 \pm 0.0362$ | $0.9411 \pm 0.0224$ |
| | LAPN | $0.9541 \pm 0.0280$ | $0.9402 \pm 0.0368$ | $0.9241 \pm 0.0401$ | $0.9345 \pm 0.0235$ |
| 0.2 | Uni | $0.9600 \pm 0.0312$ | $0.9351 \pm 0.0315$ | $0.9155 \pm 0.0299$ | $0.9229 \pm 0.0284$ |
| | UAPN | $0.9640 \pm 0.0236$ | $0.9415 \pm 0.0307$ | $0.9254 \pm 0.0331$ | $0.9468 \pm 0.0273$ |
| | LAPN | $0.9684 \pm 0.0160$ | $0.9382 \pm 0.0361$ | $0.9368 \pm 0.0376$ | $0.9454 \pm 0.0263$ |
| 0.3 | Uni | $0.9716 \pm 0.0141$ | $0.9118 \pm 0.0344$ | $0.9269 \pm 0.0383$ | $0.9346 \pm 0.0191$ |
| | UAPN | $0.9454 \pm 0.0239$ | $0.9480 \pm 0.0215$ | $\mathbf{0.9580 \pm 0.0298}$ | $\mathbf{0.9585 \pm 0.0223}$ |
| | LAPN | $0.9667 \pm 0.0174$ | $\mathbf{0.9538 \pm 0.0303}$ | $0.9558 \pm 0.0304$ | $0.9497 \pm 0.0239$ |
| 0.4 | Uni | $0.9611 \pm 0.0153$ | $0.9243 \pm 0.0524$ | $0.9350 \pm 0.0319$ | $0.9222 \pm 0.0403$ |
| | UAPN | $0.9647 \pm 0.0257$ | $0.9387 \pm 0.0315$ | $0.9476 \pm 0.0264$ | $0.9005 \pm 0.1257$ |
| | LAPN | $0.9615 \pm 0.0194$ | $0.9323 \pm 0.0421$ | $0.9257 \pm 0.0212$ | $0.9509 \pm 0.0323$ |
| 0.5 | Uni | $0.9615 \pm 0.0301$ | $0.9391 \pm 0.0268$ | $0.9001 \pm 0.0555$ | $0.9145 \pm 0.0649$ |
| | UAPN | $0.9526 \pm 0.0179$ | $0.9390 \pm 0.0267$ | $0.9275 \pm 0.0322$ | $0.8766 \pm 0.2320$ |
| | LAPN | $0.9495 \pm 0.0215$ | $0.9085 \pm 0.1230$ | $0.9369 \pm 0.0187$ | $0.9533 \pm 0.0248$ |

Table 8: $\mathbf{\Omega}_{\text{all}}$ Results as a function of different memory replacement policies and sample selection strategies for (*i*) i.i.d, (*ii*) class-i.i.d, (*iii*) instance, and (*iv*) class-instance ordering on CORe50.

| Memory Replacement | Sample Selection | CORe50 | | | |
|---|---|---|---|---|---|
| | | iid | Class-iid | instance | Class-instance |
| LAWCBR | Uni | $1.0069 \pm 0.0058$ | $\mathbf{0.9686 \pm 0.0122}$ | $0.8644 \pm 0.0237$ | $0.7913 \pm 0.0434$ |
| | UAPN | $\mathbf{1.0079 \pm 0.0047}$ | $0.8890 \pm 0.1965$ | $0.8976 \pm 0.0189$ | $0.7835 \pm 0.0701$ |
| | LAPN | $1.0065 \pm 0.0045$ | $0.8871 \pm 0.2356$ | $0.8888 \pm 0.0198$ | $0.6979 \pm 0.2043$ |
| LAWRRR | Uni | $0.9974 \pm 0.0075$ | $0.9146 \pm 0.0275$ | $0.9711 \pm 0.0077$ | $0.9093 \pm 0.0222$ |
| | UAPN | $0.9935 \pm 0.0050$ | $0.9200 \pm 0.0408$ | $0.9824 \pm 0.0090$ | $\mathbf{0.9384 \pm 0.0130}$ |
| | LAPN | $0.9933 \pm 0.0056$ | $0.9101 \pm 0.0538$ | $\mathbf{0.9835 \pm 0.0046}$ | $0.8932 \pm 0.0671$ |

### G   ImageNet-100

In this paper, we used a subset of ImageNet-1000 (ILSVRC-2012) (Russakovsky et al., 2015) that contains randomly chosen 100 classes. To ease a relevant study, we release the list of these 100 classes that we used to evaluate the streaming learner's performance in our experiments, as mentioned in Table 9.

Table 9: The list of classes from ImageNet-100, which are randomly chosen from the original ImageNet-1000 (ILSVRC-2012) (Russakovsky et al., 2015).

| List Of ImageNet-100 Classes | | | |
|---|---|---|---|
| n01632777 | n01667114 | n01744401 | n01753488 |
| n01768244 | n01770081 | n01798484 | n01829413 |
| n01843065 | n01871265 | n01872401 | n01981276 |
| n02006656 | n02012849 | n02025239 | n02085620 |
| n02086079 | n02089867 | n02091831 | n02094258 |
| n02096294 | n02100236 | n02100877 | n02102040 |
| n02105251 | n02106550 | n02110627 | n02120079 |
| n02130308 | n02168699 | n02169497 | n02177972 |
| n02264363 | n02417914 | n02422699 | n02437616 |
| n02483708 | n02488291 | n02489166 | n02494079 |
| n02504013 | n02667093 | n02687172 | n02788148 |
| n02791124 | n02794156 | n02814860 | n02859443 |
| n02895154 | n02910353 | n03000247 | n03208938 |
| n03223299 | n03271574 | n03291819 | n03347037 |
| n03445777 | n03529860 | n03530642 | n03602883 |
| n03627232 | n03649909 | n03666591 | n03761084 |
| n03770439 | n03773504 | n03788195 | n03825788 |
| n03866082 | n03877845 | n03908618 | n03916031 |
| n03929855 | n03954731 | n04009552 | n04019541 |
| n04141327 | n04147183 | n04235860 | n04285008 |
| n04286575 | n04328186 | n04347754 | n04355338 |
| n04423845 | n04442312 | n04456115 | n04485082 |
| n04486054 | n04505470 | n04525038 | n07248320 |
| n07716906 | n07730033 | n07768694 | n07836838 |
| n07860988 | n07871810 | n11939491 | n12267677 |

### H   Average Accuracy $(\mu_{\text{all}})$

In the main paper, we use $\mathbf{\Omega}_{\text{all}}$ metric to compare the performance of streaming learners across datasets and data-orderings. However, it can hide the raw performance, since it provides a relative performance with respect to an Offline model. Therefore, in this section, we provide average accuracy metric over all testing events, similar to (Rebuffi et al., 2017; Hayes et al., 2019b; Kemker et al., 2018):

$$\boldsymbol{\mu}_{\text{all}} = \frac{1}{T} \sum_{t=1}^{T} \boldsymbol{\alpha}_t \tag{11}$$

where $T$ is the total number of testing events, and $\boldsymbol{\alpha}_t$ is the accuracy of the streaming learner at time $t$.

In Table 14, we provide $\boldsymbol{\mu}_{\text{all}}$ results with their associated standard-deviations (for corresponding $\mathbf{\Omega}_{\text{all}}$ results, refer Table 3) comparing the performance of BISLERi (Ours) and other baselines in different streaming learning scenarios on different datasets. We do not consider GDumb (Prabhu et al., 2020) as the best performing model even when it achieves higher accuracy on several datasets on class-i.i.d ordering, since it violates the streaming learning constraints (refer Section 2).

Figure 10, 11, 12 plots the accuracy ($\boldsymbol{\alpha}_t$) of BISLERi (Ours) and other baselines on ($i$) CIFAR10, ($ii$) CIFAR100, ($iii$) ImageNet100 dataset. Figure 13 is the (partial) zoom-in version of the plot in Figure 3 comparing the accuracy ($\boldsymbol{\alpha}_t$) of BISLERi (Ours) and other baselines on iCubWorld 1.0 & CORe50 datasets.

## I  Ablation Study: $\mu_{\text{all}}$ Results As A Function Of Feature-Extractor

In this section, we compare the performance of various baselines (in Table 10) for ($i$) class-instance and ($ii$) instance ordering on iCubWorld 1.0 using a feature extractor trained with ($i$) supervised image-classification loss, and ($ii$) self-supervised loss. For this experiment, we have used ResNet-50 (He et al., 2016) as the base architecture of the feature-extractor. In supervised setup, ResNet-50 is trained with cross-entropy loss, whereas in self-supervised setup, ResNet-50 is trained with momentum contrastive loss (MoCoV2 (Chen et al., 2020)).

It can be observed that in both ordering the final accuracy drops across streaming learning methods including BISLERi (Ours), if we use a feature-extractor trained with self-supervised loss. However, it is worth mentioning that BISLERi (Ours) still achieves superior performance compared to the other streaming learning baselines.

Table 10: $\mu_{\text{all}}$ results as a function of feature-extractor trained with ($i$) supervised loss and ($ii$) self-supervised loss on iCubWorld 1.0.
Note: Methods in Red use fine-tuning, implying that these methods violate streaming learning (SL) constraints and have an extra advantage over true streaming learning (SL) methods, such as 'Ours'.

| Method | Class-instace | | instace | |
|---|---|---|---|---|
| | Resnet50 Supervised | Resnet50 Self-Supervised (MoCoV2) | Resnet50 Supervised | Resnet50 Self-Supervised (MoCoV2) |
| Fine tune | $0.3282 \pm 0.0003$ | $0.3276 \pm 0.0008$ | $0.1000 \pm 0.0000$ | $0.1000 \pm 0.0002$ |
| VCL | $0.3281 \pm 0.0003$ | $0.3269 \pm 0.0014$ | - | - |
| Coreset VCL | $0.4331 \pm 0.0296$ | $0.3949 \pm 0.0309$ | - | - |
| GDumb | $0.7455 \pm 0.0219$ | $0.4623 \pm 0.0638$ | $0.5442 \pm 0.0475$ | $0.1821 \pm 0.0274$ |
| AGEM | $0.3280 \pm 0.0006$ | $0.3277 \pm 0.0008$ | $0.1000 \pm 0.0000$ | $0.1000 \pm 0.0000$ |
| TinyER | $0.7369 \pm 0.0282$ | $0.6859 \pm 0.0273$ | $0.6503 \pm 0.0302$ | $0.6331 \pm 0.0363$ |
| ExStream | $0.7919 \pm 0.0200$ | $0.6863 \pm 0.0305$ | $0.6470 \pm 0.0325$ | $0.6026 \pm 0.0319$ |
| REMIND | $0.6754 \pm 0.0293$ | $0.6584 \pm 0.0258$ | $0.6114 \pm 0.0391$ | $0.5355 \pm 0.0750$ |
| Ours | $\mathbf{0.8303 \pm 0.0313}$ | $\mathbf{0.7596 \pm 0.0419}$ | $\mathbf{0.6926 \pm 0.0120}$ | $\mathbf{0.6872 \pm 0.0207}$ |
| Offline | 0.8676 | 0.8368 | 0.7311 | 0.7741 |

## J  Ablation Study: $\mu_{\text{all}}$ Results As A Function Of Learning-Rate

In this section, we compare the performance ($\mu_{\text{all}}$ Results) of the baselines including BISLERi (Ours) (in Table 11) as a function of learning rate (lr) across: ($i$) class-instance and ($ii$) instance ordering on iCubWorld 1.0 dataset. We can observe that BISLERi (Ours) achieves superior performance compared to the baselines while trained with different learning rates.

Table 11: $\mu_{\text{all}}$ results as a function of learning rate (lr) on iCubWorld 1.0.
Note: Methods in Red use fine-tuning, implying that these methods violate streaming learning (SL) constraints and have an extra advantage over true streaming learning (SL) methods, such as 'Ours'.

| Method | Class-instace | | | instace | | |
|---|---|---|---|---|---|---|
| | lr = 0.01 | lr = 0.001 | lr = 0.003 | lr = 0.01 | lr = 0.001 | lr = 0.003 |
| Fine tune | $0.3258 \pm 0.0022$ | $0.3264 \pm 0.0039$ | $0.3267 \pm 0.0013$ | $0.1000 \pm 0.0000$ | $0.1121 \pm 0.0265$ | $0.1070 \pm 0.0221$ |
| VCL | $0.3246 \pm 0.0024$ | $0.3242 \pm 0.0016$ | $0.3248 \pm 0.0020$ | - | - | - |
| Coreset VCL | $0.4319 \pm 0.0148$ | $0.4316 \pm 0.0161$ | $0.4288 \pm 0.0157$ | - | - | - |
| GDumb | $0.7077 \pm 0.0293$ | $0.6983 \pm 0.0273$ | $0.6989 \pm 0.0255$ | $0.5134 \pm 0.0413$ | $0.4646 \pm 0.0732$ | $0.5332 \pm 0.0408$ |
| TinyER | $0.7346 \pm 0.0287$ | $0.7177 \pm 0.0157$ | $0.7082 \pm 0.0172$ | $0.6672 \pm 0.0496$ | $0.6982 \pm 0.0392$ | $0.7071 \pm 0.0251$ |
| ExStream | $0.7740 \pm 0.0198$ | $0.7842 \pm 0.0171$ | $0.7848 \pm 0.0149$ | $0.6846 \pm 0.0414$ | $0.6964 \pm 0.0232$ | $0.7138 \pm 0.0160$ |
| REMIND | $0.6843 \pm 0.0270$ | $0.6783 \pm 0.0227$ | $0.6597 \pm 0.0206$ | $0.6237 \pm 0.0459$ | $0.6582 \pm 0.0358$ | $0.6518 \pm 0.0411$ |
| Ours | $\mathbf{0.8497 \pm 0.0191}$ | $\mathbf{0.8416 \pm 0.0262}$ | $\mathbf{0.8458 \pm 0.0186}$ | $\mathbf{0.7325 \pm 0.0228}$ | $\mathbf{0.7141 \pm 0.0271}$ | $\mathbf{0.7280 \pm 0.0268}$ |
| Offline | 0.8840 | 0.8877 | 0.8912 | 0.7646 | 0.7681 | 0.7551 |

## K    Ablation Study: GPU Memory And Computation Time Requirements For The Tested Methods

In this section, we provide: ($i$) gpu memory consumption, and ($ii$) total time required by the corresponding method for streaming learning (in Table 12) on iCubWorld 1.0 for class-instance ordering.

Table 12: Comparison of baselines w.r.t ($i$) gpu memory consumption, and ($ii$) total time required in streaming learning on iCubWorld 1.0 for class-instance ordering.

| Method | GPU Memory Consumption (Mb) | Time |
| --- | --- | --- |
| Fine-Tune | 933 | 34m 03s |
| EWC | 933 | 41m 23s |
| MAS | 933 | 54m 16s |
| VCL | 933 | 53m 25s |
| DER/DER++ | 933 | 61m 25s |
| TinyER | 933 | 54m 19s |
| REMIND | 933 | 63m 21s |
| Ours | 986 | 73m 25s |

It can be observed that the gpu memory consumption and required time to train in streaming learning, is not significantly higher than the other baselines. However, it results in superior performance compared to the baselines.

## L    Ablation Study: $\mu_{\text{all}}$ Results As A Function Of Hyperparameters ($\alpha$ & $\beta$) On DER/DER++

While in the main paper, we use hyperparameters: $\alpha = \beta = 0.5$, to evalaute the performance of DER/DER++ (Buzzega et al., 2020), in this section (Figure 9), we compare the performance ($\boldsymbol{\mu}_{\text{all}}$) of DER/DER++ w.r.t the different values of hyperparameters: $\alpha$ and $\beta$. and learning rate (lr) on class-instace ordering on iCubWorld 1.0.

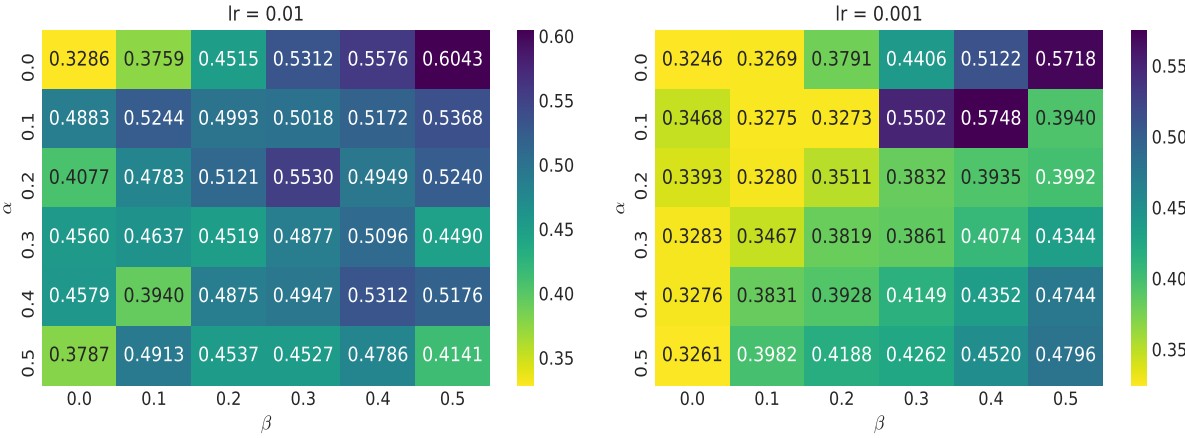

Figure 9: $\mu_{\text{all}}$ results as a function of hyperparameters ($\alpha$ & $\beta$) and learning rate (lr) on DER/DER++.

Table 13: $\mu_{\text{all}}$ results as a function of hyperparameters ($\alpha \& \beta$) and learning rate (lr) on DER/DER++ over class-instace ordering on iCubWorld 1.0.

| DER++ | lr = 0.01 | lr = 0.001 |
|---|---|---|
| $\alpha = 0.0, \beta = 0.0$ | $0.3268 \pm 0.0010$ | $0.3246 \pm 0.0060$ |
| $\alpha = 0.0, \beta = 0.1$ | $0.3759 \pm 0.0148$ | $0.3269 \pm 0.0017$ |
| $\alpha = 0.0, \beta = 0.2$ | $0.4515 \pm 0.0322$ | $0.3791 \pm 0.0154$ |
| $\alpha = 0.0, \beta = 0.3$ | $0.5312 \pm 0.0380$ | $0.4406 \pm 0.0236$ |
| $\alpha = 0.0, \beta = 0.4$ | $0.5576 \pm 0.0516$ | $0.5122 \pm 0.0406$ |
| $\alpha = 0.0, \beta = 0.5$ | $0.6043 \pm 0.0437$ | $0.5718 \pm 0.0354$ |
| $\alpha = 0.0, \beta = 0.6$ | $0.6449 \pm 0.0568$ | $0.5745 \pm 0.0349$ |
| $\alpha = 0.0, \beta = 0.7$ | $0.6442 \pm 0.0704$ | $0.5995 \pm 0.0385$ |
| $\alpha = 0.0, \beta = 0.8$ | $0.6817 \pm 0.0709$ | $0.6078 \pm 0.0396$ |
| $\alpha = 0.0, \beta = 0.9$ | $0.6846 \pm 0.0603$ | $0.6127 \pm 0.0466$ |
| $\alpha = 0.0, \beta = 1.0$ | $0.7419 \pm 0.0523$ | $0.6276 \pm 0.0408$ |
| Ours | $\mathbf{0.8497 \pm 0.0191}$ | $\mathbf{0.8416 \pm 0.0262}$ |
| Offline | 0.8840 | 0.8877 |

- For $\beta = 0.0$ and $\alpha = 0.0$, DER++ behaves similar to Fine-Tune model (lower bound). It simply updates the parameters of the network with the gradient computed against the newly available single training example in each incremental step.

- For $\beta = 0.0$ and $\alpha > 0.0$, DER++ becomes DER (left most column in both sides in Figure 9), and it uses only knowledge-distillation to mitigate catastrophic forgetting.

- For $\alpha = 0.0$ and $\beta > 0.0$, DER++ behaves similar to a method using only experience replay to mitigate catastrophic forgetting (top row in both sides in Figure 9). In Table 13, we have compared the performance ($\mu_{\text{all}}$ Results) of DER++ for $\alpha = 0.0$, $\beta \in \{0.0, 0.1, 0.2, 0.3, 0.4, 0.5, 0.6, 0.7, 0.8, 0.9, 1.0\}$, and lr $\in \{0.01, 0.001\}$. It can be observed that DER++ behaves similar to an experience replay based model with increasing value of $\beta$, and achieves the highest final accuracy when $\alpha = 0.0, \beta = 1.0$, for both learning rates (lr $\in \{0.01, 0.001\}$).

- For $\alpha > 0.0$ and $\beta > 0.0$, DER++ (Buzzega et al., 2020) uses both knowledge-distillation and experience replay to circumvent the catastrophic forgetting (French, 1999).

## M  Additional Implementation Details

We use Mobilenet-V2 (Sandler et al., 2018) pre-trained on ImageNet (Russakovsky et al., 2015) available in PyTorch (Paszke et al., 2019) TorchVision package as the base architecture for the feature extractor $G(\cdot)$. We use the convolutional base of Mobilenet-V2 (Sandler et al., 2018) as the feature extractor $G(\cdot)$ to obtain embeddings from the raw pixels; we keep it frozen throughout the streaming learning. We use 'loss-aware weighted class balancing replacement' as memory replacement policy and 'uniform sampling' strategy to select informative past samples for training on CORe50 on streaming i.i.d and streaming class-i.i.d ordering. We provide the parameter settings for the proposed method (BISLERi) and the offline models in Table 15.

Table 14: $\mu_{\text{all}}$ results with their associated standard deviations. For each experiment, the method with best performance in *'streaming-setting'* is highlighted in **Bold**. The reported results are average over 10 runs with different permutations of the data. Offline model is trained only once. '-' indicates experiments we are unable to run, because of compatibility issues.

Note: Methods in Red use fine-tuning, implying that these methods violate streaming learning (SL) constraints and have an extra advantage over true streaming learning (SL) methods, such as 'Ours'.

| Method | iid | | | Class-iid | | |
|---|---|---|---|---|---|---|
| | CIFAR10 | CIFAR100 | ImageNet100 | CIFAR10 | CIFAR100 | ImageNet100 |
| Fine-Tune | $0.1000 \pm 0.0000$ | $0.0109 \pm 0.0021$ | $0.0108 \pm 0.0025$ | $0.3250 \pm 0.0003$ | $0.1099 \pm 0.0017$ | $0.1138 \pm 0.0048$ |
| EWC | - | - | - | $0.3249 \pm 0.0002$ | $0.1111 \pm 0.0027$ | $0.1139 \pm 0.0036$ |
| MAS | - | - | - | $0.3271 \pm 0.0070$ | $0.1102 \pm 0.0022$ | $0.1148 \pm 0.0043$ |
| VCL | - | - | - | $0.3245 \pm 0.0005$ | $0.1095 \pm 0.0029$ | $0.1121 \pm 0.0014$ |
| Coreset VCL | - | - | - | $0.3488 \pm 0.0445$ | $0.1204 \pm 0.0170$ | $0.1170 \pm 0.0112$ |
| Coreset Only | - | - | - | $0.3462 \pm 0.0394$ | $0.1216 \pm 0.0194$ | $0.1182 \pm 0.0166$ |
| GDumb | $0.7391 \pm 0.0056$ | $0.3690 \pm 0.0072$ | $0.7124 \pm 0.0059$ | ***0.8324 ± 0.0050*** | $0.5560 \pm 0.0069$ | ***0.8248 ± 0.0072*** |
| AGEM | $0.1000 \pm 0.0000$ | $0.0111 \pm 0.0021$ | $0.0118 \pm 0.0035$ | $0.3251 \pm 0.0002$ | $0.1109 \pm 0.0026$ | $0.1130 \pm 0.0023$ |
| DER | $0.1000 \pm 0.0000$ | $0.0101 \pm 0.0023$ | $0.0107 \pm 0.0023$ | $0.3252 \pm 0.0002$ | $0.1101 \pm 0.0019$ | $0.1132 \pm 0.0035$ |
| DER++ | $0.1000 \pm 0.0000$ | $0.0105 \pm 0.0017$ | $0.0111 \pm 0.0034$ | $0.3374 \pm 0.0374$ | $0.1111 \pm 0.0042$ | $0.1144 \pm 0.0062$ |
| TinyER | $0.7925 \pm 0.0097$ | $0.4616 \pm 0.0078$ | $0.8021 \pm 0.0073$ | $0.8046 \pm 0.0138$ | $0.5410 \pm 0.0137$ | $0.8070 \pm 0.0108$ |
| ExStream | $0.7544 \pm 0.0208$ | $0.4772 \pm 0.0074$ | $0.7918 \pm 0.0070$ | $0.7345 \pm 0.0185$ | $0.5239 \pm 0.0146$ | $0.7854 \pm 0.0132$ |
| REMIND | $0.7581 \pm 0.0062$ | $0.3928 \pm 0.0056$ | $0.7743 \pm 0.0093$ | $0.7962 \pm 0.0176$ | $0.4984 \pm 0.0152$ | $0.7901 \pm 0.0139$ |
| **Ours** | $\mathbf{0.8151 \pm 0.0034}$ | $\mathbf{0.5279 \pm 0.0035}$ | $\mathbf{0.8213 \pm 0.0051}$ | $\mathbf{0.8099 \pm 0.0079}$ | $\mathbf{0.5611 \pm 0.0135}$ | $\mathbf{0.8224 \pm 0.0065}$ |
| Offline | 0.8509 | 0.6083 | 0.8520 | 0.8972 | 0.7154 | 0.8953 |

| Method | iid | | Class-iid | | instance | | Class-instance | |
|---|---|---|---|---|---|---|---|---|
| | iCubWorld 1.0 | CORe50 | iCubWorld 1.0 | CORe50 | iCubWorld 1.0 | CORe50 | iCubWorld 1.0 | CORe50 |
| Fine-Tune | $0.1044 \pm 0.0141$ | $0.1000 \pm 0.0000$ | $0.3625 \pm 0.0467$ | $0.3261 \pm 0.0157$ | $0.1000 \pm 0.0000$ | $0.1000 \pm 0.0000$ | $0.3258 \pm 0.0022$ | $0.3212 \pm 0.0003$ |
| EWC | - | - | $0.3539 \pm 0.0378$ | $0.3281 \pm 0.0221$ | - | - | $0.3260 \pm 0.0033$ | $0.3209 \pm 0.0007$ |
| MAS | - | - | $0.3644 \pm 0.0541$ | $0.3212 \pm 0.0004$ | - | - | $0.3259 \pm 0.0018$ | $0.3212 \pm 0.0004$ |
| VCL | - | - | $0.3550 \pm 0.0466$ | $0.3237 \pm 0.0114$ | - | - | $0.3246 \pm 0.0024$ | $0.3203 \pm 0.0009$ |
| Coreset VCL | - | - | $0.3674 \pm 0.0487$ | $0.3204 \pm 0.0018$ | - | - | $0.4319 \pm 0.0148$ | $0.4366 \pm 0.0050$ |
| Coreset Only | - | - | $0.3711 \pm 0.0813$ | $0.3443 \pm 0.0451$ | - | - | $0.4287 \pm 0.0226$ | $0.4396 \pm 0.0032$ |
| GDumb | $0.6858 \pm 0.0315$ | $0.8161 \pm 0.0106$ | ***0.8571 ± 0.0175*** | ***0.8842 ± 0.0074*** | $0.5134 \pm 0.0413$ | $0.6491 \pm 0.0215$ | $0.7077 \pm 0.0293$ | $0.6005 \pm 0.0233$ |
| AGEM | $0.1000 \pm 0.0000$ | $0.1000 \pm 0.0000$ | $0.3758 \pm 0.0555$ | $0.3238 \pm 0.0093$ | $0.1001 \pm 0.0002$ | $0.1000 \pm 0.0000$ | $0.3262 \pm 0.0029$ | $0.3211 \pm 0.0004$ |
| DER | $0.1096 \pm 0.0300$ | $0.1000 \pm 0.0000$ | $0.3779 \pm 0.0940$ | $0.3212 \pm 0.0004$ | $0.2875 \pm 0.1838$ | $0.1020 \pm 0.0063$ | $0.3787 \pm 0.1466$ | $0.3102 \pm 0.0347$ |
| DER++ | $0.1089 \pm 0.0278$ | $0.1000 \pm 0.0000$ | $0.4143 \pm 0.1146$ | $0.3211 \pm 0.0003$ | $0.3454 \pm 0.1919$ | $0.1000 \pm 0.0000$ | $0.4141 \pm 0.2016$ | $0.3211 \pm 0.0005$ |
| TinyER | $0.7313 \pm 0.0289$ | $0.8739 \pm 0.0106$ | $0.8062 \pm 0.0257$ | $0.8693 \pm 0.0111$ | $0.6672 \pm 0.0496$ | $0.7364 \pm 0.0229$ | $0.7346 \pm 0.0287$ | $0.7715 \pm 0.0221$ |
| ExStream | $0.7043 \pm 0.0445$ | $0.8597 \pm 0.0137$ | $0.7839 \pm 0.0247$ | $0.7970 \pm 0.0148$ | $0.6846 \pm 0.0414$ | $0.7211 \pm 0.0258$ | $0.7740 \pm 0.0198$ | $0.8044 \pm 0.0188$ |
| REMIND | $0.7062 \pm 0.0237$ | $0.8675 \pm 0.0100$ | $0.7623 \pm 0.0297$ | $0.8584 \pm 0.0111$ | $0.6237 \pm 0.0459$ | $0.7462 \pm 0.0215$ | $0.6843 \pm 0.0270$ | $0.7152 \pm 0.0337$ |
| **Ours** | $\mathbf{0.7409 \pm 0.0107}$ | $\mathbf{0.8794 \pm 0.0050}$ | $\mathbf{0.8417 \pm 0.0187}$ | $\mathbf{0.8793 \pm 0.0109}$ | $\mathbf{0.7325 \pm 0.0228}$ | $\mathbf{0.8579 \pm 0.0079}$ | $\mathbf{0.8497 \pm 0.0191}$ | $\mathbf{0.8531 \pm 0.0117}$ |
| Offline | 0.7626 | 0.8733 | 0.8849 | 0.9070 | 0.7646 | 0.8733 | 0.8840 | 0.9079 |

Table 15: Training parameters used for BISLERi and Offline model.

| Parameters | Datasets | | | | |
|---|---|---|---|---|---|
| | CIFAR10 | CIFAR100 | ImageNet100 | iCubWorld 1.0 | CORe50 |
| Optimizer | SGD | SGD | SGD | SGD | SGD |
| Learning Rate | 0.01 | 0.01 | 0.01 | 0.01 | 0.01 |
| Momentum | 0.9 | 0.9 | 0.9 | 0.9 | 0.9 |
| Weight Decay | 1e-05 | 1e-05 | 1e-05 | 1e-05 | 1e-05 |
| Hidden Layer | [256, 256] | [256, 256] | [256, 256] | [256, 256] | [256, 256] |
| Activation | ReLU | ReLU | ReLU | ReLU | ReLU |
| Offline Batch Size | 128 | 128 | 256 | 16 | 256 |
| Offline Epoch | 50 | 50 | 100 | 30 | 50 |
| Buffer Capacity | 1000 | 1000 | 1000 | 180 | 100 |
| Train-Set Size | 50000 | 50000 | 127778 | 6002 | 119894 |

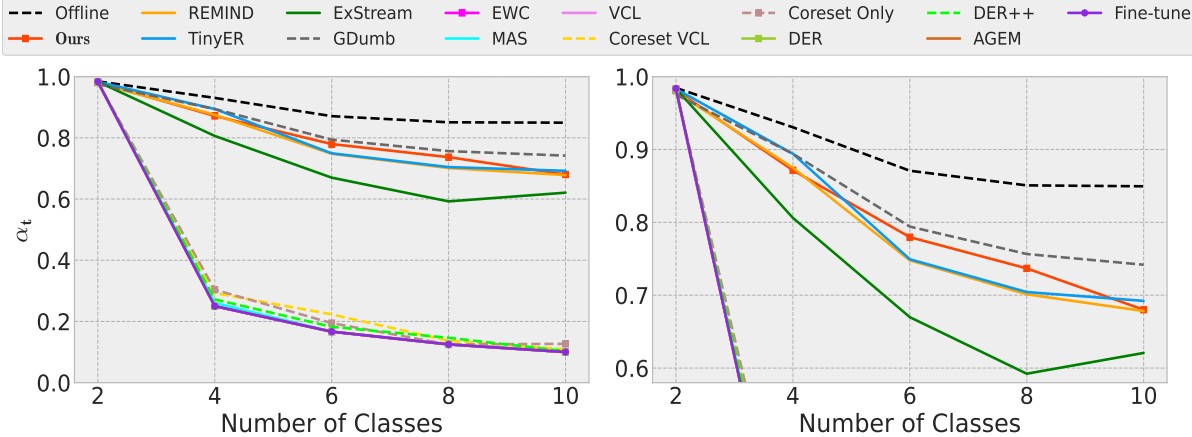

Figure 10: Performance of various incremental learning methods in streaming class iid setting on CIFAR10 dataset. The plot on the right is a (partial) zoom-in version of the left plot. It can be observed that BISLERi (Ours) remembers earlier classes better than most existing models as examples from new classes arrive in a streaming fashion. Recall that *GDumb cannot be considered as a streaming learning algorithm* as it requires fine-tuning.

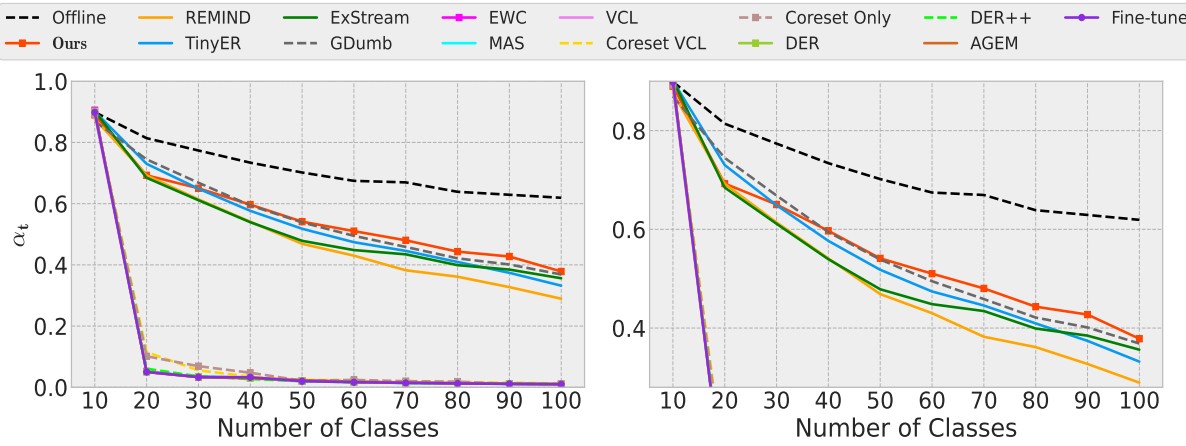

Figure 11: Performance of various incremental learning methods in streaming class iid setting on CIFAR100 dataset. The plot on the right is a (partial) zoom-in version of the left plot. It can be observed that BISLERi (Ours) remembers earlier classes better than existing models as examples from new classes arrive in a streaming fashion. Recall that *GDumb cannot be considered as a streaming learning algorithm* as it requires fine-tuning.

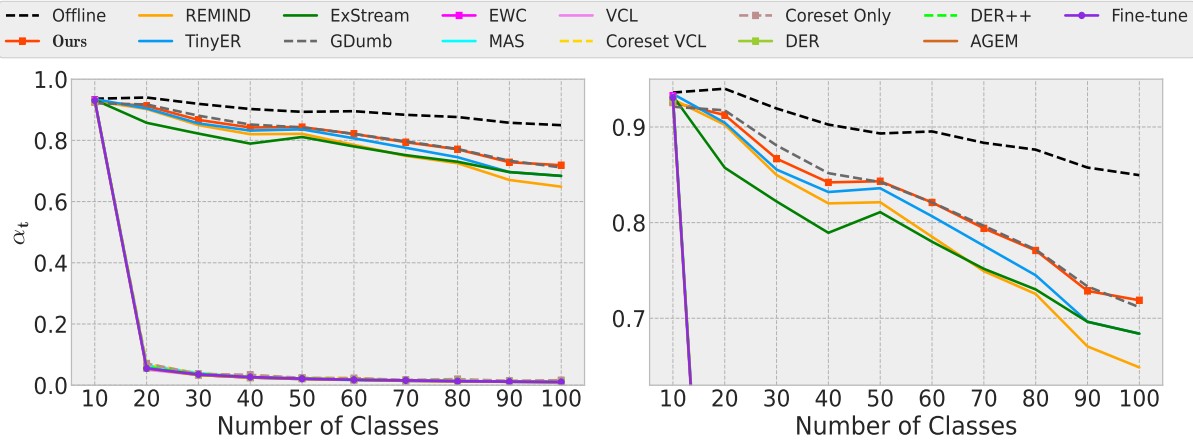

Figure 12: Performance of various incremental learning methods in streaming class iid setting on ImageNet100 dataset. The plot on the right is a (partial) zoom-in version of the left plot. It can be observed that BISLERi (Ours) remembers earlier classes better than existing models as examples from new classes arrive in a streaming fashion. Recall that *GDumb cannot be considered as a streaming learning algorithm* as it requires fine-tuning.

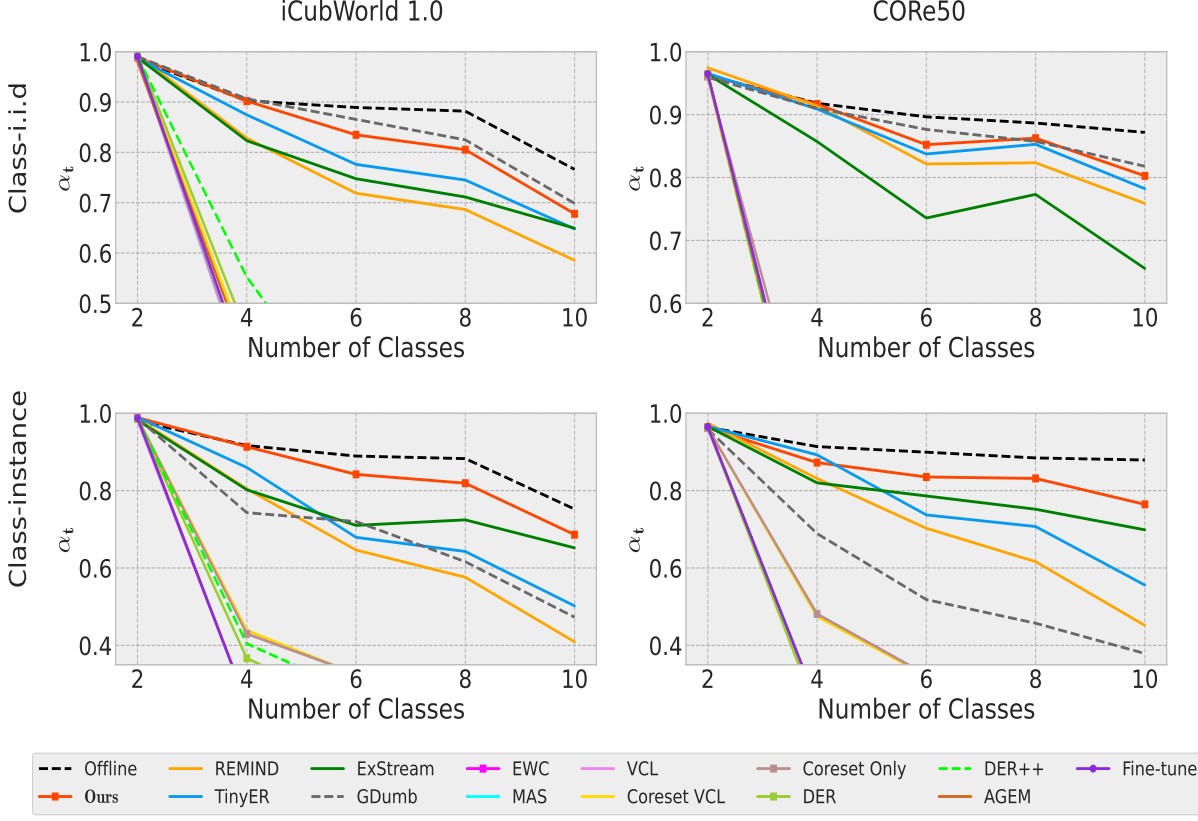

Figure 13: The patial zoon-in version of the plot in Figure 3. It shows the performance of various incremental learning models on (*i*) streaming class-i.i.d (top row) and (*ii*) streaming class-instance (bottom row) ordering on iCubWorld 1.0 & CORe50 dataset. The plots suggest BISLERi (Ours) remembers earlier classes better than most existing algorithms. The performance gain is even more pronounced in streaming class-instance ordering setting (bottom row) where the baseline incremental learners suffer from severe forgetting. Recall that *GDumb cannot be considered as a streaming learning algorithm* as it requires fine-tuning.

## N   Evaluation Over Different Data Orderings Additional Details

The proposed approach (BISLERi) is robust to various *streaming learning scenarios* that can induce catastrophic forgetting (French, 1999; McCloskey & Cohen, 1989). We evalaute the model's class-incremental streaming learning ability with the four challenging data ordering (Hayes et al., 2019b;a) schemes: (*i*) *'streaming iid'*, (*ii*) *'streaming class iid'*, (*iii*) *'streaming instance'*, and (*iv*) *'streaming class instance'*. We described this four data ordering schemes in Section 5.1.

**Note:** Only iCubWorld 1.0 (Fanello et al., 2013), and CORe50 (Lomonaco & Maltoni, 2017) contain the temporal coherent image sequences, therefore, *'streaming instance' 'streaming class instance'* setting are evaluated only on this two datasets.

In the below, we describe the following: (*i*) how the base initialization is performed, and (*ii*) how the network is trained in *streaming setting* according to various data ordering schemes on different datasets.

### N.1   CIFAR10

CIFAR10 (Krizhevsky et al., 2009) is a standard image classification dataset. It contains 10 classes with each consists of 5000 training images and 1000 testing images. Since, it does not contain any temporally ordered image sequence, we use CIFAR10 to evaluate the *streaming learner's* ability in *streaming i.i.d* and *streaming class-i.i.d* orderings.

- **streaming i.i.d:** For the base initialization, we randomly select 2% samples from the dataset and train the model in offline manner. Then we randomly shuffle the remaining samples and train the model incrementally with these samples by feeding one at a time in a streaming manner.

- **streaming class-i.i.d:** In base initialization, the model is trained in a typical offline mode with the samples from the first two classes. Then, in each incremental step, we select the samples from the next two classes, which are not included earlier. These samples are randomly shuffled and fed into the model in a streaming manner.

### N.2   CIFAR100

CIFAR100 (Krizhevsky et al., 2009) is another standard image classification dataset. It contains 100 classes with each consists of 500 training images and 100 testing images. We use CIFAR100 to evaluate the model's ability in *streaming i.i.d* and *streaming class-i.i.d* orderings.

- **streaming i.i.d:** In this setting, we follow the similar approach as mentioned for the CIFAR10 dataset, with the only exception is that the base initialization is performed with 10% randomly chosen samples, and the remaining samples are used for streaming learning.

- **streaming class-i.i.d:** This approach also follows the similar approach as mentioned for the CIFAR10 dataset. However, in each incremental step, including the base initialization, we use samples from 10 classes. For the base initialization, we select samples from the first ten classes, and in each incremental step, we select samples from the succeeding ten classes which are not observed earlier.

### N.3   ImageNet100

ImageNet100 is a subset of ImageNet-1000 (ILSVRC-2012) (Russakovsky et al., 2015) that contains randomly chosen 100 classes, with each classes containing $700-1300$ training samples and 50 validation samples. Since, for test samples, we do not have the ground truth labels, we use the validation data for testing the model's accuracy. We provide more details on ImageNet100 in Section G.

We use ImageNet100 dataset to evaluate the model's ability in *streaming i.i.d* and *streaming class-i.i.d* orderings.

- **streaming i.i.d:** In this case, we follow the similar approach as mentioned for CIFAR100 *streaming i.i.d* ordering.

- **streaming class-i.i.d:** We follow the similar approach as has been mentioned for CIFAR100 *streaming class-i.i.d* ordering.

## N.4  iCubWorld 1.0

iCubWorld 1.0 (Fanello et al., 2013) is an object recognition dataset containing the sequence of video frames, with each frame containing only a single object. It is a more challenging and realistic dataset w.r.t the other standard datasets such as CIFAR10, CIFAR100, and ImageNet100. Technically, it is an ideal dataset to evaluate a model's performance in *streaming learning* scenarios that are known to induce catastrophic forgetting (French, 1999; McCloskey & Cohen, 1989), as it requires learning from temporally ordered image sequences, which are naturally *non-i.i.d* images.

It contains 10 classes, each with 3 different object instances with $200 - 201$ images each. Overall, each class contains $600 - 602$ samples for training and $200 - 201$ samples for testing. Figure 7 shows example images of the 30 object instances in iCubWorld 1.0, where each row denotes one of the 10 categories.

We use iCubWorld 1.0 to evaluate the performance of the streaming learning models in all the four data ordering schemes, i.e., ($i$) *streaming i.i.d*, ($ii$) *streaming class-i.i.d*, ($iii$) *streaming instance*, and ($iv$) *streaming class-instance*.

- **streaming i.i.d:** In this setting, we follow the similar approach as mentioned for the CIFAR10 dataset, with the only exception, that is, 10% randomly selected samples are used for the base initialization, and the rest are used for streaming learning.

- **streaming class-i.i.d:** In this case, we follow the same strategy as mentioned for CIFAR10 *streaming class-i.i.d* ordering.

- **streaming class-instance:** In base initialization, the model is trained in a typical offline mode with the samples from the first two classes. In each incremental step, the network is trained in a streaming manner with the samples from the succeeding two classes which were not observed earlier. However, in this case, ($i$) samples within a class are temporally ordered based on different object instances, and ($ii$) all samples from one class are fed into the network before feeding any samples from the other class.

- **streaming instance:** For the base initialization, 10% randomly chosen samples are used, and the remaining samples are used to train the model incrementally with one sample at a time. In streaming setting, the samples are temporally ordered based on different object instances. Specifically, we organize the data stream by putting temporally ordered 50 frames of an object instance, then we put temporally ordered 50 frames of the second object instance, and so on. In this way, after putting 50 temporally ordered frames from each object instance, we put the next 50 temporally ordered frames of the first object instance and follow the earlier approach until all the frames of each instance have been exhausted.

## N.5  CORe50

CORe50 (Lomonaco & Maltoni, 2017), specifically designed for **C**ontinual **O**bject **Re**cognition, contains a collection of 50 domestic object instances belonging to 10 different catagories: plug adapters, mobile phones, scissors, light bulbs, cans, glasses, balls, markers, cups and remote controls. Each object instance contains 2393-2400 sample images for training. Overall, each class contains 11983-12000 samples for training and 4495-4500 samples for testing. Figure 8 shows example images of the 50 object instances in CORe50, where each column denotes one of the 10 categories.

CORe50 is a challenging dataset, similar to iCubWorld 1.0 (Fanello et al., 2013). It contains temporally coherent image sequences, which are divided into 11 distinct sessions (8 indoors and 3 outdoors) characterized by different backgrounds and lighting. Technically, it is also an ideal dataset for streaming learning

evaluations, aside from iCubWorld 1.0 (Fanello et al., 2013). It can be used to evaluate a model's robustness in all four streaming learning scenarios, i.e., i.i.d, class-i.i.d, class-instace and instance ordering.

We use CORe50 to evaluate the performance of the streaming learning models in all four data ordering schemes, i.e., (*i*) *streaming i.i.d*, (*ii*) *streaming class-i.i.d*, (*iii*) *streaming instance*, and (*iv*) *streaming class-instance* ordering.

- ***streaming i.i.d:*** In this case, we follow the similar approach as mentioned for iCubWorld 1.0 *streaming i.i.d* ordering.

- ***streaming class-i.i.d:*** We follow the similar approach as has been mentioned for iCubWorld 1.0 *streaming class-i.i.d* ordering.

- ***streaming class-instance:*** In this case, we follow the similar strategy as we use for iCubWorld 1.0 *streaming class-instance* ordering.

- ***streaming instance:*** For base initialization, 10% randomly chosen samples are used, and the remaining samples are used to train the model incrementally with one sample at a time. In streaming setting, the samples are temporally ordered based on different sessions and different object instances. Specifically, the dataset is divided into 11 sessions depending on different backgrounds and lighting with each session containing temporally coherent image of from various object instances one after another. We use the data-ordering as provided in *paths.pkl* file with CORe50 dataset for training with remaining samples in the streaming manner.

## O  Derivation of Joint Posterior

$$
\begin{aligned}
\mathcal{L}_t^1(\boldsymbol{\theta}) &= \underset{q \epsilon \mathcal{Q}}{\arg\min}\, \mathrm{KL}\left[q_t(\boldsymbol{\theta}) \,||\, \frac{1}{Z_t} q_{t-1}(\boldsymbol{\theta}) p(\mathcal{D}_t|\boldsymbol{\theta}) p(\mathcal{D}_{\mathcal{M},t}|\boldsymbol{\theta})\right] \\
&\simeq \underset{q \epsilon \mathcal{Q}}{\arg\min}\, \mathrm{KL}\left[q_t(\boldsymbol{\theta}) || q_{t-1}(\boldsymbol{\theta}) p(\mathcal{D}_t|\boldsymbol{\theta}) p(\mathcal{D}_{\mathcal{M},t}|\boldsymbol{\theta})\right] \\
&= \int q_t(\boldsymbol{\theta}) \log \frac{q_t(\boldsymbol{\theta})}{q_{t-1}(\boldsymbol{\theta}) p(\mathcal{D}_t|\boldsymbol{\theta}) p(\mathcal{D}_{\mathcal{M},t}|\boldsymbol{\theta})} d\boldsymbol{\theta} \\
&= -\int q_t(\boldsymbol{\theta}) \log \frac{q_{t-1}(\boldsymbol{\theta}) p(\mathcal{D}_t|\boldsymbol{\theta}) p(\mathcal{D}_{\mathcal{M},t}|\boldsymbol{\theta})}{q_t(\boldsymbol{\theta})} d\boldsymbol{\theta} \\
&= \arg\max \int q_t(\boldsymbol{\theta}) \log \frac{q_{t-1}(\boldsymbol{\theta}) p(\mathcal{D}_t|\boldsymbol{\theta}) p(\mathcal{D}_{\mathcal{M},t}|\boldsymbol{\theta})}{q_t(\boldsymbol{\theta})} d\boldsymbol{\theta} \\
&= \int q_t(\boldsymbol{\theta}) \log p(\mathcal{D}_t|\boldsymbol{\theta}) p(\mathcal{D}_{\mathcal{M},t}|\boldsymbol{\theta}) d\boldsymbol{\theta} + \int q_t(\boldsymbol{\theta}) \log \frac{q_{t-1}(\boldsymbol{\theta})}{q_t(\boldsymbol{\theta})} d\boldsymbol{\theta} \\
&= \int q_t(\boldsymbol{\theta}) \log p(\mathcal{D}_t|\boldsymbol{\theta}) d\boldsymbol{\theta} + \int q_t(\boldsymbol{\theta}) \log p(\mathcal{D}_{\mathcal{M},t}|\boldsymbol{\theta}) d\boldsymbol{\theta} - \int q_t(\boldsymbol{\theta}) \log \frac{q_t(\boldsymbol{\theta})}{q_{t-1}(\boldsymbol{\theta})} d\boldsymbol{\theta} \\
&= \mathbb{E}_{\boldsymbol{\theta} \sim q_t(\boldsymbol{\theta})}\left[\log p(\mathcal{D}_t|\boldsymbol{\theta})\right] + \mathbb{E}_{\boldsymbol{\theta} \sim q_t(\boldsymbol{\theta})}\left[\log p(\mathcal{D}_{\mathcal{M},t}|\boldsymbol{\theta})\right] - \mathrm{KL}\left[q_t(\boldsymbol{\theta})||q_{t-1}(\boldsymbol{\theta})\right] \\
&= \mathbb{E}_{\boldsymbol{\theta} \sim q_t(\boldsymbol{\theta})}\left[\log p(\mathcal{D}_t|\boldsymbol{\theta})\right] + \mathbb{E}_{\boldsymbol{\theta} \sim q_t(\boldsymbol{\theta})}\left[\log p(\mathcal{D}_{\mathcal{M},t}|\boldsymbol{\theta})\right] - \lambda_1 \mathrm{KL}\left[q_t(\boldsymbol{\theta})||q_{t-1}(\boldsymbol{\theta})\right] \\
&= \arg\max \mathbb{E}_{\boldsymbol{\theta} \sim q_t(\boldsymbol{\theta})}\left[\log p(y_t|\boldsymbol{\theta}, G(\boldsymbol{x}_t))\right] + \sum_{n=1}^{N_1'} \mathbb{E}_{\boldsymbol{\theta} \sim q_t(\boldsymbol{\theta})}\left[\log p(y_{\mathcal{M},t}^{(n)}|\boldsymbol{\theta}, \boldsymbol{z}_{\mathcal{M},t}^{(n)})\right] - \lambda_1 \cdot \mathrm{KL}\left(q_t(\boldsymbol{\theta}) \,||\, q_{t-1}(\boldsymbol{\theta})\right)
\end{aligned}
$$
$$(12)$$

where: (*i*) $\mathcal{D}_t = \{d_t\} = \{(\boldsymbol{x}_t, y_t)\}$, (*ii*) $\mathcal{D}_{\mathcal{M},t} = \{d_{\mathcal{M},t}^{(n)}\}_{n=1}^{N_1'} = \left\{(\boldsymbol{z}_{\mathcal{M},t}^{(n)}, y_{\mathcal{M},t}^{(n)})\right\}_{n=1}^{N_1'}$, (*iii*) $|\mathcal{D}_{\mathcal{M},t}| = N_1' \ll |\mathcal{M}|$, (*iv*) $\mathcal{D}_{\mathcal{M},t} \subset \mathcal{M}$, and (*v*) $\lambda_1$ is a hyper-parameter.

