# OpenReview forum: "BISLERi: Ask Your Neural Network Not To Forget In Streaming Learning Scenarios"
_TMLR — Rejected by TMLR_

### Review · Reviewer_LbnP · 2022-05-11

**Summary Of Contributions:**

The paper studies *streaming learning* in lifelong learning, where it is assumed that data arrives one example at a time, and it is not allowed to revisit data more than once (with the exception of small replay buffer/memory). Moreover, we should be able to evaluate the model at any time without needing extra computation.

The authors propose a Bayesian learning framework for this scenario. The model includes two main parts: a fixed feature extractor and a plastic BNN. Moreover, the learning process consists of an implicit regularizer and self-distillation to keep the prediction of the model for previous and new samples close. Since the proposed method uses a replay buffer, the authors also study various strategies for selecting samples from memory and updating the memory.

**Requested Changes:**

(1) Changing the backbone to a self-supervised pre-trained model (instead of classification) and, if possible, choosing another pre-training dataset to fix the label-overlap problem discussed above.

(2) At least for the top few performers in the experiments, use a larger hyper-parameter grid to make sure that the current hyper-parameters do not favor a specific method.

(3) Measure forgetting of the models to help the follow-up papers compare their method with BISLERi easier.

**Strengths And Weaknesses:**

### Strengths:


(1) The problem this paper addresses, i.e., the streaming learning scenario, is very important.

(2) Overall, the proposed method is sound, the claims are justified, and the experiments are comprehensive.

(3) The paper is well-written, and the experiments are comprehensive.


### Weaknesses:
(1) My main concern is the "feature extractor" part. The authors mention that "the motivation to use a pre-trained feature extractor is that the features learned by the first few layers of the neural networks are highly transferable and not specific to any particular task or dataset and can be applied to several different task(s) dataset(s)". However, the fact that the feature extractor is pre-trained on ImageNet-1K, means that it has already seen and learned the labels similar to the benchmarks are being used. So the model already knows the distribution of vehicles, animals, foods, etc., and then the forgetting is not well-defined here. The same concern is discussed in the CL works that used pre-training [1,2]. While the authors use the same feature extractor for all algorithms, I believe it is important to convince the reader that the performance benefit is not due to pre-training on ImageNet. I believe a better pre-training setup should be what people are doing for the NLP tasks: the source task should be different than the downstream task. For instance, in self-supervised prompting, the model learns the representations by figuring the masked words and learning a language model. Then, in that setup, one can argue that the knowledge of the model is related to the language model, which can be used in different downstream tasks with different data. However, here, the model is pre-trained on the same task (image classification) on a dataset that potentially has the labels of the downstream task. For instance, using a pre-trained SSL model such as SimCLR could address this problem.

(2) It seems from Table 11 that only a  fixed set of hyper-parameters was used. However, it is not clear how important the hyper-parameters could be in the setup for this paper. Recent work suggests that hyper-parameters play a significant role in CL performance [5], and I think providing results for only a specific set of hyper-parameters may not lead to a strong conclusion. I suggest the authors use a larger grid (at least for a few baselines) and compare the results of each method.

(3) The authors use only a single metric (Sec. 5.2) to compare methods. While the other metrics, such as average forgetting per task, may not be suitable for the streaming scenario, still, I believe the authors could use a modified version of the forgetting metric to measure the stability/retention of learners. From Fig 3., it seems like BISLERi also has less forgetting. However, I believe providing a tangible forgetting metric could help the follow-up works.

(4) I think the related work could be expanded to discuss some of the recent works such as [3] and [4].



**References**
[1]  Ramasesh et al. Effect of Scale on Catastrophic Forgetting in Neural Networks. ICLR 2022.
[2] Mehta et al. An Empirical Investigation of the Role of Pre-training in Lifelong Learning. ICML Theory and Foundation of Continual Learning Workshop, 2021. https://arxiv.org/abs/2112.09153
[3] Titsias et al. Functional Regularisation for Continual Learning with Gaussian Processes. ICLR 2020.
[4] Ebrahimi et al. Uncertainty-guided Continual Learning with Bayesian Neural Networks. ICLR 2020
[5] Mirzadeh et al. Understanding the Role of Training Regimes in Continual Learning, NeurIPS 2020.

---

> ### Author Response · Authors · 2022-06-04
> **Responses**
>
> We are thankful to the reviewer for their time and effort in reviewing the paper. We also appreciate all the comments \& suggestions. We try our best to address all the doubts and incorporate the suggestion in the revised version of the paper. We request to the reviewer go through the responses and revised version of the paper. If there if any further queries, please let us know. We hope the responses will convince the reviewer to vote for acceptance.
>
> $\bullet$ \textbf{My main concern is the "feature extractor" part. The authors mention that "the motivation to use a pre-trained feature extractor is that the features learned by the first few layers of the neural networks are highly transferable ...}
>
> Here, we compare the performance of various baselines for class-instance ordering on iCubWorld 1.0 using a feature extractor trained with $(i)$ supervised image-classification loss, and $(ii)$ self-supervised loss. Due to time constraint , we are unable to provide more results, but in the final version we are happy to include an ablation demonstrating the performance of the proposed work as a function of feature extractor trained with supervised loss and self-supervised loss. We also aim to provide an ablation comparing the performance of the proposed work as a function of feature extractor trained with various self-supervised loss.
>
> #################################################################
>
> Method    | Resnet50 (Supervised) | Resnet50 (Self-Supervised - MocoV2)
>
> Fine tune | 0.3282 $\pm$ 0.0003    | 0.3276 $\pm$ 0.0008
>
> VCL         | 0.3281 $\pm$ 0.0003    | 0.3269 $\pm$ 0.0014
>
> Coreset VCL | 0.4331 $\pm$ 0.0296 | 0.3949 $\pm$ 0.0309
>
> GDumb | 0.7455 $\pm$ 0.0219 | 0.4623 $\pm$ 0.0638
>
> AGEM | 0.3280 $\pm$ 0.0006 | 0.3277 $\pm$ 0.0008
>
> TinyER | 0.7369 $\pm$ 0.0282 | 0.6859 $\pm$ 0.0273
>
> ExStream | 0.7919 $\pm$ 0.0200 | 0.6863 $\pm$ 0.0305
>
> REMIND | 0.6754 $\pm$ 0.0293 | 0.6584 $\pm$ 0.0258
>
> Ours | \textbf{0.8303 $\pm$ 0.0313} | \textbf{0.7596 $\pm$ 0.0419}
>
> Offline | 0.8676 | 0.8368
>
> ##################################################################
>
> While we can observe a drop in the final accuracy across methods, it can be observed that BISLERi (Ours) achieves the superior performance compared to the baselines.
>
>
> $\bullet$ \textbf{It seems from Table 11 that only a fixed set of hyper-parameters was used. ...}
>
> We selected the same setup of hyper-parameters, such as optimizer, learning rate, momentum, activation, batch size (in Table11) as mentioned in REMIND[2] paper. We used the same hyperparameters for other baselines as well. We are not doing any explicit hyperparameter search for the proposed method (Table11). Therefore, we strongly believe that our method does not have any extra advantage w.r.t these set of hyperparameters over the baselines. We perform only hyperparameter search only for $\lambda_{2}$, sample selection strategy and memory buffer replacement policy, which is specific to our method.
>
> $\bullet$ \textbf{The authors use only a single metric (Sec. 5.2) to compare methods. While the other metrics, such as average forgetting per task, may not be suitable for the streaming scenario, ...}
>
> We have used $$\boldsymbol{\Omega}_{\text{all}}$$ metric to compare the performance of the baselines including our method. This is a standard metric for comparing the performance of various streaming learners, as used by various approaches, such as ExStream[3], REMIND[2]. In addition, we have also used $\boldsymbol{\mu}_{\text{all}}$ metric (in the appendix, SectionH) to compare the performances of various streaming learning models.
>
> Currently, we are not sure about how to measure forgetting as a metric in the streaming learning setup, we are exploring the same. We are thankful to the reviewer for this suggestion, we will consider this as a future work.
>
>
> $\bullet$ \textbf{I think the related work could be expanded to discuss ...}
>
> We have extended the related work in the revised version of the paper.
>
>
> References
>
> [1] Pietro Buzzega, Matteo Boschini, Angelo Porrello, Davide Abati, and Simone Calderara. Dark experience
> for general continual learning: a strong, simple baseline. arXiv preprint arXiv:2004.07211, 2020.
>
> [2] Tyler L Hayes, Kushal Kafle, Robik Shrestha, Manoj Acharya, and Christopher Kanan. Remind your neural network to prevent catastrophic forgetting. arXiv preprint arXiv:1910.02509, 2019b.
>
> [3] Tyler L Hayes, Nathan D Cahill, and Christopher Kanan. Memory efficient experience replay for streaming
> learning. In 2019 International Conference on Robotics and Automation (ICRA), pp. 9769–9776. IEEE,
> 2019a.

---

> > ### Comment · Reviewer_LbnP · 2022-06-10
> > **Re: Response**
> >
> >
> > I would like to thank the authors for their response.
> >
> > **(pre-training/feature extractor)**: I  believe the new ablation on SSL pre-training is very interesting. I think this deserves further discussion in the paper.
> >
> > **(hyper-parameters)**: I’m not sure if I completely agree with you here. The fact that all baselines use the same hyper-parameter does not necessarily mean they are using the most suitable hyper-parameters. For instance, algorithm A may perform better than algorithm B when the learning rate is lower[5]. I strongly suggest the authors report results on a small grid, at least for one benchmark.

---

> > > ### Author Response · Authors · 2022-06-12
> > > **Re: Response (Additional Results)**
> > >
> > > Thank you for your response. Here, we provide some additional experimental results on hyper-parameter as suggested by the Reviewer, we hope it clarifies the existing doubts.
> > >
> > > * **I believe the new ablation on SSL pre-training is very interesting. I think this deserves further discussion in the paper.**
> > >
> > > We have discussed this new ablation in Section-I (Ablation Study: ${\mu}_{all}$ Results As A Function Of Feature-Extractor) (in the appendix) in the revised version of the paper.
> > >
> > > * **I’m not sure if I completely agree with you here. The fact that all baselines use the same hyper-parameter does not ...**
> > >
> > > Here, we compare the performance of various baselines as a function of learning rate (lr) across: $(i)$ class-instance and $(ii)$ instance ordering on iCubWorld 1.0.
> > >
> > > * Class-instance Ordering
> > >
> > > | Methods | lr = 0.01 | lr = 0.001 | lr = 0.003 |
> > > | -------- | ------------- | ------------- | ------------- |
> > > | Fine-Tune | 0.3258 $\pm$ 0.0022  |  0.3264 $\pm$ 0.0039 |  0.3267 $\pm$ 0.0013  |
> > > | VCL | 0.3246 $\pm$ 0.0024  | 0.3242 $\pm$ 0.0016  |  0.3248 $\pm$ 0.0020  |
> > > | Coreset VCL | 0.4319 $\pm$ 0.0148  | 0.4316 $\pm$ 0.0161  | 0.4288 $\pm$ 0.0157   |
> > > | GDumb |  0.7077 $\pm$ 0.0293 | 0.6983 $\pm$ 0.0273  |  0.6989 $\pm$ 0.0255  |
> > > | TinyER | 0.7346 $\pm$ 0.0287  |  0.7177 $\pm$ 0.0157 | 0.7082 $\pm$ 0.0172   |
> > > | ExStream | 0.7740 $\pm$ 0.0198  | 0.7842 $\pm$ 0.0171  |  0.7848 $\pm$ 0.0149  |
> > > | REMIND | 0.6843 $\pm$ 0.0270  | 0.6783 $\pm$ 0.0227  |  0.6597 $\pm$ 0.0206  |
> > > | **Ours** | **0.8497 $\pm$ 0.0191**  | **0.8416 $\pm$ 0.0262**  |  **0.8458 $\pm$ 0.0186**  |
> > > | Offline | 0.8840  | 0.8877  |  0.8912  |
> > >
> > > * Instance Ordering
> > >
> > > | Methods | lr = 0.01 | lr = 0.001 | lr = 0.003 |
> > > | -------- | ------------- | ------------- | ------------- |
> > > | Fine-Tune | 0.1000 $\pm$ 0.0000  | 0.1121 $\pm$ 0.0265  |  0.1070 $\pm$ 0.0221  |
> > > | GDumb | 0.5134 $\pm$ 0.0413  | 0.4646 $\pm$ 0.0732  |  0.5332 $\pm$ 0.0408  |
> > > | TinyER | 0.6672 $\pm$ 0.0496  | 0.6982 $\pm$ 0.0392  |  0.7071 $\pm$ 0.0251  |
> > > | ExStream | 0.6846 $\pm$ 0.0414  |0.6964 $\pm$ 0.0232   |  0.7138 $\pm$ 0.0160  |
> > > | REMIND | 0.6237 $\pm$ 0.0459  | 0.6582 $\pm$ 0.0358  |  0.6518 $\pm$ 0.0411  |
> > > | **Ours** | **0.7325 $\pm$ 0.0228**  | **0.7141 $\pm$ 0.0271**  | **0.7280 $\pm$ 0.0268**   |
> > > | Offline | 0.7646  | 0.7681  | 0.7551   |
> > >
> > >
> > > In the above two tables, we compare the performance of the baselines including BISLERi (Ours) as a function of learning rate (lr) across: $(i)$ class-instance and $(ii)$ instance ordering on iCubWorld 1.0 dataset. We can observe that BISLERi (Ours) achieves superior performance compared to the baselines while trained with different learning rates.
> > >
> > > We have included this ablation in Section-J (Ablation Study: ${\mu}_{all}$ Results As A Function Of Learning-Rate) (in the appendix) in the revised version of the paper.
> > >
> > > Please let us know, if there is any further confusion, we are happy to provide clarification.

---

> > > > ### Comment · Reviewer_LbnP · 2022-06-20
> > > > **Reply**
> > > >
> > > > Thank you for the additional results. As you mentioned, BISLERi outperforms other methods, but we can see the gap differs across different learning rates. However, the result is interesting and makes the paper stronger.

---

### Review · Reviewer_BQQK · 2022-05-11

**Summary Of Contributions:**

This work proposes a new approach for the restrictive class-incremental streaming learning setting, which combines the (already quite difficult) online learning paradigm of seeing each example only once with the enforcement of immediate updates (i.e. we cannot wait for the whole batch). The authors analyse existing CL methods in the context of this setting and show that many of them do not fulfill these conditions. Then they propose BISLERi, their own approach, which relies on the Bayesian streaming framework as well as self-distillation to perform well in this scenario. The authors show empirically that the proposed method outperforms the baselines in various variations of the setting, as well as the batch and online CL scenarios. Finally, the paper also provides ablations studies on the performance of the method, checking different ways of adding data to the buffer, choosing data from the buffer for the current update, and hyperparameter choices.

**Broader Impact Concerns:**

I don't have any concerns about the ethical implications of this work.

**Requested Changes:**

- [Critical] Including a better explanation for the considered setting and its design decisions. The proposed method is supposed to really shine in this scenario, but at the same time, its various restrictions haven't been properly discussed.
- [Critical] Include better baselines that use batches of multiple examples in each step.
- [Major] I would appreciate a discussion on the points mentioned in the "Various questions" section of the review. In particular, I think answers to some of the points there would be good to include in the paper (e.g. hyperparameter search, ablation on the "Bayesian" part of the method).
- [Major] Discuss the memory and computational overhead of the method.
- [Improvement] Make the taxonomy in Table 1 clearer and better defined.

**Strengths And Weaknesses:**

Strengths:
- The Bayesian framework in this work is nicely motivated, grounded in literature, and makes sense in the proposed setting. Although the self-distillation part is less intuitive, the whole method is fairly simple, which makes it easier to build on top of it. The performance of the proposed method in empirical experiments is satisfactory.
- The experiments in general are thorough. The authors consider multiple datasets and settings within these datasets (iid, class-iid, class-instance, etc.). All of the experiments have been run with multiple seeds (the "main" ones from Table 3 with 10 seeds, which is very nice).
- Additionally, the paper provides ablation studies for the proposed methods, investigating the sensitivity of hyperparameters and various design choices of the algorithms.
- The paper is well written. The description of the setting is mostly clear (I have some doubts about parts of the taxonomy which I described below), the authors provide all the relevant details, and the notation is clear and helpful for understanding the paper.

Weaknesses:
- My main worry in terms of how "interesting" this paper will be for the community is how relevant and important the streaming setting really is. The setting is very restrictive, but the paper does not really explicitly motivate these restrictions. The streaming scenario assumes that the model has to be learned one sample at a time (cannot wait for the whole batch), will see each example "in the stream" exactly once, and does not have any time to "recall" before inference (e.g. as coreset VCL does). At the same time, the restrictions are a bit lax in different areas -- the authors assume that the method can utilize a replay buffer (which is sometimes a bit controversial in CL) and that they can use a pre-trained network with a frozen feature extractor with training being constrained only to fine-tuning the last few layers (I'd say this is also somewhat controversial). These design decisions for the setting are not very well motivated. To be precise, I'm not saying that the setup itself is fundamentally flawed and/or impractical, but I strongly feel that its design should be explained in more detail, in a separate section, or maybe in the appendix. It's definitely challenging but I would like to see a clear explanation of why it's a challenge that is interesting for the community.
- In terms of soundness, I'm not convinced that the baselines used in this paper are as strong as they should be. Training deep neural networks on a single example at a time is obviously very difficult even without any distribution shifts, which makes the problem challenging. At the same time, the method proposed by the authors in practice performs updates using multiple examples, see Section 3.1:
>  likelihood estimation from a single example (...) can bias the model towards the new data disproportionately and can maximize the confusion during inference in class incremental learning setup. We, therefore, propose to estimate the likelihood from both the new example: $\mathcal{D}_t$ and the previously observed examples in order to compute the new posterior: $p(θ | \mathcal{D}_1:t)$.

   The same argument can be applied to the other, non-Bayesian methods, as there we also maximize likelihood (just without any priors over the parameters). As such, a fair comparison in my opinion requires to also allow other approaches to use more examples in their batches. This doesn't have to violate the streaming assumption -- for example, if we want to have a batch size of 10, we could take the "current" example that we just received as well as the 9 previously seen examples, effectively forming a very simple FIFO buffer. Alternatively, the baselines could use the same buffer as the authors' proposed method does. Currently, I'm not convinced whether the gains come from the Bayesian formulation of the problem or simply from the fact that the proposed method has the advantage of using multiple examples to calculate the gradient. In fact, it seems that the methods which increase the batch size in some way (e.g. replay-based methods) perform the best.
- Additionally, BISLERi uses two batches of 16 examples (one for memory-replay and one for knowledge distillation). For a fairer comparison, the results for experience replay method (e.g. TinyER) with batch size of 32 should also be considered.
- Although the taxonomy summarized in Table 1 is rather nice (there are a lot of different settings and methods in CL so I think organizing them is very useful for the community), I think it should be improved:
  - I think the definition of "single-pass", which is crucial for defining the streaming setting, is rather unclear. In Section 1 the paper defines the "single-pass" as being "allowed to utilize the single new example only once". Then you say that MIR and GDumb violate this assumption, as they train on the examples in the buffer multiple times. However, the proposed method also reuses the examples from the memory buffer multiple times. What exactly makes MIR incompatible with the single-pass setting?
  - The "Efficient Buffer Replay" is only really explained in Section 3.4, much later than the rest of the taxonomy. Also, this definition is not really precise - what does it mean to have efficient replay? Does the computation time have to be sub-linear wrt to the size of the buffer? Does it have to be O(1) or can it be O(log n)?
  - I don't quite understand the "Training Time" column. What is n and k? What does $\zeta$ mean exactly? Why the training time is higher for the regularization methods?
  - The "regularize" column is also not defined. What do you exactly qualify as regularization? In particular, in a standard CL setting one could argue that experience replay is a form of regularization (reducing the training accuracy on the current task to increase the validation accuracy in general).
- The paper does not describe the memory and computation requirements for the tested methods. In particular, BNNs usually use two times as many parameters as standard networks (they capture the mean and the standard deviation of each parameter) and are quite a bit slower if we draw multiple parameter samples in the forward pass (e.g. the authors draw 5 samples to estimate uncertainty, see Section 3.2). I think a clear discussion of this overhead would be useful.
- This point is rather subjective and a minor criticism, given that TMLR's guidelines explicitly say to pay less attention to novelty, but I think it'd be good to mention it anyway. In the end, I'm not convinced that the base proposed approach (without self-distillation) is that much different than VCL. The authors argue that VCL was proposed for the task-incremental scenario, but I think if one would like to adapt it to the online class incremental setting, it would be fairly straightforward and would yield a similar method to BISLERi.

Various questions/comments:
- How was the hyperparameter search performed? Also, have you checked how TinyER (and other replay methods) perform with your proposed strategies of sampling from/to the buffer?
- Are you planning to share the codebase? You test a lot of methods and conduct many experiments so this could potentially be a useful resource.
- Have you checked how your method would perform without the Bayesian formulation, i.e. with pure likelihood maximization? Looking at Figure 5, it seems that disabling the self-distillation loss decreases the performance of the method quite a bit (comparing with Table 2, performance then is similar to TinyER). For example, what would happen if you used TinyER + self-distillation loss?
- In Figure 2, why are the results for the online setting worse than for the streaming setting, especially for TinyER, REMIND, and your method? As you put it in the paper, the streaming setting is a more restrictive version of the online setting, so it would make sense for the online version to be better.
- In Figure 2 the batch Coreset VCL performs worse than VCL. This is rather surprising -- do you have a hypothesis why this happens?
- The DER results for CIFAR10/100/ImageNet100 are (almost) completely random. Why does that happen?
- In Equations 5, 6, and 7 you sample the logits by sampling from the distribution over parameters, but you use a point estimate for the past logits. How do you gather these past logits? Are they generated by sampling a single parameter set? Are they a mean estimate (i.e. for the distribution of each parameter we use the mean)? How much of a difference does it make?
- In the sample selection for replay, you select samples with the lowest uncertainty/lowest loss values. Wouldn't this lead to the "rich get richer" phenomena (examples with the lowest loss get replayed often so their loss is even lower) and severely limit the number of examples in the buffer that we see?
- In Section 3.4 (Efficient Buffer Update) it seems that some of the examples might get stale. That is, we only update the loss and uncertainty when the example is replayed and we choose the examples based on the value of the loss, then it might be the case that some of the examples might get updated extremely rarely. Do you have any analysis on that or some intuition on whether this is a serious problem?
- In the experimental section you mention that some of the results are missing due to "compatibility issues". Could you elaborate on that, i.e. is this due to something fundamental or rather because of some technical issues?

---

> ### Author Response · Authors · 2022-06-04
> **Responses 1**
>
> We are thankful to the reviewer for their time and effort in reviewing the paper. We also appreciate all the comments \& suggestions. We try our best to address all the doubts and incorporate the suggestion in the revised version of the paper. We request to the reviewer go through the responses and revised version of the paper. If there if any further queries, please let us know. We hope the responses will convince the reviewer to vote for acceptance.
>
> $\bullet$ \textbf{My main worry in terms of how "interesting" this paper will be for the community is how relevant and important the streaming setting really is. The setting is very restrictive, but the paper does not really explicitly motivate these restrictions. The streaming scenario ...}
>
> Importance of class-incremental streaming learning (CISL) or streaming learning (SL) can be described as follows:
>
>
>
>  $(i)$ It enables practical deployment of the AI agents in real world scenarios, where an AI agent might need to learn from as few as a single (training) example without suffering from the catastrophic forgetting. For example, consider an autonomous car might meet with a rare incident/accident, then it could be lifesaving if it can be trained continuously with that single example without any forgetting. It would be impractical, if not infeasible, to wait and aggregate a batch of samples to train the autonomous agent, as we may not collect a batch of such examples due to its rare nature. Hayes et al., 2019b[5] refered to streaming learning as the closest alternative to the biological learning than the other existing lifelong learning approaches, due to the fact that it enables continual learning from a single example with no forgetting.
>
>  $(ii)$ IBL methods assume the data available in batches and can visit the data multiple times to enable CL. While it can be applicable in a static environment, it lacks the applicability in a rapidly changing dynamic environment, where a learner needs to adapt quickly in a single pass, such that, it achieves \emph{any-time-inference} ability. Although, the existing online learning approaches aim to enable continual learning in a dynamic environment from a non-stationary data-stream, these methods have number of limitations, such as: $(i)$ require batch-size, $(ii)$ require fine-tuning before each inference, $(iii)$ require large replay buffer, etc., which limits their applicability in a restrictive streaming lifelong learning. Streaming learning approaches addresses the limitations of the existing IBL and online learning approaches and enables lifelong learning following various constraints: $(i)$ single pass learning, $(ii)$ subset buffer replay, $(iii)$ tiny replay buffer, etc. It further enables \emph{any-time-inference} ability in a continual learner, which enables practical applicability in real world scenarios. Finally, it also enables lifelong learning from a temporally coherent video sequences (images), which are naturally non-i.i.d images. Please refer to Section2 in REMIND[5] paper that has defined this problem formulation to emphasize that we have not invented a new problem formulation, it was already existing.
>
>
>
>
>
>
> Implementation choices:
>
> We follow the similar implementation choices as mentioned in ExStream[4], REMIND[5]. We separate the convolutional neural network into two parts: $(i)$ the non-plastic feature extractor, and $(ii)$ a plastic neural network. We frame the plastic neural network as a Bayesian neural network and trains it incrementally to enable streaming lifelong learning. We study propose various sample selection strategies and memory buffer replacement policies. We initialize the feature extractor with the parameters learned through visual representation learning, as similar to ExStream[4], REMIND[5]. We validate the proposed components with various ablations. Experimental results demonstrate the superiority of the proposed method.

---

> > ### Comment · Reviewer_BQQK · 2022-06-12
> > **RE: Responses 1**
> >
> > Thank you for your detailed answers and changes introduced in the paper, I really appreciate time and effort the authors put into preparing the response. In summary, I think the response answered many of my questions, for which I'm grateful. However, I think there are still important issues in terms of soundness, in particular the baselines strength, which have not been sufficiently addressed in my opinion.
> >
> > I will respond separately in each thread.
> >
> > The description of this setting is clear now, thank you. In the end, I decide not to question the validity of the setting any further, given this is very subjective and TMLR guideines ask us mostly to evaluate whether the paper would be interesting to the community. Given that the setting is quite challenging, that the previous works in this setting have been accepted to top conferences, and that other reviewers find the streaming scenario interesting, I think it is, in general, a topic interesting to the community.
> >
> > I still have some doubts about the practicality of this setting, although I will try avoid being influenced by them for the final decision. Finally, in various parts of their response, the authors defend their choices by pointing out that previous work has used the same settings, i.e. ExStream and REMIND. I don't find this argument fully compelling on its own -- both of these works come from the same lab and, as the authors mention, there is not much work in this setting besides these two. As such, I don't think the "rules" for the streaming setting and its practicality have been settled, in some sense. As such, I think it's still important to motivate the design choices carefully and discuss them.

---

> > > ### Author Response · Authors · 2022-06-16
> > > **RE: Responses 1 (Motivation/Importance of Streaming Learning)**
> > >
> > > We agree as observed by the reviewer that this particular setting has not seen much work from the community except for ExStream and REMIND. However, this particular setting we believe will increasingly be important practically. For instance work on "[Towards Streaming Perception](https://www.ecva.net/papers/eccv_2020/papers_ECCV/papers/123470460.pdf)" work, ECCV 2020 by Li et al and the related [CVPR 2021 workshop](https://cvpr2021.wad.vision/) ([Streaming Perception Challenge website](https://eval.ai/web/challenges/challenge-page/800/overview)) the authors aim to optimize the inference task as that has important implications with respect to latency and accuracy. In our work and those of ExStream and REMIND, the aim is to also enable updates for the classifier using the streaming set of samples. Further work explored by "[RODEO: Replay for Online Object Detection](https://arxiv.org/pdf/2008.06439.pdf)" from the same group as ExStream and REMIND show similar abilities for object detection in streaming setting where updates of the detector are possible. Given these works, we believe that this setting will be of importance for the community.

---

> ### Author Response · Authors · 2022-06-04
> **Responses 2**
>
> $\bullet$ \textbf{In terms of soundness, Im not convinced that the baselines used in this paper are as strong as they should be. Training deep neural networks on a single example at a ... As such, a fair comparison in my opinion requires ... }
>
> To the best of our knowledge, ExStream[4] (ICRA-2019) and REMIND[5] (ECCV-2020) are the only methods which trains a deep neural network in the challenging streaming learning setup, which we use as baseline to compare with the proposed method. As also metioned in the paper, REMIND achieves state-of-the-art performance in the streaming learning setup. DER/DER++[1] (Neurips-2020) is closely related to the proposed method due to its use of knowledge-distillation and experience-replay, therefore, we select this as another baseline. Since, TinyER[3] uses experience replay, therefore, we also use it as a baseline. VCL/Coreset VCL[6] is a continual learning method, which uses Bayesian framework for enabling CL in the deep neural networks, therefore, we use it as a baseline. EWC[7], MAS[8] are highly popular methods, which uses regularization to enable CL; while it overcomes catastrophic forgetting in a static incremental batch learning setup, in this paper we demonstrate it suffers severely in streaming learning setup.
>
>
> For all the rehearsal-based baselines (AGEM, DER/DER++, TinyER, ExStream, REMIND) including the proposed method, we have replayed the same number of examples, i.e., $N^{\prime}_{1} = 16$ past examples, along with the new example throughout all the experiments in order to enable CL, as also mentioned in Section5.5. Coreset VCL, Coreset Only, GDumb uses all the samples stored in memory to fine-tune the network parameters before each inference. EWC, MAS, VCL do not use any memory replay.
>
> In this paper, we do not utilize FIFO buffer due to the fact that the input data-stream can be temporal coherent, for example, instance and class-instance ordering on iCubWorld 1.0 and CORe50 dataset, where the data-stream is organized by temporal ordered video frames from different object instances. In this case, the AI agent might observe a sequence of video frames (images) with very less to no difference at all. Thus, if we use a FIFO buffer then the replay samples might have little to no difference, therefore, replaying such samples will not help in overcoming catastrophic forgetting.
>
>
> $\bullet$ \textbf{Additionally, BISLERi uses two batches of 16 examples (one for memory-replay and one for knowledge distillation) ...}
>
>
> We follow the standard setup as mentioned by DER++[1] (Neurips 2020), which is closely related to our proposed method. DER++ utilizes a batch for experience-replay and another batch for knowledge-distillation, while comparing the performance with ER[2]/TinyER[3] which uses only a single batch of examples for experience-replay. ER/TinyER optimizes the parameters w.r.t the cross-entropy loss, where there is no notion of optimizing the parameters w.r.t the soft-labels, i.e., knowledge-distillation loss. Therefore, if we give ER/TinyER a batch having twice the size of the batch used for experience replay in our method, it will have an extra advantage and the comparison will no longer be a fair comparison. To demonstrate the importance of knowledge distillation in our models performance, we have conducted ablation in Figure5, which demonstrates the effect of knowledge-distillation in the models performance.

---

> > ### Comment · Reviewer_BQQK · 2022-06-12
> > **RE: Responses 2**
> >
> > - Thank you for the explanation. However, I would still argue that checking the FIFO buffer setting is crucial, as it brings us closer to the online setting while still abiding by the rules of the streaming learning setting. That is, if we apply this "rolling window" strategy of replay with a FIFO buffer, then we are essentially back in the online continual learning setting, with the difference being that in streaming learning there is a large overlap between subsequent batches, and in online learning the batches would be completely disjoint. Additionally, I do not think that this solution would address the problem of catastrophic forgetting, rather I think it would stabilize the training process in general -- SGD with just a single datapoint is extremely noisy and hard to optimize. And even if this approach fails completely in the setting where the data is temporally correlated, then it should work at least somewhat better in the class-iid scenarios which you also test. As such, I strongly feel this is an essential experiment.
> > - I might have missed something and I'm sorry if that's the case, but I don't quite see why using 32 examples for ER is unfair if you are also using 32 examples. If I have a batch of 32 labeled examples, then (a) using all 32 of them for replay or (b) using 16 of them for replay and 16 for knowledge distillation both seem like reasonable decisions. It would be good to check which one performs better.

---

> > > ### Author Response · Authors · 2022-06-16
> > > **RE: Responses 2 (Additional Results)**
> > >
> > > * **Thank you for the explanation. However, I would still argue that checking the FIFO buffer setting is crucial, as it brings us closer to the online setting while still abiding by the rules of the streaming learning setting. That is, if we apply this "rolling window" strategy of replay with a FIFO ...**
> > >
> > > In the below table, we compare the performance ($\mu_{\text{all}}$ Results) of BISLERi (Ours) and a streaming learning agent using a FIFO buffer (FIFO) across: $(i)$ iid, $(ii)$ class-iid, $(iii)$ instance and $(iv)$ class-instance ordering on iCubWorld 1.0.
> > >
> > > | Methods | iid | Class-iid | instance | Class-instance
> > > | :----:  | :----:  | :----:  | :----:  | :----:  |
> > > | FIFO | 0.6764 $\pm$ 0.0488  | 0.4499 $\pm$ 0.0045  |  0.1030 $\pm$ 0.0095 | 0.3260 $\pm$ 0.0025 |
> > > | **Ours** | **0.7409 $\pm$ 0.0107**  | **0.8417 $\pm$ 0.0187**  |  **0.7325 $\pm$ 0.0228**  | **0.8497 $\pm$ 0.0191** |
> > > | Offline | 0.7626 | 0.8849 | 0.7646 | 0.8840|
> > >
> > > It can be observed that FIFO model suffers severely (perhaps not surprisingly) on instance and class-instance ordering (We have discussed this in earlier reply as well). Furthermore, it also performs poorly on class-iid ordering. FIFO model performs comparatively with respect to the other baselines only on iid data, however, it still performs poorly w.r.t the baselines including BISLERi (Ours).
> > >
> > > * **I might have missed something and I'm sorry if that's the case ...**
> > >
> > > In the below table, we compare the performance ($\mu_{\text{all}}$ Results) of TinyER and a modified of version of our method, which we call BISLERi-- (Ours--). This modified version of our proposed method (Ours--) *selects a single subset of samples from memory* (where the subset size is **16**, as similar to the number of samples used for memory replay in case of TinyER) and *uses the same subset of samples for both $(i)$ memory replay and $(ii)$ snap-shot self-distillation (knowledge-distillation)*.
> > >
> > > | Methods | iid | Class-iid | instance | Class-instance
> > > | :----:  | :----:  | :----:  | :----:  | :----:  |
> > > | TinyER |  0.7313 $\pm$ 0.0289 | 0.8062 $\pm$ 0.0257  | 0.6672 $\pm$ 0.0496  | 0.7346 $\pm$ 0.0287 |
> > > | **Ours--** | **0.7481 $\pm$ 0.0192**  | **0.8285 $\pm$ 0.0287**  |  **0.6975 $\pm$ 0.0261** | **0.8095 $\pm$ 0.0263** |
> > > | Offline | 0.7626 | 0.8849 | 0.7646 | 0.8840|
> > >
> > > It can be observed that despite the fact BISLERi-- (Ours--) uses the same single subset of samples for both $(i)$ memory replay and $(ii)$ snap-shot self-distillation (knowledge-distillation), BISLERi-- (Ours--) achieves superior performance compared to the TinyER across all four data-orderings on iCubWorld 1.0. Furthermore, for class-instance and instance ordering, it obtains improvement upto 7.49% and 3.03% respectively.

---

> ### Author Response · Authors · 2022-06-04
> **Responses 3**
>
> $\bullet$ \textbf{I think the definition of "single-pass", which is crucial for defining the streaming setting, is rather unclear. ...}
>
> In online/streaming learning, each newly available (training) sample is only allowed to observe only once without storing it in a memory (replay buffer), and requires to be adapted in a single gradient update. This is refered as single-pass learning. In each incremental step, however, it is allowed to replay past observed samples stored in memory along with the newly available data. Please also refer to REMIND[5] Section2, where they have defined this single pass learning formulation to emphasize that we have not invented a new problem formulation, it was already existing.
>
>
>
> In addition, in online learning, it is allowed to fine-tune the network with the stored samples by repeating the fine-tuning for multiple epochs, multiple times. However, this implies that the network would use multiple gradient update instead of a single gradient update to improve its performance, which is essentially forbidden in streaming learning.
>
>
> MIR[9] violates the single pass learning constraint of streaming learning, by employing a two step/pass learning strategy. Initially, it uses the newly available sample(s) to perform a parameter update to select the maximally interfered past stored samples from memory to be used for experience replay. Finally, it combines the new available sample(s), already used once for a gradient update, with the selected maximally interfered samples to perform another (final) gradient update. Therefore, MIR essentially uses a two step/pass learning, instead of a single pass learning as required in streaming learning. For more details on the streaming learning constraints refer to Section2.
>
>
> GDumb[10] requires fine-tuning the network parameters for multiple epochs, multiple times with the stored replay buffer samples before each inference, as it does not employ any learning when it observes a new sample in each incremental step. It implies that GDumb requires multiple gradient update to improve its performance, ultimately violates the single pass learning constraint.
>
> $\bullet$ \textbf{The "Efficient Buffer Replay" is only really explained}
>
> To remove the confusion, we have updated this to \emph{subset buffer replay}. We descibe its importance with an example below.
>
> For example, ExStream[4] uses memory-replay to enable streaming learning, however, in doing so, it replays all the stored samples along with the newly available sample in each incremental step. While it mitigates catastrophic forgetting in the network, it limits its practical applicability due to obvious reasons. That is, if the buffer capacity is considerably large then time required to complete a single gradient update will also be large. On the other hand, the methods like REMIND[5], DER/DER++[1], BISLERi (Ours) uses subset buffer replay to enable streaming learning. That is, it select only a few samples from memory, combines them with the newly available sample in order to perform single gradient update to enable continual learning, which is computationally an efficient choice.
>
> $\bullet$ \textbf{I dont quite understand the "Training Time" column.}
>
> Training time column in Table1 denotes the number of gradient updates required by the corresponding method according to the underlying assumption that method impose. Therefore, $\zeta(n)$ denotes that the corresponding method would require $n$ gradient update in order to enable continual learning.
>
> For example, EWC[7], MAS[8], VCL/Coreset VCL[6] visits the data multiple times, performs multiple gradient update, to enable continual learning, therefore, its training time is represented with $\zeta(n)$. On the other hand, methods such as DER/DER++[1], REMIND[5], BISLERi (Ours) can adapt to the newly available in a single gradient update, hence, the training time is $\zeta(1)$.
>
> $\bullet$ \textbf{The "regularize" column is also not defined.}
>
> To remove confusion, we have updated this to \emph{regularization based}. That is, if a method qualifies as a regularization based method, then it uses parameters regularization to enable continual learning.
>
> Parsi et al. 2019[11], Delange et al. 2021[12] have classified the existing continual learning approaches on the basis of the mechanisms for mitigating catastrophic forgetting into three main categories, namely: $(i)$ parameter isolation based approaches, $(ii)$ regularization based approaches, and $(iii)$ rehearsal / memory-replay based approaches. In this paper, we have followed this same classification to classify the existing CL approaches into one of those three classes.

---

> > ### Comment · Reviewer_BQQK · 2022-06-12
> > **RE: Responses 3**
> >
> > Thank you for explaining the definitions in Table 1, this addressed most of my doubts. There are however a few details that are not clear to me yet:
> >
> > - I think MIR can still be considered a single-pass method in that meaning, at least when efficiently implemented (and I think it is this way in [the original implementation](https://github.com/optimass/Maximally_Interfered_Retrieval/blob/master/mir.py)). That is, we first compute the gradients $g_{new}$ on $D_{new}$ and check the interference scores after the update to get the dataset $D_{interferred}$. Then, we would like to compute the gradient of $D_{new} \cup D_{interferred}$. However, we already have the gradient $g_{new}$ so all we need is to compute the gradient over $D_{interferred}$ to get the gradient for the final update. As such, MIR processes the example only once. I imagine GDumb could also be considered single-pass, if we decided to train on the data only for a single epoch before the inference.
> >
> > - Training time - I still don't quite get this distinciton, as EWC, MAS and VCL can be applied to the online CL setting (we have to set the epoch number to 1), the same way as DER can be applied to the multiple-epoch CL setting. Is there a more fundamental difference here, or is it more a question on which settings they were initially used for?

---

> > > ### Author Response · Authors · 2022-06-16
> > > **RE: Responses 3**
> > >
> > > * **I think MIR can still be considered a single-pass method in that meaning, at least when efficiently implemented ...**
> > >
> > > We agree with the reviewer that the modified MIR, as suggested by the reviewer can be considered as a single pass method. That is, first compute the gradient $g_{new}$ on $D_{new}$ and check the interference score after the gradient update (virtual gradient update) to get the dataset $D_{interferred}$. Then, compute the gradient of $D_{new} \cup D_{interferred}$. However, as we already have the gradient w.r.t $g_{new}$ so all we need is to compute the gradient over $D_{interferred}$ to get the gradient for the final update.
> > >
> > >
> > >
> > > While the above modification is appealing, we have to remember that in case of streaming learning, $|D_{new}| = 1$, i.e., in each incremental step only a single training example arrives. Therefore, the gradient $g_{new}$ will be computed w.r.t a single data-point $D_{new}$, where $|D_{new}| = 1$. Since, *SGD with just a single datapoint is extremely noisy and hard to optimize*, thus, we believe that, the gradient update (virtual gradient update) with $g_{new}$ will result in poor generalization, henceforth, the selected maximally interfered dataset ($D_{interferred}$) will be sub-optimal, such that, it will suffer from catastrophic forgetting in streaming lifelong learning setup.
> > >
> > > We will discuss about the modified MIR in the revised version of the paper.
> > >
> > > * **I imagine GDumb could also be considered single-pass, if we decided to train on the data only for a single epoch before the inference.**
> > >
> > > In the below table, we compare the performance of GDumb fine-tuned with a single epoch and 30 epochs before each inference.
> > >
> > > | Methods | # Of Epochs Used<br/> For Training|  iid | Class-iid | instance | Class-instance
> > > | :----:  | :----:  | :----:  | :----:  | :----:  | :----:  |
> > > | GDumb | 1  |  0.2866 $\pm$ 0.0732 | 0.5173 $\pm$ 0.1115  | 0.2414 $\pm$ 0.0722 | 0.4793 $\pm$ 0.1139  |
> > > | GDumb | 30  | 0.6858 $\pm$ 0.0315  | 0.8571 $\pm$ 0.0175  | 0.5134 $\pm$ 0.0413 | 0.7077 $\pm$ 0.0293  |
> > >
> > >
> > > We can observe that single epoch fine-tuned GDumb performs poorly compared to GDumb fine-tuned with 30 epochs before each inference.
> > >
> > >
> > > * **Training time - I still don't quite get this distinciton, as EWC, MAS and VCL can be applied to the online CL setting ...**
> > >
> > >
> > > Training time column in Table1 denotes the number of gradient updates required by the corresponding method according to *the underlying assumption that method impose in their respective papers*. Therefore, $\zeta(n)$ denotes that the corresponding method would require $n$ gradient update in order to enable continual learning.
> > >
> > >
> > > Since EWC, MAS ([the original implementation](https://github.com/rahafaljundi/MAS-Memory-Aware-Synapses/blob/master/MAS_to_be_published/MAS_utils/MAS_based_Training.py), [demo](https://github.com/rahafaljundi/MAS-Memory-Aware-Synapses/blob/master/MAS_to_be_published/Demo.ipynb)), VCL/Coreset VCL ([the original implementation](https://github.com/nvcuong/variational-continual-learning/blob/master/ddm/alg/cla_models_multihead.py)) *visits the data multiple times in their respective setting in their respective paper*, performs multiple gradient update, to enable continual learning, therefore, its training time is represented with $\zeta(n)$, where $n$ is the total number of gradient update used for training.
> > >
> > >
> > > We have adapted EWC, MAS, VCL in the streaming lifelong learning setup, by performing a single gradient update in each incremental step. Coreset VCL is also adapted by computing a single gradient update in each incremental step, however, it still fine-tunes the network with the stored replay buffer samples before each inference.

---

> ### Author Response · Authors · 2022-06-04
> **Responses 4**
>
> $\bullet$ \textbf{The paper does not describe the memory and computation requirements for the tested methods. ...}
>
> Here, we provide: $(i)$ gpu memory consumption, and $(ii)$ total time required by the corresponding method for streaming learning on iCubWorld 1.0 for class-instance ordering.
>
> ##########################################################
>
>     Method | GPU Memory Consumption (Mb) | Time
>
>     Fine-Tune | 933 | 34m 03s
>
>     EWC | 933 | 41m 23s
>
>     MAS | 933 | 54m 16s
>
>     VCL | 933 | 53m 25s
>
>     DER/DER++ | 933 | 61m 25s
>
>     TinyER | 933 | 54m 19s
>
>     REMIND | 933 | 63m 21s
>
>     Ours | 986 | 73m 25s
>
> ############################################################
>
>
>
> It can be observed that the gpu memory consumption and required time to train in streaming learning, is not significantly higher than the other baselines. However, it results in superior performance compared to the baselines.
>
> We draw 5 samples of parameters from the distribution over the parameters to estimate uncertainty-score, however, in the rest of the training pipeline, we utilize a single parameter sampled from the distribution over the parameters. It does not increase the time required in streaming learning significantly.
>
> $\bullet$ \textbf{This point is rather subjective and a minor criticism, given that TMLRs guidelines explicitly say to pay less attention to novelty, but I think itd be good to mention it anyway. ...}
>
> Pleae refer to Differences Between VCL/Coreset VCL and BISLERi for comparison between VCL/Coreset VCL and BISLERi in the rebuttal to Reviewer DNZj. We have included these points in the appendix in the revised submission.
>
>
> $\bullet$ \textbf{How was the hyperparameter search performed?}
>
> We have performed grid search for hyper-parameter selection for the proposed method.
>
> $\bullet$ \textbf{Are you planning to share the codebase?}
>
> We intend to share the codebase upon acceptance.
>
>
> $\bullet$ \textbf{Have you checked how your method would perform without the Bayesian formulation, i.e. with pure likelihood maximization? Looking at Figure 5, it seems that disabling the self-distillation loss decreases the performance of the method quite a bit (comparing with Table 2, performance then is similar to TinyER). For example, what would happen if you used TinyER + self-distillation loss?
> }
>
> Bayesian formulation does not help only the robust learning with few/one samples but also in sample selection strategy. In the appendix (Table5, 6, 7) we have shown the results over the various sample selection methodology, for past sample replay. From the Table-7 (appendix) we can observe that for the $\lambda_2=0.3$ Uncertainty-Aware Positive-Negative Sampling (UAPN) (Section3.2) strategy outperforms the other settings. UAPN requires to calculate uncertainty and it is calculated with the help of Bayesian network. If we have likelihood maximization model only then we have to choose Uniform or Loss-Aware Positive-Negative Sampling technique for the past samples selection for the replay, which degrades the models performance. Currently, because of the time constraint we are unable to provide the result for the non-bayesian model, but in the final version we are happy to include the same. TinyER+self-distillation+Bayesian will be very similar to proposed approach and we expect a similar performance as ours. Only TinyER+self-distillation expected to show the similar results as uniform sampling that is shown in the Table5,6,7 (appendix).
>
> $\bullet$ \textbf{In Figure 2, why are the results for the online setting ...}
>
> Perhaps, this is due to the fact that, streaming learning reaches better minima than the online learning because of the total number of experience replay. Since, in online learning, data arrives in a batch having size greater than one, the model observes the whole dataset in a fewer number of incremental-steps than the number of incremental-steps required in streaming learning, which implies that online learning uses fewer number of experience-replays, than the streaming learning algorithms, resulting in a poor generalization than the streaming learning.
>
> $\bullet$ \textbf{In Figure 2 the batch Coreset VCL performs worse than VCL. ...}
>
> Here, we provide the numerical values corresponding to Figure2 plot comparing the performance of VCL and Coreset VCL on class-iid ordering on iCubWorld 1.0:
>
> #################################################################################
>
>     Metric | VCL | Coreset VCL
>
>     $\boldsymbol{\Omega}_{\text{all}}$ Results | 0.5493 $\pm$ 0.0372 | 0.5314 $\pm$ 0.0306
>
>     $\boldsymbol{\mu}_{\text{all}}$ Results | 0.5019 $\pm$ 0.0327 | 0.4859 $\pm$ 0.0269
>
> #################################################################################
>
> Although, VCL performs slightly better than Coreset VCL, it can be observed that the performance improvement is not significant.

---

> > ### Comment · Reviewer_BQQK · 2022-06-12
> > **RE: Responses 4**
> >
> > - GPU memory - Thank you! I think this is a nice additional experiment, could you add it to the paper as well?
> >
> > - BiSLERI vs VCL - as mentioned, I will not push this point, as novelty is not considered critical for TMLR. However, I would like to ask the authors, if possible, to discuss Bayesian Gradient Descent [1] in the paper and how it relates to your method. I apologize I didn't mention it in my original review, I only came across it very recently. It seems quite relevant, but on the other hand I am aware that I'm asking about this somewhat late into the review process.
> >
> > - Hyperparameter grid - But what about the baseline methods, did you perform some kind of grid search there? I think you already answered this question in your discussion with Reviewer LbnP, but I would like to ask you to add it to the paper as well.
> >
> > - Codebase, Bayesian formulation, results for online setting - Thank you for clarifying!
> >
> > - VCL vs Batch Coreset - it still seems surprising to me, as in general it seems like coresets are quite helpful (e.g. GDumb). I don't have anymore questions here though, thank you!
> >
> > [1] Zeno, Chen, et al. "Task agnostic continual learning using online variational bayes." arXiv preprint arXiv:1803.10123 (2018).

---

> > > ### Author Response · Authors · 2022-06-16
> > > **RE: Responses 4**
> > >
> > > * **GPU memory - Thank you! I think this is a nice additional experiment, could you add it to the paper as well?**
> > >
> > > We have discussed this in the revised version of the paper.
> > >
> > > * **BiSLERI vs VCL - as mentioned, I will not push this point, as novelty is not considered critical for TMLR. However, I would like to ask the authors, if possible, to discuss Bayesian Gradient Descent ...**
> > >
> > > We have extended the related work in the revised version of the paper to discuss the mentioned paper.
> > >
> > > * **Hyperparameter grid - But what about the baseline methods, did you perform some kind of grid search there? I think you already answered this question in your discussion with Reviewer LbnP, but I would like to ask you to add it to the paper as well.**
> > >
> > > We have mentioned the hyper-parameters used for the baselines in Section 5.5 ("Implementation Details") in the main paper. We have used the hyper-parameter which results in the best performance as mentioned by the respective authors in the corresponding paper.
> > >
> > > The reviewer LbnP have concerns about whether the chosen learning rate (lr) gives an advantage to our method and negatively effects the performance of the compared baselines. For which, we have conducted an ablation study, which compares the performance ($\mu_{\text{all}}$ Results) of the proposed method BISLERi (Ours) and the baselines across $(i)$ class-instance and $(ii)$ instance ordering on iCubWorld 1.0 over different learning rates (lr). From that comparison, we can observe BISLERi (Ours) achieves superior performance compared to the baselines    while trained with different learning rates. We have included this ablation in the revised version of the paper.

---

> ### Author Response · Authors · 2022-06-04
> **Responses 5**
>
> $\bullet$ \textbf{The DER results for CIFAR10/100/ImageNet100 are ...}
>
> We believe this due to the fact that DER/DER++[1] do not update the logits in an online manner like we do, and uses suboptimal (old) logits to optimize the parameters during the streaming learning, which results in poor generalization.
>
>
> $\bullet$ \textbf{In Equations 5, 6, and 7 you sample the logits by sampling from the distribution over parameters, ...}
>
>
> We sample a single parameter $(\theta)$ from the distributions over the parameters, and use it to gather the logits. We also use a single parameter $(\theta)$ sampled from the distributions over the parameters for training. Only for estimating the uncertainty-score, we sample $k = 5$ parameters from the distribution over the parameters.
>
>
> $\bullet$ \textbf{In the sample selection for replay, you select samples with the lowest uncertainty/lowest loss values. ...}
>
> Both the \emph{uncertainty-aware positive-negative sampling} and the \emph{loss-aware positive-negative sampling}, selects $N^{\prime}\_{1}/2$ samples with lowest uncertainty/loss value, and $N^{\prime}_{1}/2$ samples with highest uncertainty/loss value. For both the cases, we observe that replaying the most and the least certain samples results better than uniform sampling. We have mentioned this in Section3.2. Furthermore, during \emph{memory-buffer replacement}, we weigh each buffer sample inversely w.r.t the loss-value, such that, lesser the loss the more likely the sample to be removed from the buffer. We believe that this prevents "rich get richer" phenomena.
>
>
> $\bullet$ \textbf{In Section 3.4 (Efficient Buffer Update) it seems that some of the examples might ...}
>
> It is a fact that uncertainty/loss values corresponding to each replay buffer sample is only updated if they are selected for rehearsal, therefore, some of the examples might get updated extremely rarely. While this is a disadvantage, it is worth mentioning that if we avoid online update as mentioned in Section3.4, and replace it with an offline update, where for each stored sample, we compute the uncertainty and loss value after observing a single new example in each incremental step, it would require doing a forward pass for all the stored examples through the plastic network, which would slow down the whole learning pipeline.

---

> > ### Comment · Reviewer_BQQK · 2022-06-12
> > **RE: Responses 5**
> >
> > Thank you very much for clarifying these points.

---

> ### Author Response · Authors · 2022-06-04
> **Responses 6**
>
> $\bullet$ \textbf{In the experimental section you mention that some of the results are missing due to "compatibility issues". ...}
>
> This is more of a technical/fundamental issue. These methods assume that in each incremental step samples from the new classes will arrive, and samples from the previously observed classes is not observed during the incremental step. However, in case of streaming-i.i.d and streaming instance ordering, sample from a previously observed class can also arrive during any future incremental step in streaming learning.
>
> References
>
> [1] Pietro Buzzega, Matteo Boschini, Angelo Porrello, Davide Abati, and Simone Calderara. Dark experience
> for general continual learning: a strong, simple baseline. arXiv preprint arXiv:2004.07211, 2020.
>
> [2] Matthew Riemer, Ignacio Cases, Robert Ajemian, Miao Liu, Irina Rish, Yuhai Tu, and Gerald Tesauro. Learning to learn without forgetting by maximizing transfer and minimizing interference. arXiv preprint arXiv:1810.11910, 2018.
>
> [3] Arslan Chaudhry, Marcus Rohrbach, Mohamed Elhoseiny, Thalaiyasingam Ajanthan, Puneet K Dokania,
> Philip HS Torr, and Marc’Aurelio Ranzato. On tiny episodic memories in continual learning. arXiv
> preprint arXiv:1902.10486, 2019.
>
> [4] Tyler L Hayes, Nathan D Cahill, and Christopher Kanan. Memory efficient experience replay for streaming
> learning. In 2019 International Conference on Robotics and Automation (ICRA), pp. 9769–9776. IEEE,
> 2019a.
>
> [5] Tyler L Hayes, Kushal Kafle, Robik Shrestha, Manoj Acharya, and Christopher Kanan. Remind your neural network to prevent catastrophic forgetting. arXiv preprint arXiv:1910.02509, 2019b.
>
> [6] Cuong V Nguyen, Yingzhen Li, Thang D Bui, and Richard E Turner. Variational continual learning. arXiv preprint arXiv:1710.10628, 2017.
>
> [7] James Kirkpatrick, Razvan Pascanu, Neil Rabinowitz, Joel Veness, Guillaume Desjardins, Andrei A Rusu,
> Kieran Milan, John Quan, Tiago Ramalho, Agnieszka Grabska-Barwinska, et al. Overcoming catastrophic
> forgetting in neural networks. Proceedings of the national academy of sciences, 114(13):3521–3526, 2017.
>
> [8] Rahaf Aljundi, Francesca Babiloni, Mohamed Elhoseiny, Marcus Rohrbach, and Tinne Tuytelaars. Memory
> aware synapses: Learning what (not) to forget. In Proceedings of the European Conference on Computer
> Vision (ECCV), pp. 139–154, 2018a.
>
> [9] Rahaf Aljundi, Lucas Caccia, Eugene Belilovsky, Massimo Caccia, Min Lin, Laurent Charlin, and
> Tinne Tuytelaars. Online continual learning with maximally interfered retrieval. arXiv preprint
> arXiv:1908.04742, 2019.
>
> [10] Ameya Prabhu, Philip HS Torr, and Puneet K Dokania. Gdumb: A simple approach that questions our progress in continual learning. In European Conference on Computer Vision, pp. 524–540. Springer, 2020.
>
> [11] German I Parisi, Ronald Kemker, Jose L Part, Christopher Kanan, and Stefan Wermter. Continual lifelong
> learning with neural networks: A review. Neural Networks, 113:54–71, 2019.
>
> [12] Matthias Delange, Rahaf Aljundi, Marc Masana, Sarah Parisot, Xu Jia, Ales Leonardis, Greg Slabaugh,
> and Tinne Tuytelaars. A continual learning survey: Defying forgetting in classification tasks. IEEE
> Transactions on Pattern Analysis and Machine Intelligence, 2021.

---

### Review · Reviewer_DNZj · 2022-05-22

**Summary Of Contributions:**

The paper proposes a class incremental learning algorithm in a streaming setting, called BISLERi. The streaming setting assumes that the data arrives one example at a time and that the learner can not revisit the data point again – the so-called single pass setting – except for a small number of data points stored in the episodic memory. The main idea is to cast the learning problem in the Bayesian framework and learn the posterior over the model parameters in a sequential setting – using the previous posterior as the prior for the new training step. Towards this, similar to an already existing work variational continual learning (VCL), the authors make use of variational inference to approximate the posterior. The overall training objective consists of two loss functions; 1) variational ELBO where the log likelihood is computed on the current data and the replay buffer of the past and the KL is between the current approximate posterior and the posterior approximated at the previous step (similar to VCL), 2) a distillation loss that ensures that the network predictions at the current step remain close to the network predictions at the previous steps. Since a replay buffer is used to store the data samples of the past, the authors also study strategies to populate and sample from the replay buffer. Comprehensive experiments on various benchmarks and against various baselines demonstrate the competitiveness of the approach.


**Broader Impact Concerns:**

No ethical implications as per my knowledge.

**Requested Changes:**

Please clarify why VCL didn’t work in the class-incremental setting in your experiments? What is the non-trivial change in the ELBO computation of BISLERi that VCL didn’t have? The noted \theta_S and \theta_t VCL differences in the appendix are cosmetic as per understanding. Using the same setup, single-head and F_(\theta), and how would the VCL update be different from the BISLERi update.


**Strengths And Weaknesses:**

*Strengths*:

1. The paper is written very well and easy to follow.
2. The final model is well-motivated.
3. The experiments are very thorough and demonstrate competitiveness of the approach.

*Weaknesses*:

*Difference with the VCL*:

1. I feel that the main idea, and in fact algorithm, is very similar to VCL. VCL updates the coreset at each step by selecting a subset from the current task and a subset from the replay buffer of previous tasks. It then uses two step approximation of the posterior; 1) using the current data and replay buffer it approximates the posterior, 2) this posterior is refined using the coreset before prediction. Whereas BISELRi updates the posterior from both the current data and replay buffer in one step and there is no finetuning before prediction. I don’t think the fine tuning step is such a big difference and this makes me question the novelty of this work.

2. The authors did mention some differences. Fine-tuning, as mentioned already, is not a big difference. The other difference of class- vs task-incremental is also not very significant. From what it looks, BISELRi can be an application of VCL in the class-incremental setting. Had this application required a non-trivial change in the main VCL algorithm, I would have not raised this point. But I don’t see any significant change, at least not in how variational ELBO is computed. The other loss term, knowledge distillation, seemed ad-hoc and can also be added to the VCL. Besides, Fig 5 shows that even without this loss term the performance is quite competitive.

3. With the above two points in mind, I wonder why the reported VCL numbers are so poor compared to BISLERi.

---

> ### Author Response · Authors · 2022-06-04
> **Responses 1**
>
> We are thankful to the reviewer for their time and effort in reviewing the paper. We also appreciate all the comments \& suggestions. We try our best to address all the doubts and incorporate the suggestion in the revised version of the paper. We request to the reviewer go through the responses and revised version of the paper. If there if any further queries, please let us know. We hope the responses will convince the reviewer to vote for acceptance.
>
> $\bullet$ \textbf{I feel that the main idea, and in fact algorithm, is very similar to VCL. ...}
>
> Both VCL/Coreset VCL[1,2] do not use any memory replay while approximating the new posterior. VCL/Coreset VCL combines the previously computed posterior with the new data-likelihood to compute the new posterior (refer Eq. 4 in VCL[1]), whereas the proposed method (BISLERi) combines the past posterior with the data-likelihood, which is computed by combining a subset of replay buffer samples with the newly available sample, to approximate the posterior in each incremental step. Since, the samples are not stored in memory in a task-specific manner, and sampled across all the observed classes, it enables BISLERi to achieve \emph{any-time-inference} ability, which is a key requirement in streaming learning, as fine-tuning is forbidden in streaming learning. Please refer to REMIND[5] paper that has defined this problem formulation to emphasize that we have not invented a new problem formulation, it was already existing.
>
> Coreset VCL stores a few samples in the memory in a task-specific manner, and uses only the samples corresponding to current task for fine-tuning the network[2]. While this strategy works well in task-incremental learning setup, it results in poor generalization in the class-incremental learning setup.
>
>
> $\bullet$ \textbf{The authors did mention some differences. Fine-tuning, as mentioned already, is not a big difference. The other difference of class- vs task-incremental is also not very significant. ...}
>
>
> \emph{Class-incremental learning} is by far challenging than task-incremental learning setup. Class-incremental learning requires a classifier to learn to discriminate between different class-labels from different tasks. In class incremental learning, at the test time, the task identifier $t$ is not specified, and the accuracy is computed over all the observed classes with $\frac{1}{\mathcal{C}}$ chance, where $\mathcal{C}$ is the total number of classes accumulated so far. However, in task incremental learning, the task identifier $t$ is known.
>
> For example, consider MNIST divided into $5$ tasks: $\left\{ \left\{ 0, 1 \right\}, \dots, \left\{ 8, 9 \right\}  \right\}$, which are used for sequential learning of a classifier. Then, at the end of $5$-th task, in \emph{task incremental setting}, the classifier needs to predict a class out of $\left \{ 8, 9 \right \}$ only. However, in \emph{class incremental setting}, a class label is predicted over all the ten classes that is observed so far, i.e., $\left\{ 0, \dots, 9 \right\}$ with $\frac{1}{10}$ chance for each class. This setting is  challenging and pertinent to the continual learning setting. Solving this is critical to generalizing the continual learning framework.
>
>
> VCL/Coreset VCL follows task-incremental learning, whereas BISLERi follows class-incremental learning setup, which is considerably a challenging variant of continual learning. BISLERi does not use any explicit fine-tuning, still achieves \emph{any-time-inference} ability in the restrictive class-incremental streaming learning (CISL) setup.

---

> > ### Comment · Reviewer_DNZj · 2022-06-23
> > **Final thoughts after the rebuttal**
> >
> > Thank you very much for your response and clarifying some of the points.
> >
> > After reading the response, I still feel that the work is an application of VCL in the so-called class-incremental setting. Whereas, VCL replays the data of previous tasks (present in the coreset) to fine-tune the posterior at prediction time, this work, instead, incorporates the previous data during current posterior approximation. This in itself is not a big change and perhaps the obvious thing to do if one wants a single/ any-time predictor.
> >
> > The rest of the differences in how coresets are constructed -- whether its a task-specific or class-specific -- are not that important because even with the task-specific coresets, at prediction time, one could fine-tune the posterior on all the tasks' coresets and get a predictor that works without the task id.

---

> > > ### Author Response · Authors · 2022-06-26
> > > **Re: Responses**
> > >
> > > * **After reading the response, I still feel that the work is an application of VCL in the so-called class-incremental setting. Whereas, VCL replays the data of previous tasks (present in the coreset) to fine-tune the posterior ...**
> > >
> > > We agree with the reviewer that the basic formulation (Bayesian formulation) is motivated by VCL, however, the proposed changes (contribution) show the significant impact (anytime inference ability/superior performance) in the streaming learning setting over VCL/Coreset VCL. While the proposed contribution is simple, its impact is relevant and offers superiority over baselines in streaming lifelong learning. We believe that if a simple method demonstrates significant performance gain, it is more practical and will be highly beneficial to the community. Therefore, these kind of methods should get preference over any complex methods with similar performance.
> > >
> > >
> > > * **The rest of the differences in how coresets are constructed -- whether its a task-specific or class-specific -- are not that important because even with the task-specific coresets, at prediction time, one could fine-tune the posterior on all the tasks' coresets and get a predictor that works without the task id.**
> > >
> > > We politely disagree with the reviewer that the same approach suggested by the reviewer (*"one could fine-tune the posterior on all the tasks' coresets and get a predictor that works without the task id."*) is followed by GDumb[1]. We have evaluated the performance of GDumb[1] in the streaming learning setup (Table 3, 13) in the paper.
> > >
> > > It can be observed that GDumb[1], even though it violates the streaming learning constraint and fine-tunes the network with all the stored replay buffer samples before each inference, it does not achieve superior performance throughout all the data orderings (iid, instance, and class-instance ordering). It only achieves superior performance on class-iid ordering. However, it should be noted that it violates the single-pass learning constraint as it fine-tunes the network before each inference.
> > >
> > >
> > > [1] Ameya Prabhu, Philip HS Torr, and Puneet K Dokania. Gdumb: A simple approach that questions our progress in continual learning. In European Conference on Computer Vision, pp. 524–540. Springer, 2020.

---

> ### Author Response · Authors · 2022-06-04
> **Responses 2**
>
> $\bullet$ \textbf{Differences Between VCL/Coreset VCL and BISLERi}
>
>
>  $(i)$ While BISLERi and VCL/Coreset VCL[1,2] both utilizes Bayesian framework to enable continual learning in the deep neural networks, VCL/Coreset VCL is a incremental batch learning (IBL) mathod in nature, whereas BISLERi is a streaming/online learning method. That is, in order to approximate the posterior in each incremental step, VCL/Coreset VCL requires visiting the data multiple times, whereas BISLERi approximates the posterior with a single gradient update. In doing so we need to obtain important modifications to obtain correct estimates of likelihood and updation of the posterior. Naively using VCL can be observed to perform quite inferior to the proposed solution. Our work is a principled adaptation of the formulation to the streaming learning setting and this is quite different from the continual learning based on batch-based updates.
>
>  $(ii)$ Both VCL and Coreset VCL do not utilize any memory replay, while approximating the new posterior, whereas BISLERi replays a subset of the past stored samples along with the newly available sample in order to approximate the new posterior to enable continual learning. Approximating the posterior in this way, i.e., replaying a subset of past samples with the new sample, allows BISLERi to achieve \emph{any-time-inference} ability, which is a key-requirement in streaming learning, as fine-tuning the network parameters with the stored samples is forbidden in the streaming learning setup.
>
>  $(iii)$ While VCL do not use coreset samples during any step of the learning, Coreset VCL withholds a few past samples in memory (coreset), which are then used for fine-tuning the network before inference. However, Coreset VCL stores the samples in the coreset in a \emph{task-specific manner}, unlike the methods like GDumb[3], TinyER[4], REMIND[5], BISLERi (Ours). We explain this with the below example.
>  Consider MNIST divided into 5 tasks: $\{ \{ 0, 1 \}, \{ 2, 3 \}, \dots, \{ 8, 9 \} \}$, which are used for sequential learning of a classifier. Therefore, in each incremental step, the classifier observes sample from only two classes. In this case, Coreset VCL stores samples in memory (coreset) in a task-specific manner. That is, it divides the coreset into 5 partitions, where each partition is used to store samples from a single task. Before inference, samples corresponding to the current task is utilized to fine-tune the network parameters to improve performance. For example, at the end of 5-th task, since the classifier only needs to predict a class out of $\{ 8, 9 \}$, Coreset VCL fine-tunes the network with the withheld samples corresponding to only class $\{ 8, 9 \}$. While this strategy works nicely in case of task-incremental learning, it suffers severely in class-incremental learning setup.
>
>  In contrast methods like GDumb[3], TinyER[4], REMIND[5], BISLERi (Ours), do not store the samples in memory in a task-specific manner, instead it is populated with the samples from all the classes. Therefore, when methods like GDumb, REMIND, BISLERi replays the past samples, it observe samples across all the classes irrespective of the tasks, whereas Coreset VCL only observes samples corresponding to the specific task, which causes Coreset VCL to suffer from poor generalization in case of class-incremental learning setup.
>
>
>  $(iv)$ Coreset VCL[1,2] withhold few data-points from the dataset, and do not utilize them during the incremental learning. These withheld samples are only used for fine-tuing the network parameters before inference. In contrast BISLERi maintains a replay buffer which is updated in an online manner as mentioned in Section3.3. During streaming learning, in each incremental step a subset of samples are selected (Section3.2) and combined with the newly available sample to compute the new posterior, which enables the network with the \emph{any-time-inference} ability, a crucial property required in streaming learning.

---

> ### Author Response · Authors · 2022-06-04
> **Responses 3**
>
> $\bullet$ \textbf{How VCL/Coreset VCL is adapted in the Streaming Learning?}
>
> For both VCL and Coreset VCL, we have used a single-headed Bayesian network as the plastic network $(F)$, as also mentioned in Section5.5. We follow the same strategy as mentioned by [1,2] to approximate the new posterior in each incremental step by combining the previously computed posterior with the new data-likelihood. For Coreset VCL, the samples are stored in memory (coreset) in a task-specific manner, while arriving one datum at a time in each incremental step. At the end of each task, Coreset VCL selects the samples specific to the current task and fine-tunes the network parameters. While fine-tuning the network is forbidden in streaming learning, it does not improve the networks overall performance in class-incremental learning setup due to the above mentioned reasons.
>
>
> References
>
> [1] Cuong V Nguyen, Yingzhen Li, Thang D Bui, and Richard E Turner. Variational continual learning. arXiv preprint arXiv:1710.10628, 2017.
>
> [2] Sebastian Farquhar and Yarin Gal. Towards robust evaluations of continual learning. arXiv preprint arXiv:1805.09733, 2018.
>
> [3] Ameya Prabhu, Philip HS Torr, and Puneet K Dokania. Gdumb: A simple approach that questions our progress in continual learning. In European Conference on Computer Vision, pp. 524–540. Springer, 2020.
>
> [4] Arslan Chaudhry, Marcus Rohrbach, Mohamed Elhoseiny, Thalaiyasingam Ajanthan, Puneet K Dokania,
> Philip HS Torr, and Marc’Aurelio Ranzato. On tiny episodic memories in continual learning. arXiv
> preprint arXiv:1902.10486, 2019.
>
> [5] Tyler L Hayes, Kushal Kafle, Robik Shrestha, Manoj Acharya, and Christopher Kanan. Remind your neural network to prevent catastrophic forgetting. arXiv preprint arXiv:1910.02509, 2019b.

---

### Comment · Action_Editors · 2022-06-20
**Hyperparameter selection, underfitting**

Dear Authors,

I wanted to ask a few questions about hyperparameter selection and optimization:

1. You mention that you used grid search to tune hyperparameters of BISLERi, but reused best hyperparameters from papers proposing baselines. However, this seems to favour the proposed method, because the setting is generally rarely used. Could you please add hyperparameter selection for selected methods that are closely related (and also including DER). Please also remember about including learning rate in the grid. Tuning learning rate (and more broadly optimization related hyperparameters) seems critical in streaming setting.

2. Could you report a measure of underfitting that makes most sense? It is important to disentangle underfitting from catastrophic forgetting, and other effects that impact final performance.

3. In this context, why is DER marked as using "single pass"? From the original paper (https://arxiv.org/pdf/2004.07211.pdf), it seems that the Authors are actually training on a larger batch of examples, or is my understanding incorrect?

Thank you,
AC

---

> ### Author Response · Authors · 2022-06-26
> **Responses: Hyperparameter Search**
>
> * **You mention that you used grid search to tune hyperparameters of BISLERi, but reused best hyperparameters from papers proposing ...**
>
> In Section L (Figure 9) ("Ablation Study: $\mu_{all}$ Results As A Function Of Hyperparameters ($\alpha$ and $\beta$) On DER/DER++") in the revised version of the paper, we have compared the performance ($\mu_{all}$ Results) of DER/DER++ w.r.t the different values of hyperparameters ($\alpha$ & $\beta$) and the learning rate (lr) on class-instance ordering on iCubWorld 1.0.
>
> * For $\beta = 0.0$ and $\alpha = 0.0$, DER++ behaves similar to Fine-Tune model (lower bound), where it updates the parameters of the network with the gradient computed against the newly available single training example in each incremental step.
>
> * For $\beta = 0.0$ and $\alpha > 0.0$, DER++ becomes DER. It uses only knowledge-distillation to mitigate catastrophic forgetting.
>
> * For $\alpha = 0.0$ and $\beta > 0.0$, DER++ behaves similar to a method using only experience replay to mitigate catastrophic forgetting. DER++ would converge to TinyER, if it uses $\alpha = 0.0$ and $\beta = 1.0$.
>
> * For $\alpha > 0.0$ and $\beta > 0.0$, DER++ uses both knowledge-distillation and experience replay to circumvent the catastrophic forgetting.
>
> Methods such as REMIND, ExStream, Coreset VCL, TInyER are the other closely related rehearsal based lifelong learning approaches which we have used to compare the performance of BISLERi (Ours) in streaming learning setup. These methods however do not use any hyperparameter, which might effect the network optimization during incremental learning. We have already compared the performances of these methods across various learning rates and has reported in Section J (Table 11) ("Ablation Study: ${\mu}_{\text{all}}$ Results As A Function Of Learning-Rate").
>
>
> * **Could you report a measure of underfitting that makes most sense? ...**
>
> Currently, we are not sure about how to measure underfitting as a metric in the streaming learning setup. We could not find any relevant literature, which measures underfitting in the continual learning setup. In continual learning, the available literature trains an Offline model and compares the performance of the lifelong learning agents w.r.t the Offline learner. In this paper, we have followed the same strategy and reported the performance with $\mu_{all}$ (Table 3), $\Omega_{all}$ (Table 13) and $\alpha_{t}$ (Figure 3, 10, 11, 12, 13) metrics.
>
>
> We are thankful to the AE for this suggestion, we will consider this as a future work.

---

> > ### Comment · Action_Editors · 2022-06-27
> > **Direct comparison**
> >
> > Thank you for the additional results. I have two follow-up questions about the experiment in Figure 9.
> >
> > 1. Given that $\alpha+\beta$ control the relative strength of regularization, and that actually the top-right corner performs best, could you please extend the grid so that larger values of $\beta$ are covered?
> >
> > 2. Could you please add for reference performance of your method to Figure 9? It is a bit challenging to cross-reference it.

---

> > > ### Author Response · Authors · 2022-06-27
> > > **Additional Results**
> > >
> > > In the below table, we extend the top-row of both sides (left and right) of Figure 9 experiments and compare the performance ($\mu_{all}$ Results) of DER++ for $\alpha = 0.0$, $\beta \in [0.0, 0.1, 0.2, 0.3, 0.4, 0.5, 0.6, 0.7, 0.8, 0.9, 1.0]$, and lr $\in [0.01, 0.001]$ over class-instance ordering on iCubWorld 1.0.
> > >
> > > It can be observed that DER++ behaves similar to an experience replay based model with increasing value of $\beta$, and achieves the highest final accuracy when $\alpha = 0.0, \beta = 1.0$, for both learning rates (lr $\in [0.01, 0.001]$).
> > >
> > >
> > > | DER++ | lr = 0.01 | lr = 0.001 |
> > > | -------- | ------------- | ------------- |
> > > | $\alpha = 0.0, \beta = 0.0$ | 0.3268 $\pm$ 0.0010 | 0.3246 $\pm$ 0.0060 |
> > > | $\alpha = 0.0, \beta = 0.1$ | 0.3759 $\pm$ 0.0148 | 0.3269 $\pm$ 0.0017 |
> > > | $\alpha = 0.0, \beta = 0.2$ | 0.4515 $\pm$ 0.0322 | 0.3791 $\pm$ 0.0154 |
> > > | $\alpha = 0.0, \beta = 0.3$ | 0.5312 $\pm$ 0.0380 | 0.4406 $\pm$ 0.0236 |
> > > | $\alpha = 0.0, \beta = 0.4$ | 0.5576 $\pm$ 0.0516 | 0.5122 $\pm$ 0.0406 |
> > > | $\alpha = 0.0, \beta = 0.5$ | 0.6043 $\pm$ 0.0437 | 0.5718 $\pm$ 0.0354 |
> > > | $\alpha = 0.0, \beta = 0.6$ | 0.6449 $\pm$ 0.0568 | 0.5745 $\pm$ 0.0349 |
> > > | $\alpha = 0.0, \beta = 0.7$ | 0.6442 $\pm$ 0.0704 | 0.5995 $\pm$ 0.0385 |
> > > | $\alpha = 0.0, \beta = 0.8$ | 0.6817 $\pm$ 0.0709 | 0.6078 $\pm$ 0.0396 |
> > > | $\alpha = 0.0, \beta = 0.9$ | 0.6846 $\pm$ 0.0603 | 0.6127 $\pm$ 0.0466 |
> > > | $\alpha = 0.0, \beta = 1.0$ | 0.7419 $\pm$ 0.0523 | 0.6276 $\pm$ 0.0408 |
> > > | **Ours** | **0.8497 $\pm$ 0.0191**  | **0.8416 $\pm$ 0.0262**  |
> > > | Offline | 0.8840  | 0.8877  |
> > >
> > >
> > > We have included this result in Section L (Table 13) in the revised version of the paper. As suggested by the AE, we have also included the performance of the proposed method in the table.

---

> > ### Comment · Action_Editors · 2022-06-27
> > **REMIND hyperparameters**
> >
> > Could you please check the impact of REMIND hyperparameters on its performance? Please see Table S1 in https://arxiv.org/abs/1910.02509. The authors used different hyperparameters for different tasks, suggesting it is important to tune them.

---

> > > ### Author Response · Authors · 2022-06-27
> > > **REMIND Additional Results**
> > >
> > > Here, we compare the performance of REMIND as a function of learning rate (lr) across: $(i)$ class-instance and $(ii)$ instance ordering on iCubWorld 1.0.
> > >
> > > * Class-instance Ordering
> > >
> > > | Methods | lr = 0.01 | lr = 0.001 | lr = 0.003 |
> > > | -------- | ------------- | ------------- | ------------- |
> > > | REMIND | 0.6843 $\pm$ 0.0270  | 0.6783 $\pm$ 0.0227  |  0.6597 $\pm$ 0.0206  |
> > > | **Ours** | **0.8497 $\pm$ 0.0191**  | **0.8416 $\pm$ 0.0262**  |  **0.8458 $\pm$ 0.0186**  |
> > > | Offline | 0.8840  | 0.8877  |  0.8912  |
> > >
> > > * Instance Ordering
> > >
> > > | Methods | lr = 0.01 | lr = 0.001 | lr = 0.003 |
> > > | -------- | ------------- | ------------- | ------------- |
> > > | REMIND | 0.6237 $\pm$ 0.0459  | 0.6582 $\pm$ 0.0358  |  0.6518 $\pm$ 0.0411  |
> > > | **Ours** | **0.7325 $\pm$ 0.0228**  | **0.7141 $\pm$ 0.0271**  | **0.7280 $\pm$ 0.0268**   |
> > > | Offline | 0.7646  | 0.7681  | 0.7551   |
> > >
> > >
> > > In the above two tables, we compare the performance of the REMIND as a function of learning rate (lr) across: $(i)$ class-instance and $(ii)$ instance ordering on iCubWorld 1.0 dataset. It can be observed that BISLERi (Ours) achieves superior performance compared to REMIND while trained with different learning rates.
> > >
> > > We have included this result in Table 11 (in the appendix) in the paper.

---

> ### Author Response · Authors · 2022-06-26
> **Responses: Single Pass Method**
>
> * **In this context, why is DER marked as using "single pass"? ..**
>
> In online/streaming learning, each newly available (training) sample(s) is only allowed to observe only once without storing it in a memory (replay buffer), and requires to be adapted in a single gradient update. This is refered as single-pass learning. In each incremental step, however, it is allowed to replay past observed samples stored in memory along with the newly available data. Please also refer to REMIND (Section 2), where they have defined this single pass learning formulation to emphasize that we have not invented a new problem formulation, it was already existing.
>
>
>
> In addition, in online learning, it is allowed to fine-tune the network with the stored samples by repeating the fine-tuning for multiple epochs, multiple times. However, this implies that the network would use multiple gradient update instead of a single gradient update to improve its performance, which is essentially forbidden in streaming learning.
>
>
> MIR violates the single pass learning constraint of streaming learning, by employing a two step/pass learning strategy. Initially, it uses the newly available sample(s) to perform a parameter update to select the maximally interfered past stored samples from memory to be used for experience replay. Finally, it combines the new available sample(s), already used once for a gradient update, with the selected maximally interfered samples to perform another (final) gradient update. Therefore, MIR essentially uses a two step/pass learning, instead of a single pass learning as required in streaming learning. For more details on the streaming learning constraints refer to Section 2 ("Problem Formulation").
>
>
> GDumb requires fine-tuning the network parameters for multiple epochs, multiple times with the stored replay buffer samples before each inference, as it does not employ any learning when it observes a new sample in each incremental step. It implies that GDumb requires multiple gradient update to improve its performance, ultimately violates the single pass learning constraint.
>
> *Since, DER/DER++ observes the newly available data only once, it is termed as single pass learning.*
>
> **Note:** In online learning, it is assumed that in each incremental step $t$, the newly available data $D_{t}$ has a size greater than 1, i.e., $|D_{t}| \gg 1$, whereas in streaming learning, it is assumed in each incremental step only a single training example arrives, i.e., $|D_{t}| = 1$. Therefore, DER/DER++ still can be a single pass method while observing a batch of new samples ($|D_{t}| \gg 1$) in online learning setup. We have discussed this in detail in Section 2 ("Problem Formulation") in the paper.

---

### Decision · Action_Editors · 2022-07-06

**Recommendation:** Reject

**Comment:**

The paper introduced BISLERI, a method for class-incremental streaming learning that formulates the problem in the Bayesian framework, in a closely related manner to VCL.

The setting while unpopular would be interesting to some in the audience. The setting was used in several papers such as ExStream and REMIND. Reviewer LbnP said “The problem this paper addresses, i.e., the streaming learning scenario, is very important.”. At the same time, Reviewer BQQK said “My main worry in terms of how "interesting" this paper will be for the community is how relevant and important the streaming setting really is.”. I think this difference in opinions reflects well how the community thinks about the setting.

The key issue relates to the presentation. All the reviewers and the meta-reviewer were confused by the difference of the proposed method to VCL, as well as the relation to other methods. I think a reasonable framework for understanding the paper is that it is an adaptation of VCL to class-incremental streaming learning. In response, Authors say that “VCL/Coreset VCL is a incremental batch learning (IBL) method in nature”, but my understanding is that VCL is a framework and can be in a rather straightfoward manner adapted to streaming learning.

There is also a potential issue with soundness. Authors have tuned hyperparameters only for their method, while taking default hyperparameters for competing methods. While Authors engaged in discussion during the review process, the added experiments were insufficient to fully address this concern: (a) the added experiments were done on a limited set of experiments; (b) they were done in larger scope only for DER++ (for REMIND only the learning rate was tuned); (c) for DER++ the best performing set of hyperparameters (alpha=0, beta=1) was also the highest one checked (beta>1.0 was not examined); (d) no clear comparison to the most closely related VCL (with adaptation to the setting) was included.

Hyperparameter tuning seems quite critical given that (a) we are training one sample at a time. It seems to the meta-reviewer that it might be very easy to underfit (in the sense of loss on each example, a concern also mentioned by one reviewer) in the streaming learning scenario, and (b) REMIND seems to perform quite similarly on certain benchmarks.

Reviewers appreciated the breadth and scope of experiments. They were deemed clearly sufficient in terms of the number of tested datasets, settings, and methods.

Reviewers also appreciated to choice of the method, e.g. Reviewer BQQK says “The Bayesian framework in this work is nicely motivated, grounded in literature, and makes sense in the proposed setting”.

All in all, it is an interesting work but requires major improvements in the presentation to meet the bar acceptance. Most importantly, I would recommend adding a much clearer discussion of the VCL framework (perhaps even rephrasing the work as an adaptation of the VCL framework to the streaming learning scenario) and a more thorough comparison to baselines.

Thank you for considering TMLR and your hard work. I hope the remarks will be helpful in improving your submission. The work would be welcomed as a resubmission to TMLR after making these changes.